# Learning High-Degree Parities: The Crucial Role of the Initialization

**Emmanuel Abbe**
Ecole Polytechnique Fédérale de Lausanne (EPFL)

**Elisabetta Cornacchia**
INRIA Paris, DI ENS, PSL
Massachusetts Institute of Technology (MIT)

**Jan Hązła & Donald Kougang-Yombi**
African Institute for Mathematical Sciences (AIMS), Kigali, Rwanda

## Abstract

Parities have become a standard benchmark for evaluating learning algorithms. Recent works show that regular neural networks trained by gradient descent can efficiently learn degree $k$ parities on uniform inputs for constant $k$, but fail to do so when $k$ and $d-k$ grow with $d$ (here $d$ is the ambient dimension). However, the case where $k = d - O_d(1)$, including the degree $d$ parity (the *full parity*), has remained unsettled. This paper shows that for gradient descent on regular neural networks, learnability depends on the initial weight distribution. On one hand, the discrete Rademacher initialization enables efficient learning of almost-full parities, while on the other hand, its Gaussian perturbation with large enough constant standard deviation $\sigma$ prevents it. The positive result for almost-full parities is shown to hold up to $\sigma = O(d^{-1})$, pointing to questions about a sharper threshold phenomenon. Unlike statistical query (SQ) learning, where a singleton function class like the full parity is trivially learnable, our negative result applies to a fixed function and relies on an *initial gradient alignment* measure of potential broader relevance to neural networks learning.

## 1 Introduction

Initialization plays a crucial role in the performance of neural network training algorithms. It has been shown that a proper initialization can help avoiding issues such as vanishing or exploding gradients, or set the foundation for efficient convergence and improved generalization (He et al. (2015); Glorot & Bengio (2010); Sutskever et al. (2013); Kumar (2017)). In this work, we show that the choice of initialization can be critical when learning complex functions, such as high-degree parities.

Parity functions are a well-known class of challenging problems for differentiable learning models, where the task is to determine the parity of bits belonging to an unknown subset of input coordinates. Due to their inherent non-linearity and extreme sensitivity to small input changes, parity functions often serve as a challenging benchmark for evaluating and comparing learning algorithms, including gradient descent on neural networks (Abbe & Sandon (2020); Daniely & Malach (2020)). For instance, they have been used for showing the advantages of using convolutional architectures over fully connected ones (Malach & Shalev-Shwartz (2020)), the superiority of differentiable models compared to kernel methods (Abbe et al. (2021)), and the efficacy of curriculum learning in contrast to standard training (Abbe et al. (2024b); Cornacchia & Mossel (2023)).

Previous research has mainly focused on the family of *sparse* parities, also known as $k$-parities, where the size of the support of the target parity, $k$, is bounded, i.e., it does not grow with input dimension $d$. It has been shown that on uniform inputs, $k$-parities can be learned by gradient descent algorithms (GD/SGD) on standard architectures, such as 2-layer fully connected (Barak et al. (2022);

Email: `emmanuel.abbe@epfl.ch`, `elisabetta.cornacchia@inria.fr`, `{jan.hazla, donald.yombi}@aims.ac.rw`.

Abbe & Boix-Adsera (2022); Glasgow (2023); Kou et al. (2024)), with a sample complexity of $\tilde{O}(d^{k-1})$[1] (Kou et al. (2024)).

In contrast, for *dense* parities, where the support of the target parity is unbounded ($k = \omega_d(1)$), the picture is less clear. It has been shown that when both $k$ and $d - k$ are unbounded, stochastic gradient descent (SGD) with large batch size and limited gradient precision on fully connected architectures cannot learn dense parities *with any initialization*[2] (Abbe & Sandon (2020)). The difficulty in learning parities stems from their orthogonality on uniform inputs, leading to a low cross-predictability (CP) (Abbe & Sandon (2020)). However, this only occurs if a given class of $k$-parities is sufficiently large. Since the cardinality of this class is $\binom{d}{k} = \binom{d}{d-k}$, this hardness result does not extend to *almost-full* parities, where $k = d - O_d(1)$, including the special case of the single $d$-parity (the *full parity*).

In fact, it is known that the full parity, as a symmetric function, is learnable by gradient descent methods with specific initialization (Nachum & Yehudayoff (2020)), such as setting all first layer weights to $1$. For random and symmetric initializations, Abbe & Boix-Adsera (2022) showed that almost-full parities are weakly-learnable[3] by gradient descent on a two-layer fully connected network with discrete Rademacher initialization.

In this paper, we focus on almost-full parity functions and provide a deeper understanding of how the initialization impacts their learning. First, we show that SGD on a two-layer fully connected ReLU network with Rademacher initialization can achieve perfect accuracy for $k = d - O_d(1)$, thus going beyond weak learning. Next, we investigate the robustness of this positive result and argue that it is a special case. In particular, we prove that with Gaussian initialization GD with limited gradient precision with the correlation loss cannot learn high degree parities on two layer ReLU networks. We then introduce an intermediate case of *perturbed*-Rademacher initialization, where the weights are initialized from a mixture of two Gaussian distributions with means $+1$ and $-1$ and a standard deviation of $\sigma$. In the case of full parity we prove that when $\sigma = O(d^{-1})$, the positive result still holds, while if $\sigma$ is a large enough constant, learning does not occur. We leave the analysis for the remaining range of $\sigma$ and the investigation of a potential threshold to future work. While our theoretical analysis focuses on Gaussian perturbations, our experiments also explore other perturbations, both discrete and continuous, supporting our claim that the success of the Rademacher initialization is a special case. In our experiments, we also explore other settings beyond our theoretical analysis in order to justify the robustness of our findings.

Crucially, the proof technique for our negative result does not rely on constructing an orbit class or using measures for function classes (as in the cross-predictability case). Instead, it introduces a new approach centered on a novel measure, the *initial gradient alignment*, which may be relevant for evaluating the suitability of an initialization for a target distribution beyond the specific parity setting discussed in this paper.

## 2 RELATED WORK

**Learning Parities.** Learning parities on uniform inputs is easy with specialized techniques like Gaussian elimination over two-element fields or through emulation networks trained with Stochastic Gradient Descent (SGD) using small batch sizes (Abbe & Sandon (2020)). However, in the statistical query (SQ) setting (Kearns (1998)) and with gradient descent methods that have limited gradient precision (Abbe & Sandon (2020)), learning parities presents computational barriers. Recent works have focused on sparse parities, or $k$-parities (where $k = O_d(1)$), as a classical benchmark for evaluating learning algorithms (Suzuki et al. (2024); Edelman et al. (2023); Barak et al. (2022); Daniely & Malach (2020); Malach et al. (2021); Abbe & Boix-Adsera (2022); Malach & Shalev-Shwartz (2020); Abbe et al. (2024b); Cornacchia & Mossel (2023); Wei et al. (2019); Ji & Telgarsky (2019)). In particular, in the special case of $k = 2$ (i.e. the XOR problem), Glasgow (2023) proved a sample complexity upper bound of $\tilde{O}(d)$ on a 2-layer network of logarithmic width, while for general $k$, Kou et al. (2024) proved a sample complexity of $\tilde{O}(d^{k-1})$, matching the SQ lower bounds in both cases. For dense parities, it has been established that if both $k$ and $d - k$ grow with $d$, SGD

---

[1]Where $\tilde{O}(d^c) = O(d^c \operatorname{poly} \log(d))$, for $c \in \mathbb{R}$.

[2]Assuming the initialization is invariant to permutation of the input neurons.

[3]i.e., an inverse polynomial edge over the trivial estimator is achieved with constant probability.

with large batch sizes fail at learning in polynomial time (Abbe & Sandon (2020)). We build on top of Abbe & Boix-Adsera (2022), which showed that almost-full parities are weakly-learnable by gradient descent on a two-layer fully connected network with Rademacher initialization. We provide a more complete picture on the role of the initialization for learning the full parity, and we argue that the Rademacher initialization is in some sense a special case.

**The Role of the Initialization.** Several studies have shown that initialization is crucial for optimizing neural networks, preventing vanishing or exploding gradients (Glorot & Bengio (2010)), speeding up convergence (He et al. (2015)), ensuring informative gradient flow in early layers (Sutskever et al. (2013)), and enabling learning challenging targets (Zhang et al. (2019); Hanin & Rolnick (2018)). While these works focus on improving learning through tailored initializations, our paper addresses the more fundamental question of what can gradient descent learn with standard initializations. Thus, our work aligns more closely with (Abbe & Boix-Adsera (2022); Abbe & Sandon (2020)), which characterize functions learnable by gradient descent on shallow networks, but without exploring initialization. Another work (Edelman et al. (2023)) shows that sparse initialization aids in learning sparse parities. However, the main challenge in their case is identifying the support of the sparse parity. In contrast, when learning the full parity, sparsifying the Rademacher initialization does not aid in learning the full parity (see Figure 3, Section 6).

**Complexity Measures.** Previous works have studied the sample and time complexity of learning with SGD on neural networks, proposing various measures, such as: the noise sensitivity (O'Donnell (2014); Zhang et al. (2021); Abbe et al. (2022b); Hahn & Rofin (2024)), which applies mostly to settings with i.i.d. inputs and is related to the degree of the functions, is known to be loose for strong learning (Abbe et al. (2022a; 2023)); the *globality degree* (Abbe et al. (2024a)), which generalizes the degree and sensitivity notions to non-i.i.d. settings but remains focused on weak rather than strong learning; the statistical query (SQ) dimension (Kearns (1998); Feldman (2016)) and the cross-predictability (Abbe & Sandon (2020)), which are usually defined for a class of targets/distributions rather than a single distribution (in particular the full parity is efficiently SQ learnable since there is a single function); the neural tangent kernel (NTK) alignment (Jacot et al. (2018); Cortes et al. (2012)) that are limited to the NTK framework; the information exponent (Arous et al. (2021); Bruna et al. (2023)), generative exponent (Damian et al. (2024); Dandi et al. (2024)), leap (Abbe et al. (2023)) and Approximate Message Passing (AMP) complexity (Troiani et al. (2024)), which measure when fully connected neural networks can strongly learn target functions on i.i.d. or isotropic input distributions and sparse or single/multi-index functions. In particular, few works studied measures based on the alignment between the networks initialization and the target distribution, as in this paper. (Mok et al. (2022); Ortiz-Jiménez et al. (2021)) studied the label-gradient-alignment (LGA), defined as the norm of the target function in the RKHS induced by the NTK (Jacot et al. (2018)) at initialization, showing its empirical relevance for predicting network performance. In contrast, we focus on a theoretical analysis, with our measure of initial gradient alignment being loss-dependent. Abbe et al. (2022c) defined the initial alignment (INAL) as the maximum average correlation of any neuron with the target, providing a lower bound for functions with small INAL, though their result relies on input embedding and orbit hardness, which does not apply to almost-full parities.

## 3 SETTING AND INFORMAL CONTRIBUTIONS

We consider learning with a neural network of $P$ parameters, $\mathrm{NN}(x; \theta)$, $\theta \in \mathbb{R}^P$, initialized as $\theta^0 \sim \mathcal{D}^0$, for some distribution $\mathcal{D}^0$, and trained using noisy stochastic gradient descent (noisy-SGD, see Def. 3). We assume that the network has access to data samples $(x, f(x))$, where $x \sim \mathcal{D}$, for $\mathcal{D}$ being a distribution in $\mathbb{R}^d$ and $f : \mathbb{R}^d \to \{\pm 1\}$ is an unknown target function. We focus on learning parity functions on uniform inputs ($\mathcal{D} = \mathrm{Unif}\{\pm 1\}^d$). A parity function over a subset $S$ of the input coordinates $[d] := \{1, 2, \ldots, d\}$ is a function $\chi_S : \{\pm 1\}^d \to \{\pm 1\}$, defined as $\chi_S(x) := \prod_{i \in S} x_i$, where $S \subseteq [d]$. We will focus on the case where $S = [d]$ (full parity) or $|S| = d - O_d(1)$ (almost full parity). Let us define our notion of perturbed initialization.

**Definition 1** (Perturbed Initialization). *Consider a neural network with parameters $\theta \in \mathbb{R}^P$ and two independent random vectors $A, H_\sigma \in \mathbb{R}^P$ with independent coordinates where $A$ is arbitrary and $H_\sigma$ has independent entries $(H_\sigma)_p \sim \mathcal{N}(0, \sigma^2 \cdot \mathbb{I}_P)$. We say that a neural network $\mathrm{NN}(x; \theta)$*

has a $(A, \sigma)$-perturbed initialization *with noise level $\sigma$ if its parameters are initialized to $\theta_p^0 = A_p + \sqrt{\operatorname{Var} A_p}(H_\sigma)_p$.*

We will mostly consider the case where $A \sim \operatorname{Unif}\{\pm 1\}^P$ (Rademacher initialization). In this scenario, we refer to the initialization as *$\sigma$-perturbed* Rademacher.

**Theorem 1** (Informal, Positive Almost-Full Parities). *Let $f(x) = \chi_S(x)$, with $S \subseteq [d], |S| = d - O_d(1)$. A two-layer $\operatorname{ReLU}$ network with some $\operatorname{poly}(d)$ hidden units and $\sigma$-perturbed Rademacher initialization with $\sigma = O(d^{-1})$, trained by GD or SGD with any batch-size with the correlation[4] or the hinge loss, will learn $f$ to perfect accuracy in $\operatorname{poly}(d)$ steps.*

For our negative result, we introduce the following notion of Gradient Alignment.

**Definition 2** (Gradient Alignment). *For a neural network $\operatorname{NN}(x; \theta)$, an input distribution $\mathcal{D}$, a target function $f : \mathbb{R}^d \to \mathbb{R}$, and a loss function $L : \mathbb{R} \times \mathbb{R} \to \mathbb{R}$, we denote the population gradient as*

$$\Gamma_f(\theta) := \mathbb{E}_x \left[ \nabla_\theta L(\operatorname{NN}(x; \theta), f(x)) \right] . \tag{1}$$

*If $\theta$ is a random initialization then we define the* gradient alignment *of $\theta$ as*

$$\operatorname{GAL}_f(\theta) := \mathbb{E}_\theta \|\Gamma_f(\theta) - \Gamma_r(\theta)\|_2^2 , \tag{2}$$

*where $\Gamma_r(\theta) := \mathbb{E}_{x,y}[\nabla_\theta L(\operatorname{NN}(x; \theta), y)]$ for $y \sim \operatorname{Rad}(1/2)$ and independent of $x$. That is, $\Gamma_r(\theta)$ is the gradient of a random classification task.*

We remark that for the squared and the correlation loss, the Gradient Alignment at initialization corresponds to the Label-Gradient-Alignment (LGA) of Ortiz-Jiménez et al. (2021); Mok et al. (2022), thus our GAL generalizes LGA to other losses.

We first prove that, under some conditions, if the Gradient Alignment at initialization is small, the network does not learn. We remark that this result holds for general input distributions (beyond Boolean and uniform) and for all networks with a linear output layer (see Section 5.1 for details).

**Theorem 2** (Informal, Negative General). *Let $f : \mathbb{R}^d \to \{\pm 1\}$ be a target function, and let $\operatorname{NN}(x; \theta)$ be a neural network with a linear output layer, trained by noisy-GD with noise level $\tau$ and the correlation loss. Assume either: 1) Gaussian initialization of the weights and homogeneous activation, or 2) $(A, \sigma)$-perturbed initialization, polynomially bounded gradients, and $\tau$ small enough (see details in Corollary 3). If $\operatorname{GAL}_f(\theta^0) < \exp(-\Omega(d))$, then after $\operatorname{poly}(d)$ training steps, the network will achieve an accuracy of at most $\frac{1}{2} + O(\exp(-\Omega(d)))$.*

We then apply this result to the case of almost-full parities on uniform inputs.

**Theorem 3** (Informal, Negative Almost-Full Parities). *Let $f(x) = \chi_S(x)$, for $S \subseteq [d]$ such that $|S| \geq d/2$. Noisy-GD with correlation loss and any noise level $\tau = \Omega(1/\operatorname{poly}(d))$ on any two-layer fully connected $\operatorname{ReLU}$ network of $\operatorname{poly}(d)$ size, initialized with Gaussian initialization will not achieve accuracy better than random guessing in $\operatorname{poly}(d)$ training steps.*

We expect Theorem 3 to hold also in case of $\sigma$-perturbed Rademacher initialization for $\sigma > \sigma^*$ for some fixed $\sigma^* > 0$. To that end in Section 5.2.2 we prove the gradient alignment bound for the hidden layer weights in the perturbed case. Together with a similar bound for the output layer weights (which we omit from this version of the paper) that would give the statement of Theorem 3 also for the $\sigma$-perturbed initialization, with $\sigma > \sigma^*$.

Full versions of Theorems 1 and 3 presented in the following sections provide the following rigorous separation between Rademacher and Gaussian initializations: Noisy-GD for correlation loss, when applied to a two-layer fully connected $\operatorname{ReLU}$ network with some $\operatorname{poly}(d)$ hidden neurons, can learn the full parity function in $\operatorname{poly}(d)$ steps if the network is initialized with Rademacher weights. However, using Gaussian initialization while leaving all other aspects of the algorithm unchanged requires exponential time to learn. Furthermore, the negative result is robust to details like changing hyperparameters, and as discussed above, both positive and negative results are also valid for some ranges of $\sigma$-perturbed Rademacher initializations.

---

[4]The correlation loss is defined as $L_{\operatorname{corr}}(y, \hat{y}) = -y\hat{y}$.

# 4 POSITIVE RESULT FOR RADEMACHER INITIALIZATION

In both positive and negative results we will be working with the noisy SGD and GD algorithm specified below:

**Definition 3** (Noisy-(S)GD). *Consider a neural network* $\mathrm{NN}(.;\theta)$*, with initialization of the weights* $\theta^0$*. Let* $f : \mathcal{X} \to \mathbb{R}$ *be a target function defined on an input space* $\mathcal{X}$*. Assume we are given fresh samples* $x \sim \mathcal{D}$*, for some input distribution* $\mathcal{D}$ *defined on* $\mathcal{X}$*. Given a weakly differentiable loss function L, the updates of the noisy-SGD algorithm with learning rate* $\gamma$ *are defined by*

$$\theta^{t+1} = \theta^t - \gamma \left( \frac{1}{B} \sum_{s=1}^{B} \nabla_{\theta^t} L(\mathrm{NN}(x^s; \theta^t), f(x^s)) + Z^t \right), \tag{3}$$

*where for all* $t \in \{0, \dots, T-1\}$*,* $Z^t$ *are i.i.d.* $\mathcal{N}(0, \tau^2)$*, for some noise level* $\tau$*, and they are independent from other variables, and B is the batch size. If the average over the batch size* $\frac{1}{B} \sum_{s=1}^{B} \nabla_{\theta^t} L(\mathrm{NN}(x^s; \theta^t), f(x^s))$ *is replaced by the population mean* $\mathbb{E}_{x \sim \mathcal{D}} [\nabla_{\theta^t} L(\mathrm{NN}(x; \theta^t), f(x))]$*, we refer to the algorithm as (full batch)* noisy-GD.

In this section we consider two layer neural networks with Rademacher initialization for the hidden layer weights. Our results imply that with large enough poly$(d)$ number of hidden neurons, the hidden layer embedding induced by the Rademacher distribution makes the almost-full parities for $k = d - O_d(1)$ linearly separable. Then:

1. When trained with the correlation loss on the uniform input distribution, the network achieves *perfect accuracy in one step of full GD* or in poly$(d)$ steps of SGD.

2. When trained with the hinge loss on *any input distribution*, the neural network achieves classification error $\epsilon$ in poly$(d)/\epsilon$ steps of SGD. (For simplicity we restrict this result to full parity.)

As mentioned, our positive result for the full parity holds also for a perturbed Rademacher initialization with deviation up to $C/d$ for some constant $C > 0$. We demonstrate this for hinge loss, see Section 4.2.

## 4.1 GD AND SGD WITH CORRELATION LOSS

We consider a fully connected network $N(x) = \sum_{i=1}^{n} v_i \sigma(w_i.x + b_i)$, where $\sigma$ is an arbitrary activation function. In the corollaries we will take $\sigma$ to be either ReLU or its clipped version. The network is trained with correlation loss $L(y, \hat{y}) = -y\hat{y}$ where only the output layer weights $v$ are trained. This is in contrast to the hinge loss result in Section 4.2 where we allow training of both layers. The gradient of output weights on input $x \in \{\pm 1\}^d$ is given by $\nabla_v L = -f_a(x)\sigma(Wx + b)$, where $W$ is an $n \times d$ matrix with rows $w_1, \dots, w_n$ and $f_a(x) = \prod_{i=1}^{d-a} x_i$ is the almost full parity function. During training, the inputs are sampled from the uniform distribution on $\{\pm 1\}^d$.

The hidden layer weights $w_i$ are initialized as i.i.d. Rademacher and the output weights as $v_i = 0$. The biases are i.i.d. according to some distribution $b_i \sim \mathcal{B}$. Our result depends on the following quantity:

$$\Delta_{d,b,\sigma}^{(a)} := \mathbb{E}_{x \sim \{\pm 1\}^d} \left[ (-1)^{(d-a-\sum_{j=1}^{d-a} x_j)/2} \sigma \left( \sum_{j=1}^{d} x_j + b \right) \right]. \tag{4}$$

In the following, let us assume that $\mathbb{E}_{b \sim \mathcal{B}} \left( \Delta_{d,b,\sigma}^{(a)} \right)^2 = \Delta^2$ and $|\sigma(w \cdot x + b)| \le R$, where both $\Delta^2$ and $R$ can vary with $d$. Furthermore, we assume that there exists a constant $C$ not depending on $d$ such that for every $b$ in the support of $\mathcal{B}$ it holds $|\Delta_{d,b,\sigma}| \le C\Delta$. (The last assumption is satisfied for any distribution $\mathcal{B}$ with a support of constant size. The distributions we consider in the corollaries have this property.)

**Theorem 4.** *Consider a network as above trained for one step with the GD algorithm. If* $n \ge \Omega(d \frac{R^2}{\Delta^2})$*, then, except with probability at most* $2 \exp(-d)$ *over the choice of initialization, we have* $\mathrm{sign}(N^1(x)) = f_a(x)$ *for every* $x \in \{\pm 1\}^d$*, where* $N^t(x)$ *denotes the output of the network at time t. This conclusion holds also in the presence of GD noise of magnitude* $\tau$ *up to* $O(\Delta^2/R)$*.*

**Theorem 5.** *Consider the above network trained with SGD of any batch size. Let $n \geq \Omega(d\frac{R^2}{\Delta^2})$. Then, except with probability $3\exp(-d)$, after $T \geq \Omega(\frac{R^4}{\Delta^4}(d + \log n))$ steps, the network predicts correctly $\mathrm{sign}(N^T(x)) = f_a(x)$ for every $x \in \{\pm 1\}^d$ in the presence of GD noise of magnitude $\tau$ up to $O(\frac{\sqrt{T}\Delta^2}{R})$.*

We present an application of Theorem 4 to a specific setting. By estimating $\Delta$, we prove a corollary for the full parity function for ReLU activation and its bounded variant, i.e., clipped ReLU. For clipped ReLU, order of $d^2$ neurons are sufficient for learning with high probability. We also provide a result for the almost full $(d-a)$-parities for ReLU activation and any $a = O(1)$:

**Corollary 1.** *In case of the full parity $a = 0$ and $\sigma = \mathrm{ReLU}$, let $b_i = 0$ if $d$ is even or $b_i = -1$ if $d$ is odd. Then, we have $\Delta^2 = \Theta(1/d)$ and $R = d+1$. Hence, $\Omega(d^4)$ hidden neurons are sufficient for strong learning in one step of GD. In the case of clipped ReLU $\sigma(x) = \max(0, \min(x, 5))$ it holds $\Delta^2 = \Theta(1/d)$ and $R = 5$, hence $\Omega(d^2)$ hidden neurons are sufficient.*

**Corollary 2.** *Let $a \in \mathbb{N}$. Take $b \sim \mathcal{B}$ such that $b_i = a+2$ with probability 1/2 and $b_i = a+2+0.1$ with probability 1/2. Then, for $\sigma = \mathrm{ReLU}$ it holds $\Delta^2 \geq \Omega(d^{-1-2\lceil a/2 \rceil})$. Accordingly, $n \geq \Omega(d^{4+2\lceil a/2 \rceil})$ hidden neurons are sufficient for strong learning in one GD step.*

In the corollaries above, we have chosen convenient bias values for simplicity, but the precise values are not crucial except for "unlucky" choices where $\Delta$ can become too small. In particular learning should hold for random biases for most reasonable distributions. For the clipped ReLU activation, we expect (but do not prove) that the bound on the number of neurons in Corollary 2 could be improved to $n \geq \Omega(d^{2+2\lceil a/2 \rceil})$ using a similar modification as in Corollary 1.

## 4.2 SGD ANALYSIS FOR HINGE LOSS

One of the implications of Theorem 4 is that under Rademacher initialization, with high probability the hidden layer embeddings of the parity function are linearly separable. We use known techniques (in particular, we borrow parts of the analysis from Nachum & Yehudayoff (2020)) to show that this implies learning for SGD under the hinge loss. For simplicity in this section we restrict ourselves to the ReLU activation and full parity. We refer to Appendix A.5 for details.

# 5 NEGATIVE RESULTS

## 5.1 NEGATIVE RESULTS FOR GENERAL TARGETS

In this section we prove a negative result that holds for all neural networks with a linear output layer:

**Definition 4** (Linear Output Layer). *We say that a neural network $\mathrm{NN}(x; \theta)$ has linear output layer if its output can be written as $\mathrm{NN}(x; \theta) = \sum_{i=1}^{n} v_i \mathrm{NN}_i(x; \psi)$, where $\theta = (v, \psi)$ are the trainable weights of the network, and $n$ denotes the number of neurons in the last hidden layer.*

In the context of binary classification, the network's $\pm 1$ label prediction is given by $\mathrm{sign}(\mathrm{NN}(x; \theta))$. Let us state our main negative result.

**Theorem 6** (Negative Result for General Targets). *Let $\mathrm{NN}(x; \theta)$ be a network with a linear output layer. Let the weights $\theta^0$ be initialized according to an $(A, \sigma)$-perturbed initialization (Def. 1), for $A \in \mathbb{R}^P$ with independent coordinates with distributions symmetric around 0. Assume the network is given samples $(x, f(x))$ where $x \sim \mathcal{D}$, for $\mathcal{D}$ being a distribution on $\mathbb{R}^d$. Let $\mathrm{NN}(x; \theta^T)$ be the output of the noisy-GD algorithm with noise level $\tau$ and learning rate $\gamma$ after $T$ steps of training with the correlation loss. Assume that there exists some bound $\varepsilon > 0$ such that for every $0 \leq \lambda^2 \leq T\gamma^2\tau^2$ we have*

$$\mathrm{GAL}_f(\theta^0 + \lambda H) \leq \varepsilon, \tag{5}$$

*where $H \sim \mathcal{N}(0, \mathbb{I}_P)$. Then, $\mathbb{P}\Big[\mathrm{sign}(\mathrm{NN}(x; \theta^T)) = f(x)\Big] \leq \frac{1}{2} + \frac{T\sqrt{\varepsilon}}{2\tau}$.*

In words, this theorem states that if equation 5 holds for $\varepsilon$ which is small compared to the noise level $\tau$, then noisy gradient descent will require a large number of training steps to achieve performance better than random guessing. Therefore, even the weakest form of learning is impossible. We provide here a brief outline of the proof, and refer to Appendix B for the full proof.

**Proof Outline of Theorem 6.** Our proof is composed of three steps: 1) We define the 'junk-flow', i.e. the training dynamics of a network trained on random noise (Definition 6); 2) We show that if the GAL remains small along the junk-flow, then the noisy-GD dynamics stay close to the junk flow in total variation (TV) distance, meaning that the network does not learn (Lemma 5); 3) For correlation loss, we demonstrate that if equation 5 holds, then the GAL remains small along the junk flow. Notably, steps (1) and (2) apply to any architecture with a linear output layer, symmetric initialization, and any loss function. However, step (3) is currently limited to the correlation loss, as tracking junk-flow dynamics for other losses is more complex.

Let us make a few remarks.

**Remark 1.** *For simplicity, we present Theorem 6 in the context of full batch noisy-GD. However, we note that the proof can be extended to noisy-SGD with a sufficiently large batch size, by leveraging the concentration of the effective gradient around the population mean, similarly to e.g. (Abbe & Sandon (2020), Theorem 3).*

**Remark 2.** *We propose using $\mathrm{GAL}_f$ as a measure for hardness of learning. However, the condition in equation 5 requires verifying that $\mathrm{GAL}_f$ remains small for all Gaussian perturbations of the initialization, with variance within the specified range. In Corollaries 3 and 4, we demonstrate that, in some settings, the condition in equation 5 can be simplified and expressed uniquely in terms of $\mathrm{GAL}_f(\theta^0)$.*

**Remark 3.** *We emphasize that Theorem 6 applies to any binary classification task and network architecture with a linear output layer, unlike, for example, Abbe et al. (2022c), which is specific to Boolean functions and product measures. Importantly, our result is restricted to the correlation loss, as the proof relies on coupling the gradient descent dynamics with the 'junk flow', as mentioned in the proof outline. We empirically observe that also for hinge loss, the $\mathrm{GAL}_f$ remains small along the junk flow over time (see Figure 2 in Section 6).*

As a first corollary, we show that when the GD noise level $\tau$ is small compared to the variances in the initial $H_\sigma$, the distributions of $H_\sigma$ and $H_\sigma + \lambda H$ are similar. As a result, equation 5 can be expressed in terms of $\mathrm{GAL}_f(\theta^0)$.

**Corollary 3.** *Let $f : \mathbb{R}^d \to \{\pm 1\}$ be a target function under a given input distribution $\mathcal{D}$. Let $\mathrm{NN}(x; \theta)$ be network with linear output layer, with weights initialized according to an $(A, \sigma)$-perturbed initialization, for 0-symmetric independent $A \in \mathbb{R}^P$. Assume that $\|\mathbb{E}_x|\nabla \mathrm{NN}(x; \theta)|\|_2^2 \le R$ for all $\theta$.[5] Let $\mathrm{NN}(x; \theta^T)$ be the output of the noisy-GD algorithm with noise level $\tau$ and learning rate $\gamma$ such that $\tau^2 \le \frac{\sigma^2 \min_p \mathrm{Var} A_p}{PT\gamma^2}$, after $T$ steps with the correlation loss. Then,*

$$\mathbb{P}(\mathrm{NN}(x; \theta^T) = f(x)) \le \frac{1}{2} + \frac{T\sqrt{4R+1}}{2\tau} \cdot \mathrm{GAL}_f(\theta^0)^{1/18}. \tag{6}$$

The proof of Corollary 3 is deferred to Appendix B.3. While the above corollary applies to general perturbed initializations, it relies on the assumption that the GD noise level $\tau$ is sufficiently small. However, we also show that in the specific case of Gaussian initialization and assuming a homogeneous architecture, this assumption can be removed.

**Gaussian Initialization.** Let us restrict ourselves to the special case of Gaussian initialization, i.e. when $A = 0_P$. We assume that the activation $h$ satisfies the following homogeneity property.

**Definition 5** (*H*-Weakly Homogeneous.)**.** *Let $h : \mathbb{R} \to \mathbb{R}$ be an activation function. We say that $h$ is $H$-weakly homogeneous if for all $x \in \mathbb{R}$ and $C \ge 0$, $h(Cx) = C^H h(x)$.*

For example, $\mathrm{ReLU}(x) = \max\{0, x\}$ is 1-weakly homogeneous. $x^k$ is $k$-weakly homogeneous, for all $k \in \mathbb{N}$. We prove the following result.

**Corollary 4.** *Let $\mathrm{NN}(x; \theta)$ be a fully connected network of depth $L$, with $H$-weakly homogeneous activation and with weights initialized as $\theta_p^0 \sim \mathcal{N}(0, \sigma_{l_p}^2)$ where $l_p$ denotes the layer of parameter $\theta_p$, for $p \in [P]$. Let $f : \mathbb{R}^d \to \{\pm 1\}$ be a balanced target function. Let $\mathrm{NN}(x; \theta^T)$ be the output of the noisy-GD algorithm with noise-level $\tau$, after $T$ steps of training with the correlation loss. Then,*

$$\mathbb{P}(\mathrm{NN}(x; \theta^T) = f(x)) \le \frac{1}{2} + \frac{T}{2\tau} \prod_{l=1}^{L} \left(1 + \frac{T\gamma^2\tau^2}{\sigma_l^2}\right)^H \cdot \mathrm{GAL}_f(\theta^0)^{1/2}. \tag{7}$$

---

[5]This always holds if we assume gradient clipping.

## 5.2 Negative Results for High-Degree Parities

### 5.2.1 Small Alignment for Gaussian Initialization

In this section we state a rigorous lower bound for learning of large degree parities with pure Gaussian initialization. This is in the setting of two layer ReLU neural networks. Then we will discuss extending this result to a perturbed Rademacher initialization for a large enough constant perturbation.

**Theorem 7.** *Let $\theta = (w, b, v)$ for $w \in \mathbb{R}^{n \times d}, b \in \mathbb{R}^n, v \in \mathbb{R}^n$ and $\mathrm{NN}(x; \theta) = \sum_{i=1}^n v_i \mathrm{ReLU}(w_i \cdot x + b_i)$. Let $a = a(d) \leq d/2$ and $f_a(x) = \prod_{i=1}^{d-a} x_i$. Consider the i.i.d. initialization $w \sim \mathcal{N}\left(0, \frac{1}{d} \mathrm{Id}_{n \times d}\right), b \sim \mathcal{N}(0, \sigma^2 \mathrm{Id}_n)$ for any $\sigma^2 = O(1), v \sim \mathcal{N}\left(0, \frac{1}{n} \mathrm{Id}_n\right)$.*

*Then, for any number of hidden neurons $n = \exp(o(d))$, any number of time steps $T = \exp(o(d))$, any learning rate $0 \leq \gamma \leq \exp(o(d))$, any noise level $\exp(-o(d)) \leq \tau \leq \exp(o(d))$, after $T$ steps of the noisy GD algorithm with correlation loss,*

$$\Pr\left[\mathrm{sign}(\mathrm{NN}^T(x; \theta)) = f_a(x)\right] \leq \frac{1}{2} + \exp(-\Omega(d)) . \tag{8}$$

Theorem 7 follows from Theorem 6 and the following bound on the gradient alignment:

**Proposition 1.** *Let a neural network be as in Theorem 7. Then, for every $\sigma_0^2 > 0$, there exists $C, C' > 0$ such that for any network with $\sigma^2 \leq \sigma_0^2$ we have a gradient alignment bound*

$$\mathrm{GAL}_{f_a}(\theta) \leq PC' \exp(-Cd) , \tag{9}$$

*where $P := nd + 2n$ is the total number of parameters.*

### 5.2.2 Small Alignment for Perturbed Initialization

Consider the perturbed Rademacher initialization $\frac{1}{\sqrt{d}}(r + g)$ for $g \sim \mathcal{N}(0, \sigma^2)$ for some constant $\sigma > 0$. In order to prove a rigorous lower bound like in Theorem 7 for this initialization, we need to establish the alignment bound for $\mathrm{GAL}_f(r + g)$. Once this bound is proved, the remaining steps of the proof are similar as for Theorem 7.

**Theorem 8.** *There exists $\sigma_0 > 0$ such that for all $\sigma = \sigma(d)$ such that $\sigma_0 \leq \sigma \leq \exp(o(d))$ the following holds:*

*Let $\theta = (w, v)$ for $w \in \mathbb{R}^{n \times d}, v \in \mathbb{R}^n$ and $\mathrm{NN}(x; \theta) = \sum_{i=1}^n v_i \mathrm{ReLU}(w_i \cdot x)$. Let $f(x) = \prod_{i=1}^d x_i$. Consider the i.i.d. initialization $w = \frac{1}{\sqrt{d}}(r + g)$ where $r \sim \mathrm{Rad}(1/2), g \sim \mathcal{N}(0, \sigma^2)$ with all coordinates independent, $v \sim \mathcal{N}\left(0, \frac{1}{n} \mathrm{Id}_n\right)$.*

*Then, for any number of hidden neurons $n = \exp(o(d))$, any number of time steps $T = \exp(o(d))$, any learning rate $0 \leq \gamma \leq \exp(o(d))$, any noise level $\exp(-o(d)) \leq \tau \leq \exp(o(d))$, after $T$ steps of the noisy GD algorithm with correlation loss,*

$$\Pr\left[\mathrm{sign}(\mathrm{NN}^T(x; \theta)) = f(x)\right] \leq \frac{1}{2} + \exp(-\Omega(d)) . \tag{10}$$

**Proposition 2.** *There exists $\sigma_0, C, C' > 0$ such that the following holds: Let the setting be as in Theorem 8. For any network with perturbed initialization with $\sigma \geq \sigma_0$ we have a gradient alignment bound*

$$\mathrm{GAL}_f(\theta) \leq \sigma^2 PC' \exp(-Cd) , \tag{11}$$

*where $P := nd + n$ is the total number of parameters.*

The proofs for this section can be found in Appendices C and D.

## 6 Experiments

In this section, we show empirical results on the impact of the initialization in learning the full parity. As our model, we use a multi-layer perceptron (MLP) with 3 hidden layers of neurons sizes 512, 512 and 64 with ReLU activation, and we train it with SGD with batch size 64 on the hinge loss, training all layers simultaneously. Each experiment is repeated for 7 random seeds and we report the 95% confidence intervals. In Appendix E, we report further experiment details, as well as additional experiments.

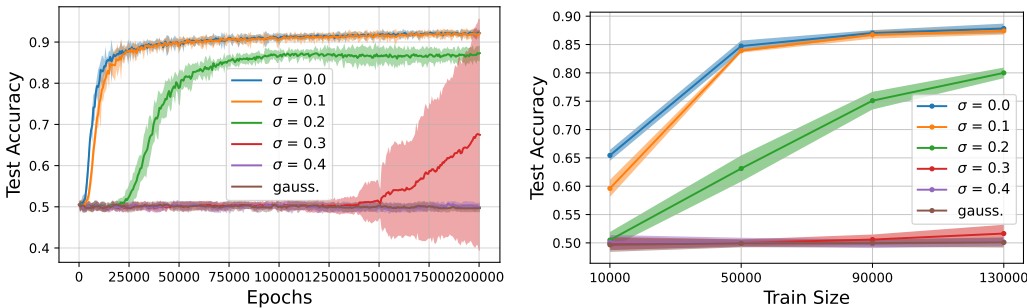

Figure 1: Learning the full parity with $\sigma$-perturbed initialization by SGD with the hinge loss on a 4-layer MLP, with $d = 50$, with online fresh samples (left) and with an offline fixed dataset (right).

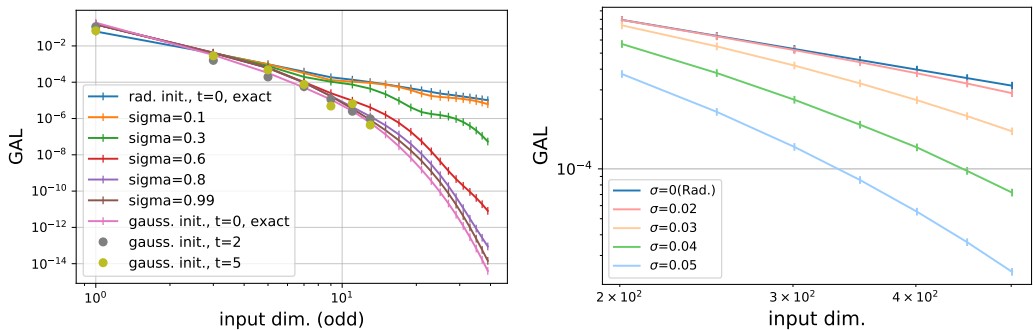

Figure 2: Computing numerically the alignment $\mathrm{GAL}_f$ with the hinge loss (left) and the correlation loss (right), for a one-neuron network.

**$\sigma$-Perturbed Initialization.** We first consider learning the full parity function with $\sigma$-perturbed initializations and investigate the effect of varying $\sigma$ (Figure 1). To make different initializations comparable, we normalize them such that the variance entering each neuron is 1 (see Appendix E for details). We observe that the test accuracy after training decreases as $\sigma$ increases. This pattern is seen in both the online setting (left plot), where fresh batches are sampled at each iteration, and the offline setting (right plot), where the network is trained on a fixed dataset until the training loss decreases to $10^{-2}$, and evaluated on a separate test set. For input dimension $d = 50$, as in Figure 1, we find that some learning occurs for $\sigma \in \{0.1, 0.2\}$. However, in the Appendix, we report experiments with larger input dimensions, where learning does not occur for these values of $\sigma$ (Figure 5).

**Gradient Alignment.** In Figure 2 we compute the gradient alignment for a one-neuron $\mathrm{ReLU}$ network under different initializations and losses, which are not covered by our theoretical results. The left plot shows the $\mathrm{GAL}_f$ at initialization for the hinge loss with Rademacher, $\sigma$-perturbed Rademacher and Gaussian initializations, across input dimensions up to $d = 40$. We observe that $\mathrm{GAL}_f$ seems to decrease at an inverse-polynomial rate for Rademacher, but super-polynomially fast for Gaussian and $\sigma$-perturbed initializations for large $\sigma$ (e.g. $\sigma \in \{0.8, 0.99\}$). The case of smaller $\sigma \in \{0.1, 0.3\}$ is less conclusive. We also estimate, with Montecarlo, the $\mathrm{GAL}_f$ after training the neuron for a few steps ($t = 2$ and $t = 5$) on random labels (dots). We observe that training on random labels does not increase the $\mathrm{GAL}_f$. A theoretical understanding of this observation would allow to extend our negative result to the hinge loss.

In the right plot, we estimate numerically the initial $\mathrm{GAL}_f$ for the correlation loss for a single threshold neuron. We consider $\sigma$-perturbed initializations with small $\sigma$, contrasting Theorem 8 and Proposition 2, which apply only for large $\sigma$. For small $\sigma$, $\mathrm{GAL}_f$ deviates from the Rademacher case, suggesting that it could be super-polynomially small for all constants $\sigma > 0$. Further investigation for small $\sigma$ is left for future work, and Appendix E shows that $\mathrm{GAL}_f$ remains super-polynomially small for larger values of $\sigma$, confirming our theory.

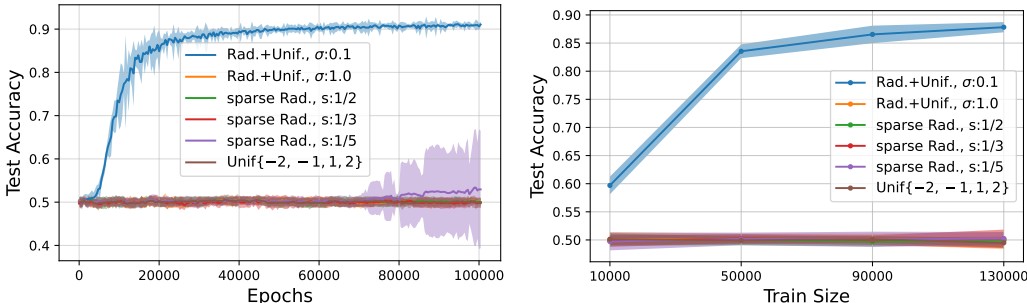

Figure 3: Learning the full parity with perturbations of the Rad. initialization by SGD with the hinge loss on a $4$-layer MLP, with $d = 50$, with online fresh samples (left) and with an offline dataset (right).

**Other Perturbed Initializations.** We next explore perturbations beyond mixtures of Gaussians. In Figure 3, we consider three types: 1) a mixture of two continuous uniform distributions with means $+1$ and $-1$, and standard deviations $\sigma \in \{0.1, 1.0\}$; 2) a sparsified Rademacher initialization, where a fraction $s \in \{1/2, 1/3, 1/5\}$ of the weights are set to $0$, and the rest follow a $\mathrm{Rad}(1/2)$ distribution; and 3) a symmetric discrete initialization, where the weights are randomly chosen from $\{-2, -1, 1, 2\}$. We find that the mixture of continuous uniforms behaves similarly to the mixture of Gaussians: for $\sigma = 0.1$ and input dimension $d = 50$, the network successfully learns, but learning is prevented at larger $\sigma$. Additionally, we observe that all other discrete initializations fail to enable learning, suggesting that the Rademacher initialization is a special case.

# 7 CONCLUSION

In this paper, we advance the understanding of whether high degree parities can be learned using noisy-GD on standard neural networks with i.i.d. initializations. Specifically, we show that while the full parity is easily learnable with Rademacher initialization, it becomes challenging when Gaussian perturbations with large variance are introduced. This constitutes a separation between SQ algorithms and gradient descent on neural networks: the full parity is an example of a function that is trivially learnable in the statistical query (SQ) framework but difficult for noisy-GD on neural networks with most typical initializations, with the Rademacher being a special case. It raises interesting questions about a threshold where learning behavior changes based on the perturbation level $\sigma$. result, e.g., to hinge loss and/or deeper architectures.

Additionally, we propose a novel, loss-dependent measure for assessing alignment between the initialization and the target distribution, and prove a negative result for the correlation loss that applies to general input distributions, beyond the specific case of full parity and Boolean inputs. We leave to future work investigating if this technique can be applied more generally, especially in other scenarios where the dynamics remain stuck near initialization for a significant time (hardness of weak-learning). For example, such behavior is plausible for targets presenting symmetries and requiring some level of 'logical reasoning' (e.g. arithmetic, graphs, syllogisms), for which parities are a simple model. Similarly, we leave open strengthening of the negative result, e.g., to hinge loss and/or deeper architectures.

# 8 ACKNOWLEDGMENTS

The authors thank Nati Srebro for a useful discussion and Ryan O'Donnell for a helpful communication regarding binomial coefficients. Part of this work was done while EC was visiting the African Institute for Mathematical Sciences (AIMS), Rwanda. JH and DKY were supported by the Alexander von Humboldt Foundation German research chair funding and the associated DAAD project No. 57610033. EC was partly supported by the NSF DMS-2031883 and the Bush Faculty Fellowship ONR-N00014-20-1-2826.

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

## A  PROOFS AND DETAILS FOR SECTION 4

A useful identity to be remembered for later is, for every $x, w \in \{\pm 1\}^d$:

$$\prod_{j=1}^{d} x_j w_j = (-1)^{(d - w \cdot x)/2} .$$ (12)

### A.1 PROOF OF THEOREM 4

First, consider the case without GD noise. One step of GD with learning rate $\gamma$ results in the following update:

$$v_i^{t+1} = v_i^t + \gamma \mathbb{E}_{x \sim \{\pm 1\}^d} \left( \prod_{j=1}^{d-a} x_j \right) \sigma(w_i \cdot x + b_i) \tag{13}$$

$$= v_i^t + \gamma \mathbb{E}_{x \sim \{\pm 1\}^d} \left( \prod_{j=1}^{d-a} w_{ij} \right) (-1)^{(d-a-\sum_{j=1}^{d-a} w_{ij}x_j)/2} \sigma(\sum_{j=1}^{d} w_{ij}x_j + b_i) \tag{14}$$

$$= v_i^t + \gamma \left( \prod_{j=1}^{d-a} w_{ij} \right) \Delta_{d,b_i,\sigma}^{(a)}, \tag{15}$$

keeping in mind from equation 4 that

$$\Delta_{d,b,\sigma}^{(a)} = \mathbb{E}_{x \sim \{\pm 1\}^d} \left[ (-1)^{(d-a-\sum_{j=1}^{d-a} x_j)/2} \sigma \left( \sum_{j=1}^{d} x_j + b \right) \right]. \tag{16}$$

Since $w_i \cdot x$ is distributed as a sum of i.i.d. Rademachers regardless of $w_i$, the value in equation 14 indeed can be replaced with the factor $\Delta_{d,b_i,\sigma}^{(a)}$ which does not depend on $w_i$.

Accordingly, after one step of GD for starting zero weights $v^0 = 0$, the output of the network is given by

$$N^1(x) = \sum_{i=1}^{n} \gamma \Delta_{d,b_i,\sigma}^{(a)} \prod_{j=1}^{d-a} w_{ij} \sigma(w_i \cdot x + b_i). \tag{17}$$

For fixed $x \in \{\pm 1\}^d$ and in expectation over $w$ and $b$, this is, using equation 12,

$$\mathbb{E}_{w,b} N^1(x) = \gamma \sum_{i=1}^{n} \mathbb{E}_{b_i} \left[ \Delta_{d,b_i,\sigma}^{(a)} \mathbb{E}_{w \sim \{\pm 1\}^d} \left[ \prod_{j=1}^{d-a} w_{ij} \sigma(w_i \cdot x + b_i) \right] \right] \tag{18}$$

$$= \gamma \left( \prod_{j=1}^{d-a} x_j \right) \sum_{i=1}^{n} \mathbb{E}_{b_i} \left[ \Delta_{d,b_i,\sigma}^{(a)} \mathbb{E}_{w \sim \{\pm 1\}^d} \left[ (-1)^{(d-a-\sum_{j=1}^{d-a} w_{ij}x_j)/2} \sigma(w_i \cdot x + b_i) \right] \right] \tag{19}$$

$$= \gamma \left( \prod_{j=1}^{d-a} x_j \right) \sum_{i=1}^{n} \mathbb{E}_{b_i} \left[ [\Delta_{d,b_i,\sigma}^{(a)}]^2 \right] = \gamma \left( \prod_{j=1}^{d-a} x_j \right) n\Delta^2. \tag{20}$$

Let us come back to the expression $f_a(x)N^1(x)$ for a fixed $x \in \{\pm 1\}^d$. Its value is a random variable depending on the hidden layer initialization $W$. By equation 17, it can be written as a sum of $n$ i.i.d. random variables, and each of them has absolute value at most $\gamma RC\Delta$. Furthermore, it follows from equation 20 that $\mathbb{E}_w f_a(x)N^1(x) = \gamma n\Delta^2$. Therefore, we can upper bound the prediction error probability by Hoeffding's inequality:

$$\Pr_{w,b}[f_a(x)N^1(x) \le 0] \le \Pr_{w,b}\left[ f_a(x)N^1(x) \le \frac{\gamma n\Delta^2}{2} \right] \le \exp\left( -\frac{n\Delta^2}{8R^2C^2} \right) \le \exp(-2d), \tag{21}$$

where the last inequality holds for $n \ge \Omega(d\frac{R^2}{\Delta^2})$. Therefore, by union bound, the network will make correct predictions $f_a(x)N^1(x) > 0$ for all $x \in \{\pm 1\}^d$ except with probability $\exp(-d)$.

In the presence of gradient noise, the weights are given as $\tilde{v}^1 = v^1 + \gamma\xi$, where $\xi \sim \mathcal{N}(0, \tau^2 \operatorname{Id})$. Then,

$$f_a(x)\tilde{N}^1(x) = f_a(x)N^1(x) + \gamma f_a(x) \sum_{i=1}^{n} \xi_i \sigma(w_i \cdot x + b_i) \ge f_a(x)N^1(x) - \gamma R \sum_{i=1}^{n} |\xi_i|. \tag{22}$$

Using equation 21, except with probability $\exp(-d)$, we will have $f_a(x)\tilde{N}^1(x) > 0$ for every $x$ as long as $\gamma R \sum_i |\xi_i| \leq \frac{\gamma n \Delta^2}{2}$, or equivalently $\sum_i |\xi_i| \leq \frac{n\Delta^2}{2R}$. Note that $\mathbb{E}|\xi_i| = \tau \sqrt{2/\pi}$, so by assumption $\tau \leq O(\frac{\Delta^2}{R})$ we have $\mathbb{E}\sum_i |\xi_i| \leq \frac{n\Delta^2}{4R}$.

Furthermore, as $\xi_i$ has Gaussian distribution, its absolute value $|\xi_i|$ is sub-Gaussian (see, e.g., Proposition 2.5.2 in Vershynin (2018)). Therefore, by sub-Gaussian concentration, we can estimate

$$\Pr\left[\sum_{i=1}^n |\xi_i| \geq \frac{n\Delta^2}{2R}\right] \leq \Pr\left[\sum_{i=1}^n |\xi_i| - \mathbb{E}|\xi_i| \geq \frac{n\Delta^2}{4R}\right] \leq \exp\left(-\Omega\left(\frac{n\Delta^4}{R^2\tau^2}\right)\right) \leq \exp(-d) , \tag{23}$$

where the last inequality holds as $n \geq \Omega(d\frac{R^2}{\Delta^2})$ and $\tau^2 = O(\frac{\Delta^4}{R^2})$. All in all, the noisy network classifies all inputs correctly except with probability at most $2\exp(-d)$. $\qquad\square$

## A.2 PROOF OF THEOREM 5

In the general case of noisy SGD, let $v = v^1 \in \mathbb{R}^n$ be the update given by GD, that is $v_i = \mathbb{E}_x f_a(x)\sigma(w_i \cdot x + b_i)$. The SGD update can be written as

$$\hat{v}_i^{t+1} = \hat{v}_i^t + \gamma e_i^t + \gamma \xi_i^t , \tag{24}$$

where: (a) $e_i^t$ for $1 \leq i \leq n$ is a random variable with expectation $\mathbb{E}e_i^t = v_i$ and bounded by $|e_i^t| \leq R$; (b) $\xi^t \sim \mathcal{N}(0, \tau^2 \,\mathrm{Id})$; and where those random variables are independent across time.

From equation 21, if $n > \Omega(d\frac{R^2}{\Delta^2})$, except with probability $\exp(-d)$ over the choice of hidden layer weights $w$ and biases $b$, for every $x \in \{\pm 1\}^d$ it holds

$$f_a(x)\sum_{i=1}^n v_i\sigma(w_i \cdot x + b_i) > \frac{\gamma n\Delta^2}{2} . \tag{25}$$

Let $x \in \{\pm 1\}^d$. We estimate

$$f_a(x)\hat{N}^T(x) = f_a(x)\gamma\sum_{i=1}^n \left(Tv_i + \sum_{t=1}^T e_i^t - v_i + \xi_i^t\right)\sigma(w_i \cdot x + b_i) \tag{26}$$

$$> \frac{\gamma T n\Delta^2}{2} - \gamma R\left(\sum_{i=1}^n \left|\sum_{t=1}^T e_i^t - v_i\right| + \sum_{i=1}^n \left|\sum_{t=1}^T \xi_i^t\right|\right) . \tag{27}$$

Accordingly, if $\sum_i |\sum_t \xi_i^t| \leq \frac{Tn\Delta^2}{8R}$ and $|\sum_t e_i^t - v_i| \leq \frac{Tn\Delta^2}{8R}$ for every $1 \leq i \leq n$, then $f_a(x)\hat{N}^T(x) > 0$. We now show that each of those two events fails to occur with only exponentially small probability.

First, recall that we have almost surely $|e_i^t| \leq R$. By Hoeffding's inequality,

$$\Pr\left[\sum_{t=1}^T |e_i^t - v_i| \geq \frac{Tn\Delta^2}{8R}\right] \leq 2\exp\left(-\frac{T\Delta^4}{2^7 \cdot R^4}\right) \leq \exp(-d)/n ,$$

as soon as $T \geq \Omega(\frac{R^4}{\Delta^4}(d+\log n))$. By union bound, $|\sum_t e_i^t - v_i| \leq \frac{Tn\Delta^2}{8R}$ holds for every $1 \leq i \leq n$, except with probability $\exp(-d)$.

As for the additional Gaussian noise, observe that for $\tau = O(\frac{\sqrt{T}\Delta^2}{R})$ we have

$$\mathbb{E}\left[\sum_{i=1}^n \left|\sum_{t=1}^T \xi_i^t\right|\right] = n\tau\sqrt{T}\sqrt{2/\pi} \leq \frac{Tn\Delta^2}{16R} . \tag{28}$$

Similarly as in the GD case, $\left|\sum_t \frac{\xi_i^t}{\sqrt{T}\tau}\right|$ is a sub-Gaussian random variable. Therefore, we have concentration

$$\Pr\left[\sum_{i=1}^n \left|\sum_{t=1}^T \xi_i^t\right| \geq \frac{Tn\Delta^2}{8R}\right] \leq \Pr\left[\sum_{i=1}^n \left|\sum_{t=1}^T \xi_i^t\right| - n\tau\sqrt{T}\sqrt{2/\pi} \geq \frac{Tn\Delta^2}{16R}\right] \tag{29}$$

$$\leq \exp\left(-\Omega\left(\frac{Tn\Delta^4}{\tau^2 R^2}\right)\right) \tag{30}$$

$$\leq \exp(-d) , \tag{31}$$

since $\tau = O(\sqrt{T}\Delta^2/R)$ and $n = \Omega(d\frac{R^2}{\Delta^2})$. $\qquad\square$

## A.3 PROOFS OF COROLLARY 1 AND COROLLARY 2

Let $\Delta_{d,b,\sigma} := \Delta_{d,b,\sigma}^{(0)}$. We need to find asymptotic bounds on $|\Delta_{d,b,\sigma}|$ for $\sigma = \text{ReLU}$ and $\sigma = $ clipped-ReLU (let's denote it as CReLU) and $|\Delta_{d,b,\text{ReLU}}^{(a)}|$ for $a > 0$. We therefore turn to developing formulas for $\Delta_{d,b,\text{CReLU}}$ and $\Delta_{d,b,\text{ReLU}}^{(a)}$. Let's first consider the following combinatorial claim:

**Claim 1.** *For any integer $d, c > 1$ and $c'$ such that $c \leq c'$:*

1. $\sum_{k=c}^d (-1)^k \binom{d}{k} = (-1)^c \binom{d-1}{c-1}$

2. $\sum_{k=c}^d (-1)^k k \binom{d}{k} = (-1)^c d \binom{d-2}{c-2}$

3. $\sum_{k=c}^{c'} (-1)^k \binom{d}{k} = (-1)^c \binom{d-1}{c-1} + (-1)^{c'} \binom{d-1}{c'}$

4. $\sum_{k=c}^{c'} (-1)^k k \binom{d}{k} = (-1)^c d \binom{d-2}{c-2} + (-1)^{c'} d \binom{d-2}{c'-1}$ .

*Proof.* Here and below, we follow the convention $\binom{d}{k} = 0$ for $k < 0$ or $k > d$.

1. This follows by observing that it is a telescopic sum, with the term $(-1)^d \binom{d-1}{d} = 0$ by convention:

$$\sum_{k=c}^d (-1)^k \binom{d}{k} = \sum_{k=c}^d (-1)^k \left(\binom{d-1}{k} + \binom{d-1}{k-1}\right) = (-1)^c \binom{d-1}{c-1} . \tag{32}$$

2. Here we use the above and the fact that $k\binom{d}{k} = d\binom{d-1}{k-1}$,

$$\sum_{k=c}^d (-1)^k k \binom{d}{k} = d \sum_{k=c}^d (-1)^k \binom{d-1}{k-1} = (-1)^c d \binom{d-2}{c-2} . \tag{33}$$

3. This follows from equation 32, indeed

$$\sum_{k=c}^{c'} (-1)^k \binom{d}{k} = \sum_{k=c}^d (-1)^k \binom{d}{k} - \sum_{k=c'+1}^d (-1)^k \binom{d}{k} \tag{34}$$

$$= (-1)^c \binom{d-1}{c-1} + (-1)^{c'} \binom{d-1}{c'} \tag{35}$$

4. Similarly this follows from equation 33,

$$\sum_{k=c}^{c'} (-1)^k k \binom{d}{k} = \sum_{k=c}^d (-1)^k k \binom{d}{k} - \sum_{k=c'+1}^d (-1)^k k \binom{d}{k} \tag{36}$$

$$= (-1)^c d \binom{d-2}{c-2} + (-1)^{c'} d \binom{d-2}{c'-1} . \qquad\square$$

**Lemma 1.** *Let $d > 1$, $b \in \mathbb{R}$, $c = c(d,b) := \lceil (d-b)/2 \rceil$ and $c' = \lfloor (d-b+5)/2 \rfloor$. Then,*

$$\Delta_{d,b,\mathrm{ReLU}} = \frac{(-1)^{d+c}}{2^d} \left[ (d+b)\binom{d-2}{c-2} - (d-b)\binom{d-2}{c-1} \right] , \tag{37}$$

$$\Delta_{d,b,\mathrm{CReLU}} = \frac{(-1)^{d+c}}{2^d} \left[ (d+b)\binom{d-2}{c-2} - (d-b)\binom{d-2}{c-1} \right] \tag{38}$$

$$+ \frac{(-1)^{d+c'}}{2^d} \left[ (d+b-5)\binom{d-2}{c'-1} - (d-b+5)\binom{d-2}{c'} \right] . \tag{39}$$

*Proof.* Recall that $x$ in the definition of $\Delta_{d,b,\sigma}$ is distributed as i.i.d. uniform Rademachers. Therefore, we can write $x_j = -1 + 2z_j$, where $z$ are i.i.d uniform Bernoullis. Using Claim 1 and the definition of $\Delta_{d,b,\sigma}$:

$$\Delta_{d,b,\mathrm{ReLU}} = (-1)^d \mathbb{E}_z \left[ (-1)^{\sum_j z_j} \mathrm{ReLU}\left( b - d + 2\sum_{j=1}^d z_j \right) \right] \tag{40}$$

$$= (-1)^d 2^{-d} \sum_{k=c}^d (-1)^k \binom{d}{k} (b - d + 2k) , \tag{41}$$

$$= (-1)^{d+c} 2^{-d} \left( (b-d)\binom{d-1}{c-1} + 2d\binom{d-2}{c-2} \right) \tag{42}$$

$$= (-1)^{d+c} 2^{-d} \left( d\binom{d-2}{c-2} - d\binom{d-2}{c-1} + b\binom{d-1}{c-1} \right) \tag{43}$$

$$= \frac{(-1)^{d+c}}{2^d} \left[ (d+b)\binom{d-2}{c-2} - (d-b)\binom{d-2}{c-1} \right] . \tag{44}$$

Similarly we have,

$$\Delta_{d,b,\mathrm{CReLU}} = (-1)^d \mathbb{E}_z \left[ (-1)^{\sum_j z_j} \mathrm{CReLU}\left( b - d + 2\sum_{j=1}^d z_j \right) \right] \tag{45}$$

$$= (-1)^d 2^{-d} \sum_{k=c}^d (-1)^k \binom{d}{k} \min(5, b - d + 2k) , \tag{46}$$

$$= (-1)^d 2^{-d} \sum_{k=c}^{c'} (-1)^k \binom{d}{k} (b - d + 2k) + 5(-1)^d 2^{-d} \sum_{k=c'+1}^d (-1)^k \binom{d}{k} , \tag{47}$$

$$= (-1)^{d+c} 2^{-d} \left[ (b-d)\binom{d-1}{c-1} + 2d\binom{d-2}{c-2} \right] \tag{48}$$

$$+ (-1)^{d+c'} 2^{-d} \left[ (b-d-5)\binom{d-1}{c'} + 2d\binom{d-2}{c'-1} \right] , \tag{49}$$

$$= (-1)^{d+c} 2^{-d} \left[ d\binom{d-2}{c-2} - d\binom{d-2}{c-1} + b\binom{d-1}{c-1} \right] \tag{50}$$

$$+ (-1)^{d+c'} 2^{-d} \left[ d\binom{d-2}{c'-1} - d\binom{d-2}{c'} + (b-5)\binom{d-1}{c'} \right] , \tag{51}$$

$$= \frac{(-1)^{d+c}}{2^d} \left[ (d+b)\binom{d-2}{c-2} - (d-b)\binom{d-2}{c-1} \right] \tag{52}$$

$$+ \frac{(-1)^{d+c'}}{2^d} \left[ (d+b-5)\binom{d-2}{c'-1} - (d-b+5)\binom{d-2}{c'} \right] . \qquad \square$$

### A.3.1 PROOF OF COROLLARY 1

Recall the value $c = \lceil (d-b)/2 \rceil$ from Lemma 1. For ReLU activation, in the case of even $d$ (recall that the bias is $b_i = 0$), we have $c = d/2$ and:

$$|\Delta_{d,0,\mathrm{ReLU}}| = \frac{d}{2^d}\left|\binom{d-2}{d/2-2} - \binom{d-2}{d/2-1}\right| = \frac{4}{2^d}\binom{d-3}{d/2-1} \tag{53}$$

$$= \Theta\left(\frac{1}{\sqrt{d}}\right), \tag{54}$$

where in the last line we applied an estimate $\frac{2^d}{8}\frac{2}{3\sqrt{d}} \le \binom{d-3}{d/2-1} \le \frac{2^d}{8}\frac{2}{\sqrt{d}}$. In the case of odd $d$ (with bias $b_i = -1$) it holds $c = (d+1)/2$, and we proceed similarly

$$|\Delta_{d,-1,\mathrm{ReLU}}| = \frac{1}{2^d}\binom{d-1}{(d-1)/2} = \Theta\left(\frac{1}{\sqrt{d}}\right). \tag{55}$$

For the CReLU activation, in the even case, $c = d/2$ and $c' = d/2 + 2$

$$|\Delta_{d,0,\mathrm{CReLU}}| = \frac{1}{2^d}\left|d\binom{d-2}{\frac{d}{2}-2} - d\binom{d-2}{\frac{d}{2}-1} + (d-5)\binom{d-2}{d/2+1} - (d+5)\binom{d-2}{d/2+2}\right|, \tag{56}$$

$$= \frac{1}{2^d}\left|-4\binom{d-3}{d/2-1} + \frac{5}{d-1}\binom{d-1}{d/2+2}\right|, \tag{57}$$

$$= \Theta\left(\frac{1}{\sqrt{d}}\right). \tag{58}$$

The last equality holds because as $d$ grows $\binom{d-3}{d/2-1}$ dominates over $\frac{1}{d-1}\binom{d-1}{d/2+2}$. In the case $d$ odd we have $c = (d+1)/2$, $c' = (d+1)/2 + 2$ and we proceed similarly to get

$$|\Delta_{d,-1,\mathrm{CReLU}}| = \frac{1}{2^d}\left|-\binom{d-1}{(d-1)/2} + \frac{6}{d-1}\binom{d-1}{(d-5)/2}\right| = \Theta\left(\frac{1}{\sqrt{d}}\right). \tag{59}$$

The rest of the corollary is an application of Theorem 4. $\qquad\square$

### A.3.2 PROOF OF COROLLARY 2

Let $b \in \mathbb{R}$, by a straightforward calculation we have

$$\Delta_{d,b,\sigma}^{(a)} = \mathbb{E}_{x\sim\{\pm 1\}^d}\left[(-1)^{(d-a-\sum_{j=1}^{d-a} x_j)/2}\sigma\left(\sum_{j=1}^d x_j + b\right)\right],$$

$$= \mathbb{E}_{x\sim\{\pm 1\}^d}\left[(-1)^{(d-a-\sum_{j=1}^{d-a} x_j)/2}\sigma\left(\sum_{j=1}^{d-a} x_j + \sum_{j=d-a+1}^d x_j + b\right)\right],$$

$$= \mathbb{E}_{z\sim\{0,1\}^a}\mathbb{E}_{x\sim\{\pm 1\}^{d-a}}\left[(-1)^{(d-a-\sum_{j=1}^{d-a} x_j)/2}\sigma\left(\sum_{j=1}^{d-a} x_j - a + 2\sum_{j=d-a+1}^d z_j + b\right)\right],$$

$$= \sum_{\ell=0}^a \frac{\binom{a}{\ell}}{2^a}\mathbb{E}_{x\sim\{\pm 1\}^{d-a}}\left[(-1)^{(d-a-\sum_{j=1}^{d-a} x_j)/2}\sigma\left(\sum_{j=1}^{d-a} x_j - a + 2\ell + b\right)\right],$$

$$= \sum_{\ell=0}^a \frac{\binom{a}{\ell}}{2^a}\Delta_{d-a,b-a+2\ell,\sigma}^{(0)}.$$

Recall that $\sigma = \text{ReLU}$. Applying Lemma 1, where $c = c(d, b) = \lceil \frac{d-b}{2} \rceil$ and consequently $c(d - a, b - a + 2\ell) = c - \ell$, we have

$$\Delta_{d,b,\sigma}^{(a)} = \frac{1}{2^a} \sum_{\ell=0}^{a} \binom{a}{\ell} \Delta_{d-a,b-a+2\ell,\sigma} \ , \tag{60}$$

$$= \frac{1}{2^a} \sum_{\ell=0}^{a} \binom{a}{\ell} \frac{(-1)^{d-a+c-\ell}}{2^{d-a}} \left[ (d+b+2(\ell-a)) \binom{d-a-2}{c-\ell-2} \right. \tag{61}$$

$$\left. - (d-b-2\ell) \binom{d-a-2}{c-\ell-1} \right] \ , \tag{62}$$

$$=: dT(d, c, a) + C(d, c, a) + bB(d, c, a) \ , \tag{63}$$

where

$$B(d, c, a) = \frac{(-1)^{d-a+c}}{2^a} \sum_{\ell=0}^{a} \binom{a}{\ell} \frac{(-1)^{\ell}}{2^{d-a}} \left[ \binom{d-a-2}{c-\ell-2} + \binom{d-a-2}{c-\ell-1} \right] \ . \tag{64}$$

The following claim shows that a suitable lower bound for $|B(d, c, a)|$ is sufficient to obtain a lower bound for $|\Delta_{d,b,\sigma}^{(a)}|$.

**Claim 2.** *Let us assume that $b \in \mathbb{Z}$. If $|B(d, c, a)| > Cd^{-\alpha}$ (for some $C, \alpha > 0$), then either $|\Delta_{d,b,\sigma}^{(a)}| > \frac{C}{100} d^{-\alpha}$ or $|\Delta_{d,b+0.1,\sigma}^{(a)}| > \frac{C}{100} d^{-\alpha}$.*

*Proof.* Let us suppose there exist $C$ and $\alpha > 0$ such that $|B(d, c, a)| > Cd^{-\alpha}$. If $|\Delta_{d,b,\sigma}^{(a)}| > \frac{C}{100} d^{-\alpha}$, then we are done with the proof. On the other hand, if $|\Delta_{d,b,\sigma}^{(a)}| \leq \frac{C}{100} d^{-\alpha}$, then we have

$$|\Delta_{d,b+0.1,\sigma}^{(a)}| = |dT(d, c, a) + C(d, c, a) + (b+0.1)B(d, c, a)| \ , \tag{65}$$

$$= |\Delta_{d,b,\sigma}^{(a)} + 0.1B(d, c, a)| \ , \tag{66}$$

$$\geq 0.1|B(d, c, a)| - |\Delta_{d,b,\sigma}^{(a)}| \ , \tag{67}$$

$$\geq 0.1Cd^{-\alpha} - 0.01Cd^{-\alpha} \ , \tag{68}$$

$$> \frac{C}{100} d^{-\alpha} \ . \tag{69}$$

Equation 65 holds because for every $d$, $c(d, b) = \lceil \frac{d-b}{2} \rceil = \lceil \frac{d-(b+0.1)}{2} \rceil = c(d, b + 0.1)$, so the values of $T(d, c, a)$, $C(d, c, a)$ and $B(d, c, a)$ (see equation 63) are the same for $b$ and $b + 0.1$. $\quad\square$

It remains to find the order of magnitude for $|B(d, c, a)|$, in order to do so, let us consider a certain recursive sequence of differences of binomial coefficients.

Let $d \in \mathbb{N}$ and let $n = n(d) := \lfloor d/2 \rfloor$. For $n - d \leq k \leq n - a$ let

$$A_0(d, k) := 2^{-d} \binom{d}{n-k} \ , \tag{70}$$

$$A_a(d, k) := A_{a-1}(d, k) - A_{a-1}(d, k+1) \qquad \text{for } a > 0. \tag{71}$$

The main lemma that we will need is the following combinatorial bound:

**Lemma 2.** *Let $a \in \mathbb{N}$ and $k \in \mathbb{Z}$ such that either $a$ is even or $k \geq 0$. Then,*

$$|A_a(d, k)| = \Theta\left( d^{-1/2 - \lceil a/2 \rceil} \right) \ . \tag{72}$$

*Furthermore, for large enough $d$, it holds $\text{sign}(A_a(d, k)) = (-1)^{\lfloor a/2 \rfloor}$.*

We will prove the lemma only for the case of even $d$, as the calculations for $d$ odd are analogous. To that end, first let's give another formula for $A_a(d, k)$. Let

$$P_0(n, k) := 1 \ , \tag{73}$$

$$P_a(n, k) := (n + k + a)P_{a-1}(n, k) - (n - k)P_{a-1}(n, k+1) \quad \text{for } a > 0. \tag{74}$$

**Claim 3.** *For every $a \in \mathbb{N}$, even $d$, and $-n \le k \le n - a$:*

$$A_a(d, k) = 2^{-d} \binom{d}{n-k} \left( \prod_{i=1}^{a} \frac{1}{n+k+i} \right) P_a(n, k) . \tag{75}$$

*Proof.* By induction on $a$. The base case $a = 0$ is clear. For $a > 0$ we use induction and the definitions of $A_a$ and $P_a$:

$$A_a(d, k) = A_{a-1}(d, k) - A_{a-1}(d, k+1) \tag{76}$$

$$= 2^{-d} \binom{d}{n-k} \left( \prod_{i=1}^{a-1} \frac{1}{n+k+i} \right) P_{a-1}(n, k) \tag{77}$$

$$- 2^{-d} \binom{d}{n-k-1} \left( \prod_{i=1}^{a-1} \frac{1}{n+k+1+i} \right) P_{a-1}(n, k+1) \tag{78}$$

$$= 2^{-d} \binom{d}{n-k} \left( \prod_{i=1}^{a} \frac{1}{n+k+i} \right) \Big( (n+k+a)P_{a-1}(n, k) - (n-k)P_{a-1}(n, k+1) \Big) \tag{79}$$

$$= 2^{-d} \binom{d}{n-k} \left( \prod_{i=1}^{a} \frac{1}{n+k+1} \right) P_a(n, k) . \qquad \square$$

We will say that a degree $t$ polynomial of one variable $Q(k)$ has positive coefficients if all its coefficients until degree $t$ are positive. We state without proof a self-evident claim:

**Claim 4.** *Let $Q(k)$ be a polynomial with positive coefficients of degree $t > 0$. Then, $Q(k+1) - Q(k)$ is a polynomial with positive coefficients of degree $t - 1$.*

**Claim 5.** *Let $a \ge 0$. Then, there exist some polynomials $Q_{a,i}(k)$ such that*

$$P_a(n, k) = \sum_{i=0}^{\lfloor a/2 \rfloor} (-1)^i Q_{a,i}(k) \cdot n^i \tag{80}$$

*and $Q_{a,i}(k)$ is a degree $a - 2i$ polynomial with positive coefficients.*

*Proof.* We proceed by induction on $a$. For $a = 0$, the statement is clear with $Q_{0,0}(k) = 1$. Let $a > 0$. By equation 74,

$$P_a(n, k) = (n+k+1)P_{a-1}(n, k) - (n-k)P_{a-1}(n, k+1) \tag{81}$$

$$= n\big(P_{a-1}(n, k) - P_{a-1}(n, k+1)\big) + (k+a)P_{a-1}(n, k) + kP_{a-1}(n, k+1) . \tag{82}$$

By induction, the degree in $n$ of $P_{a-1}$ is $t := \lfloor \frac{a-1}{2} \rfloor$, and therefore the degree of $P_a$ is at most $t+1$. From equation 82, and assuming for convenience $Q_{a-1,-1}(k) = Q_{a-1,t+1}(k) = 0$, we have for every $0 \le i \le t+1$:

$$(-1)^i Q_{a,i}(k) = (-1)^{i-1} Q_{a-1,i-1}(k) - (-1)^{i-1} Q_{a-1,i-1}(k+1) \tag{83}$$

$$+ (-1)^i (k+a) Q_{a-1,i}(k) + (-1)^i k Q_{a-1,i}(k+1) \tag{84}$$

and hence

$$Q_{a,i}(x) = Q_{a-1,i-1}(k+1) - Q_{a-1,i-1}(k) + (k+a)Q_{a-1,i}(k) + kQ_{a-1,i}(k) . \tag{85}$$

For $0 \le i \le t$, by induction it holds that $(k+a)Q_{a-1,i}(k) + kQ_{a-1,i}(k)$ is a polynomial with positive coefficients of degree $a - 2i$. On the other hand $Q_{a-1,i-1}(k+1) - Q_{a-1,i-1}(k)$ is either zero (for $i = 0$ or when $Q_{a-1,i-1}$ has degree 0) or, by Claim 4, a polynomial with positive coefficients of degree $a - 2i$. Either way, $Q_{a,i}(k)$ is a polynomial with positive coefficients of degree $a - 2i$.

It remains to consider

$$Q_{a,t+1}(x) = Q_{a-1,t}(k+1) - Q_{a-1,t}(k) \tag{86}$$

If $a$ is even, then $\lfloor a/2 \rfloor = t+1$. Then, $Q_{a-1,t}(k)$ is a polynomial of degree 1 with positive coefficients and the right-hand side of equation 86 is a positive constant. If $a$ is odd (hence $\lfloor a/2 \rfloor = t$), then $Q_{a-1,t}(k)$ is a constant and therefore $Q_{a,t+1}(k) = 0$. In either case, we get that $P_a(n, k)$ has the decomposition according to equation 80. $\qquad \square$

*Proof of Lemma 2.* Let $d$ be even, $a \in \mathbb{N}$ and $k \in \mathbb{Z}$. Recall that by Claim 3,

$$A_a(d, k) = 2^{-d} \binom{d}{n - k} \left( \prod_{i=1}^{a} \frac{1}{n + k + i} \right) P_a(n, k) . \tag{87}$$

By Claim 5, the degree of $n$ in $P_a(n, k)$ is $t := \lfloor a/2 \rfloor$.

Furthermore, if $a$ is even, then the coefficient of $P_a$ at $n^t$ is equal to $(-1)^{a/2}$ multiplied by a positive constant. If $a$ is odd, the leading coefficient is $(-1)^{\lfloor a/2 \rfloor}$ multiplied by a linear function in $k$ with positive coefficients. It is easy to see that for $a$ even or $k \geq 0$, the leading coefficient of $P_a$ evaluated at $k$ is equal to $(-1)^{\lfloor a/2 \rfloor}$ multiplied by a positive constant. From this indeed it follows $\text{sign}(A_a(d, k)) = \text{sign}(P_a(n, k)) = (-1)^{\lfloor a/2 \rfloor}$ for $d$ large enough.

Furthermore, using known bounds on binomial coefficients

$$\left| A_a(d, k) \right| = \Theta \left( d^{-1/2 - a + t} \right) = \Theta \left( d^{-1/2 - \lceil a/2 \rceil} \right) . \tag{88}$$

As mentioned, the case of odd $d$ is proved by an analogous calculation. $\qquad\square$

Expanding the recursive definition, we can also write $A_a(d, k)$ as follows:

**Claim 6.** *Let $a, n \in \mathbb{N}$, and $k \in \mathbb{Z}$,*

$$A_a(d, k) = \sum_{\ell=0}^{a} (-1)^\ell \binom{a}{\ell} A_0(d, k + \ell) = \frac{1}{2^d} \sum_{\ell=0}^{a} (-1)^\ell \binom{a}{\ell} \binom{d}{n - k - \ell} . \tag{89}$$

*Proof.* The proof proceeds by induction on $a$.

For $a = 0$, there is nothing to prove. Let $a > 0$,

$$\sum_{\ell=0}^{a} (-1)^\ell \binom{a}{\ell} A_0(d, k + \ell) = \sum_{\ell=0}^{a} (-1)^\ell \binom{a-1}{\ell} A_0(d, k + \ell) + \sum_{\ell=0}^{a} (-1)^\ell \binom{a-1}{\ell-1} A_0(d, k + \ell) ,$$
$$\tag{90}$$

$$= \sum_{\ell=0}^{a-1} (-1)^\ell \binom{a-1}{\ell} A_0(d, k + \ell) + \sum_{\ell=1}^{a} (-1)^\ell \binom{a-1}{\ell-1} A_0(d, k + \ell) ,$$
$$\tag{91}$$

$$= \sum_{\ell=0}^{a-1} (-1)^\ell \binom{a-1}{\ell} A_0(d, k + \ell) \tag{92}$$

$$+ \sum_{\ell=0}^{a-1} (-1)^{\ell+1} \binom{a-1}{\ell} A_0(d, k + \ell + 1) , \tag{93}$$

$$= A_{a-1}(d, k) - A_{a-1}(d, k + 1) , \tag{94}$$

$$= A_a(d, k) . \tag{95}$$

Equation 91 holds by the convention $\binom{a-1}{a} = \binom{a-1}{-1} = 0$. $\qquad\square$

Recall that equation 64 can be rewritten as

$$B(d, c, a) = \frac{(-1)^{d-a+c}}{2^d} \sum_{\ell=0}^{a} (-1)^\ell \binom{a}{\ell} \left[ \binom{d - a - 2}{c - \ell - 2} + \binom{d - a - 2}{c - \ell - 1} \right] . \tag{96}$$

Comparing this with Claim 6, we have

$$B(d, c, a) = \frac{(-1)^{d-a+c}}{2^{a+2}} \left( A_a(d - a - 2, n - c + 2) + A_a(d - a - 2, n - c + 1) \right) , \tag{97}$$

where $n = n(d - a - 2) = \lfloor \frac{d-a-2}{2} \rfloor$ and $c = \lceil \frac{d-b}{2} \rceil$.

From this point we can conclude the proof of Corollary 2. Recall that we choose the biases $b_i$ uniformly from the set $\{a + 2, a + 2 + 0.1\}$. In particular, taking $b = a + 2$,

$$n - c + 1 = \lfloor \frac{d - a - 2}{2} \rfloor - \lceil \frac{d - a - 2}{2} \rceil + 1 \geq 0 . \tag{98}$$

Furthermore, it is easy to see that $\{n - c + 1, n - c + 2\} \subseteq \{0, 1, 2\}$. Therefore, applying Lemma 2 with $k = 0, 1, 2$ we get, for large enough $d$,

$$|B(d, c, a)| = \frac{1}{2^{a+2}} \left( \left| A_a(d - a - 2, n - c + 2) \right| + \left| A_a(d - a - 2, n - c + 1) \right| \right) , \tag{99}$$

$$= \Theta \left( d^{-\frac{1}{2} - \lceil \frac{a}{2} \rceil} \right) . \tag{100}$$

Equation 100 holds because $a$ is a fixed natural number and from Lemma 2, for $d$ large enough, $\text{sign}(A_a(d - a - 2, n - c + 2)) = \text{sign}(A_a(d - a - 2, n - c + 1))$.

From equation 100 and Claim 2, we get that $\Delta^2 = \mathbb{E}_{b \sim \mathcal{B}} \left[ (\Delta_{d,b,\sigma}^{(a)})^2 \right] \geq C d^{-1 - 2\lceil \frac{a}{2} \rceil}$, for some $C > 0$. The rest of the proof of Corollary 2 follows from Theorem 4.

### A.4 Almost Full Parities $d - 1$ and $d - 2$

Here, we provide a simpler calculation for the specific cases of almost full-parities, namely $k = d - 1$ and $k = d - 2$.

**Corollary 5.** *In the cases of almost full parities $a = 1$ and $a = 2$, let $b = -2$ if $d$ is even and $b = -1$ for $d$ odd. For $\sigma = \text{ReLU}$ it holds $\Delta^2 = \Theta(d^{-3})$. Accordingly, $n \geq \Omega(d^6)$ hidden neurons are sufficient for strong learning in one GD step.*

Let $c := c(d, b) = \lceil \frac{d-b}{2} \rceil$, thus $c(d - 1, b - 1) = c$ and $c(d - 1, b + 1) = c - 1$. Using equation 37 and equation 60, we have:

$$\Delta_{d,b,\text{ReLU}}^{(1)} = \frac{1}{2} \Delta_{d-1,b-1,\text{ReLU}} + \frac{1}{2} \Delta_{d-1,b+1,\text{ReLU}} , \tag{101}$$

$$= \frac{1}{2} \left[ \frac{(-1)^{d-1+c}}{2^{d-1}} \left( (d + b - 2) \binom{d-3}{c-2} - (d - b) \binom{d-3}{c-1} \right) \right] \tag{102}$$

$$+ \frac{1}{2} \left[ -\frac{(-1)^{d-1+c}}{2^{d-1}} \left( (d + b) \binom{d-3}{c-3} - (d - b - 2) \binom{d-3}{c-2} \right) \right] , \tag{103}$$

$$= \frac{(-1)^{d+c}}{2^d} \left[ (d - b) \binom{d-3}{c-1} - 2(d - 2) \binom{d-3}{c-2} + (d + b) \binom{d-3}{c-3} \right] . \tag{104}$$

Then for $d$ even i.e. $b = -2$ and $c = \frac{d+2}{2}$, we obtain

$$|\Delta_{d,-2,\text{ReLU}}^{(1)}| = \frac{1}{2^d} \left| (d + 2) \binom{d-3}{\frac{d+2}{2} - 1} - 2(d - 2) \binom{d-3}{\frac{d+2}{2} - 2} + (d - 2) \binom{d-3}{\frac{d+2}{2} - 3} \right| ,$$

$$= \frac{1}{2^d} \frac{(d-3)!}{(\frac{d+2}{2} - 1)! (d - \frac{d+2}{2})!} \left| (d + 2)(d - \frac{d+2}{2} - 1)(d - \frac{d+2}{2}) \right.$$

$$\left. - 2(d - 2)(\frac{d+2}{2} - 1)(d - \frac{d+2}{2}) + (d - 2)(\frac{d+2}{2} - 2)(\frac{d+2}{2} - 1) \right| ,$$

$$= \frac{1}{2^d (d-1)(d-2)} \binom{d-1}{\frac{d+2}{2} - 1} \left| \frac{(d - 2) [(d + 2)(d - 4) - d(d - 2)]}{4} \right| ,$$

$$= \frac{2}{2^d (d-1)} \binom{d-1}{\frac{d+2}{2} - 1} ,$$

$$= \Theta \left( \frac{1}{d\sqrt{d}} \right) .$$

Similarly, for $d$ odd i.e. $b = -1$ and $c = \frac{d+1}{2}$, we have

$$|\Delta^{(1)}_{d,-1,\text{ReLU}}| = \frac{1}{2^d(d-2)}\binom{d-1}{\frac{d+1}{2}-1} = \Theta(\frac{1}{d\sqrt{d}}) \, . \tag{105}$$

Let's provide a similar analysis for $\Delta^{(2)}_{d,b,\text{ReLU}}$. We have $c(d-2,b-2) = c, c(d-2,b) = c-1$ and $c(d-2,b+2) = c-2$, so

$$\begin{aligned}
\Delta^{(2)}_{d,b,\text{ReLU}} &= \frac{1}{4}\Delta_{d-2,b-2,\text{ReLU}} + \frac{1}{2}\Delta_{d-2,b,\text{ReLU}} + \frac{1}{4}\Delta_{d-2,b+2,\text{ReLU}} \, , \\
&= \frac{(-1)^{d+c}}{2^d}\Big[(d+b-4)\binom{d-4}{c-2} - (d-b)\binom{d-4}{c-1} - 2(d+b-2)\binom{d-4}{c-3} \\
&\quad + 2(d-b-2)\binom{d-4}{c-2} + (d+b)\binom{d-4}{c-4} - (d-b-4)\binom{d-4}{c-3}\Big] \, , \\
&= \frac{(-1)^{d+c}}{2^d}\Big[-(d-b)\binom{d-4}{c-1} + (3d-b-8)\binom{d-4}{c-2} \\
&\quad - (3d+b-8)\binom{d-4}{c-3} + (d+b)\binom{d-4}{c-4}\Big] \, .
\end{aligned}$$

Then for $d$ even i.e. $b = -2$ and $c = \frac{d+2}{2}$, and observing that $\binom{d-4}{\frac{d+2}{2}-2} = \binom{d-4}{\frac{d+2}{2}-4}$, we get

$$\begin{aligned}
|\Delta^{(2)}_{d,-2,\text{ReLU}}| &= \frac{1}{2^d}\left|-(d+2)\binom{d-4}{\frac{d+2}{2}-1} + (4d-8)\binom{d-4}{\frac{d+2}{2}-2} - (3d-10)\binom{d-4}{\frac{d+2}{2}-3}\right| \, , \\
&= \frac{2(d-6)}{2^d(d-3)(d-2)}\binom{d-2}{\frac{d+2}{2}-1} \, , \\
&= \Theta\left(\frac{1}{d\sqrt{d}}\right) \, .
\end{aligned}$$

With the same procedure we can show that for $d$ odd i.e. $b = -1$ and $c = \frac{d+1}{2}$ we have

$$\left|\Delta^{(2)}_{d,-1,\text{ReLU}}\right| = \frac{2}{2^d(d-2)}\binom{d-2}{\frac{d+1}{2}-1} = \Theta\left(\frac{1}{d\sqrt{d}}\right) \, .$$

The rest of Corollary 5 follows easily from Theorem 4.

## A.5 POSITIVE RESULT: SGD FOR HINGE LOSS

As in Section 4.1, we consider the two layer architecture, this time with possibly perturbed Rademacher hidden layer initialization $N(x) = \sum_{i=1}^n v_i \text{ReLU}((w_i + g_i) \cdot x + b_i)$, that is $w_{ij} \sim \text{Rad}(1/2)$ and $g_{ij} \sim \mathcal{N}(0, \sigma^2)$. Other weights are initialized as before, i.e., hidden layer biases are $b_i = 0$ for $d$ even and $b_i = -1$ for $d$ odd, and output layer weights are $v_i = 0$. As in the case of the correlation loss, the exact bias values are not crucial.

The training is with hinge loss $L_\beta(y, \hat{y}) = \max(0, \beta - y\hat{y})$ for some $\beta \geq 0$ under i.i.d. samples from *any fixed probability distribution* on $\{\pm 1\}^d$. For simplicity we consider batch size 1 SGD, though larger batches could also be used. This time we allow a more realistic setting where both layers are trained.

**Theorem 9.** *For the network described above, for $\sigma \leq C/d$ for sufficiently small $C > 0$, except with probability $3\exp(-d)$ over the choice of initialization the following holds:*

*Let $\mathcal{D}$ be a distribution on $\{\pm 1\}^d$, $\epsilon > 0$ and $0 < \delta \leq 1/2$. If $n \geq \Omega(d^4)$ and $n \leq \text{poly}(d)$, then after training with batch size one SGD for some choices of $T = \text{poly}(d)\frac{1}{\epsilon}\ln\frac{1}{\delta}$ and learning rate $\gamma = 1/\text{poly}(d)$, using hinge loss $L_\beta$ for $0 \leq \beta \leq O(d^2 n\gamma)$, except with probability $\delta$ over the choice of i.i.d. training samples from $\mathcal{D}$, it holds $\Pr_{x \sim \mathcal{D}}\left[\text{sign}(N^T(x)) \neq f(x)\right] \leq \epsilon$ .*

Theorem 9 follows from the bound on the number of nonzero SGD updates:

**Theorem 10.** *If $\sigma \leq C/d$ for sufficiently small $C > 0$, $n \geq \Omega(d^4)$ and $n \leq \text{poly}(d)$, then, except with probability $3\exp(-d)$ over the choice of initialization, the above network trained with batch size one SGD algorithm on the full parity function on any sequence of samples from $\{\pm 1\}^d$ with learning rate $0 < \gamma \leq O(d^{-3.5})$ and the hinge loss $L_\beta$ for $0 \leq \beta \leq 16d^2n\gamma$, will perform at most $O(d^3)$ nonzero updates.*

A crucial consequence of Theorem 4 is that the full parity is linearly separable at initialization:

**Lemma 3.** *If $v^1$ are the output weights after one step of noiseless GD correlation loss algorithm, and we take $v^* = v^1/\|v^1\|$, then, $|v_i^*| = \frac{1}{\sqrt{n}}$ for every $1 \leq i \leq n$ and, except with probability $\exp(-d)$, for all $x \in \{\pm 1\}^d$,*

$$f_a(x) \sum_{i=1}^{n} v_i^* \sigma(w_i \cdot x + b_i) \geq \frac{\sqrt{n}\Delta}{2} . \tag{106}$$

*Proof.* By equation 21, except with probability $\exp(-d)$ for every $x \in \{\pm 1\}^d$ we have

$$f_a(x)N^1(x) = f_a(x) \sum_{i=1}^{n} v_i^1 \sigma(w_i \cdot x + b_i) \geq \frac{\gamma n\Delta^2}{2} .$$

Recall that $v_i^1 = \gamma\Delta_{d,b_i,\sigma}^{(a)} \prod_{j=1}^{d-a} w_{ij}$ and let $v^* = v^1/\|v^1\|$. In particular, it follows $|v_i^*| = 1/\sqrt{n}$ for every $i$. We also have $\|v^1\| \leq \gamma\Delta\sqrt{n}$

$$f_a(x) \sum_{i=1}^{n} v_i^* \sigma(w_i \cdot x + b_i) \geq \frac{\gamma n\Delta^2}{2\|v^1\|} \geq \frac{\sqrt{n}\Delta}{2} . \qquad \square$$

### A.6 Proof of Theorem 9

In this proof we will apply the following result about hinge loss SGD:

**Lemma 4** (Lemma 4 in Nachum & Yehudayoff (2020)). *Let $f : \mathcal{X} \to \{-1, 1\}$ be a function from some finite domain $\mathcal{X} \subseteq \mathbb{R}^d$ such that $\|x\| \leq R$ for every $x \in \mathcal{X}$ and some $R \geq 1$. Consider a one layer $\text{ReLU}$ neural network at initialization. For $x \in \mathcal{X}$, let $z_x \in \mathbb{R}^n$ be the embedding vector $z_{x,i} = \text{ReLU}(w_i \cdot x + b_i)$ and assume that $\|z_x\| \leq R_z$ for every $x \in \mathcal{X}$.*

*Furthermore, assume that there exists $c > 0$ and a choice of output layer weights $v^* \in \mathbb{R}^n$ with $\|v^*\| = 1$ such that $f(x) \sum_{i=1}^{n} v_i^* \text{ReLU}(w_i \cdot x + b_i) \geq c$ for every $x \in \mathcal{X}$.*

*Then, using learning rate $0 < \gamma \leq \frac{1}{500R} \cdot \frac{c^2}{R_z^2}$ and $0 \leq \beta \leq 4R_z^2\gamma$, the batch size one SGD algorithm using hinge loss $L(x, y) = \max(0, \beta - N(x)y)$ run on any sequence of samples from $\mathcal{X}$ will perform at most $20R_z^2/c^2$ nonzero updates.*

Let $\mathcal{X} := \{\pm 1\}^d$, for all $x \in \mathcal{X}$ we have $\|x\| = \sqrt{d}$. First, let us consider the case of non-perturbed Rademacher initialization.

For $x \in \mathcal{X}$, let $z_x \in \mathbb{R}^n$ be its embedding vector i.e., $z_{x,i} = \text{ReLU}(w_i \cdot x + b_i)$, we have $\|z_x\| \leq (d+1)\sqrt{n} \leq 2d\sqrt{n}$. By Lemma 3 applied for $a = 0$, (see equation 106), except with probability $\exp(-d)$ over the choice of $w$, there exists $v^* \in \mathbb{R}^n$ with $|v_i^*| = 1/\sqrt{n}$ such that, for all $x \in \mathcal{X}$ we have

$$f(x) \sum_{i=1}^{n} v_i^* \text{ReLU}(w_i \cdot x + b_i) \geq \frac{\sqrt{n}}{18\sqrt{d}} . \tag{107}$$

Now consider the perturbed initialization $\text{ReLU}((w_i + g_i) \cdot x + b_i)$, where $g \sim \mathcal{N}\left(0, \frac{C^2}{d^2} \cdot \mathbb{I}\right)$ for some $C \leq \frac{1}{72}$. Let $\mathcal{E}_1$ be the event that there exists $1 \leq i \leq n$ such that $\|g_i\| \geq \sqrt{d}$ and $\mathcal{E}_2$ that there exists $x$ such that $\sum_{i=1}^{n} |g_i \cdot x| \geq \frac{n}{36\sqrt{d}}$. First, let us establish that each of these events occurs with probability at most $\exp(-d)$.

Let us start with $\mathcal{E}_1$. Since $\mathbb{E}\|g_i\|^2 = \frac{C^2}{d}$, by subgaussian concentration we have

$$\Pr\left[\|g_i\|^2 \geq \frac{C^2}{d} + t\right] \leq \exp\left(-\Omega\left(\frac{d^2 t^2}{C^4}\right)\right) . \tag{108}$$

Substituting $t = d/2$, we have in particular $\Pr[\|g_i\|^2 \geq d] \leq \exp(-\Omega(d^4))$. Taking union bound over $n = \text{poly}(d)$, we have $\Pr[\exists i : \|g_i\|^2 \geq d] \leq \exp(-d)$.

As for $\mathcal{E}_2$, note that $\mathbb{E}|g_i \cdot x| = \frac{\sqrt{2}C}{\sqrt{\pi d}} \leq \frac{1}{72\sqrt{d}}$. Therefore, again by subgaussian concentration, for any fixed $x$,

$$\Pr\left[\sum_{i=1}^n |g_i \cdot x| \geq \frac{n}{36\sqrt{d}}\right] \leq \Pr\left[\sum_{i=1}^n |g_i \cdot x| \geq \mathbb{E}\left[\sum_{i=1}^n |g_i \cdot x|\right] + \frac{n}{72\sqrt{d}}\right] \leq \exp(-\Omega(n)) , \tag{109}$$

which is smaller than $\exp(-d)$ for $n \geq \Omega(d^4)$.

If neither $\mathcal{E}_1$ or $\mathcal{E}_2$ happens, then for every $x$ we have

$$f(x) \sum_{i=1}^n v_i^* \text{ReLU}((w_i + g_i) \cdot x + b_i) \geq f(x) \sum_{i=1}^n v_i^* \text{ReLU}(w_i \cdot x + b_i) - \frac{1}{\sqrt{n}} \sum_{i=1}^n |g_i \cdot x| \tag{110}$$

$$\geq \frac{\sqrt{n}}{36\sqrt{d}} . \tag{111}$$

Furthermore, for every $x$ and $i$ it holds $|(w_i + g_i) \cdot x + b_i| \leq (\|w_i\| + \|g_i\|)\sqrt{d} + 1 \leq 3d$ and consequently $\|z_x\| \leq 3d\sqrt{n}$.

Therefore by applying Lemma 4 with $R := \sqrt{d}, R_z := 3d\sqrt{n}$ and $c := \frac{\sqrt{n}}{36\sqrt{d}}$, we conclude that using learning rate $0 < \gamma \leq \frac{1}{500\sqrt{d}} \cdot \frac{1}{9 \cdot 36^2 d^3} = O\left(\frac{1}{d^{3.5}}\right)$, the SGD algorithm using the hinge loss $L(y, \hat{y}) = \max\{0, \beta - y\hat{y}\}$, with $0 \leq \beta \leq 36d^2 n\gamma$, will perform at most $O(d^3)$ nonzero updates after which all samples will be classified correctly. $\square$

## A.7 PROOF OF THEOREM 9

First, note that we can choose values of $\gamma = 1/\text{poly}(d)$ and $0 \leq \beta \leq O(d^2 n\gamma)$ such that Theorem 10 applies. In line with Theorem 10, fix an initialization such that the SGD algorithm running on i.i.d. samples from $\mathcal{D}$ performs at most $C_0 := Cd^3$ nonzero updates, where $C$ is a universal constant.

Let us run the training until there are $K := \frac{1}{\epsilon}(\ln 1/\delta + \ln C_0)$ zero updates in a row. As the number of nonzero updates is at most $C_0$, the algorithm runs for at most $C_0(1 + K) = \text{poly}(d)\frac{1}{\epsilon}\ln\frac{1}{\delta}$ steps.

Finally, let us argue that that the classification error does not exceed $\epsilon$ except with probability $\delta$. To that end, define a "bad event" $\mathcal{E}$ as follows: There exists $t$ such that:

1. A nonzero update occurs at time $t$.
2. There are $K$ zero updates in a row immediately following $t$.
3. $\Pr_{x \sim \mathcal{D}}[\text{sign}(N^{t+1}(x)) \neq f(x)] > \epsilon$.

It should be clear that if $\mathcal{E}$ does not occur, then at the final time $T$ it holds $\Pr_{x \sim \mathcal{D}}[\text{sign}(N^T(x)) \neq f(x)] \leq \epsilon$.

Fix some time $t$ such that the first and third condition above are satisfied. Clearly, if the error probability exceeds $\epsilon$, then so does the probability of a nonzero update. By independence (and the fact that only a nonzero update can change the network), the probability that there will be $K$ zero updates in a row is at most $(1 - \epsilon)^K$. By union bound over at most $C_0$ nonzero updates,

$$\Pr[\mathcal{E}] \leq C_0(1 - \epsilon)^K \leq \delta .$$

# B  PROOFS FOR SECTION 5.1

## B.1  PROOF OF THEOREM 6

For brevity, we denote the population gradient at $\theta$ for a target function $f$ by

$$\Gamma_f(\theta) := \mathbb{E}_x \left[ \nabla_\theta L(f(x), \theta, x) \right]. \tag{112}$$

To prove our results we couple the dynamics of the network's weights $\theta^t$ with the dynamics of the 'Junk-Flow'. The junk-flow is the dynamics of the parameters of a network trained on random labels. For that purpose let

$$\Gamma_r(\theta) := \mathbb{E}_x \left[ \frac{1}{2} \left( \nabla_\theta L(1, \theta, x) + \nabla_\theta L(-1, \theta, x) \right) \right]. \tag{113}$$

In other words, $\Gamma_r(\theta)$ is the expected population gradient of random classification problem where $r(x) \sim \mathrm{Rad}(1/2)$ independently for every input $x$.

**Definition 6** (Junk-Flow). *Let us define the junk-flow as the sequence $\psi^t \in \mathbb{R}^P$ that satisfies the following iterations:*

$$\psi^0 = \theta^0, \tag{114}$$

$$\psi^{t+1} = \psi^t - \gamma \left( \Gamma_r(\psi^t) + \xi^t \right), \tag{115}$$

*where $\xi^t \overset{iid}{\sim} \mathcal{N}(0, \mathbb{I}\tau^2)$ We call $\gamma$ the learning rate and $\tau$ the noise-level of the noisy-GD algorithm used to train the network $\mathrm{NN}(x; \theta)$.*

We show that $\theta^T$ and $\psi^T$ are close in terms of the total variation distance. Let us look at the total variation distance between the law of $\theta^T$ and $\psi^T$, which, by abuse of notation, we denote by $\mathrm{TV}(\theta^T; \psi^T)$.

**Lemma 5.** *Let $\mathrm{TV}(\theta^T; \psi^T)$ be the total variation distance between the law of $\theta^T$ and $\psi^T$. Then,*

$$\mathrm{TV}(\theta^T; \psi^T) \leq \frac{1}{2\tau} \sum_{t=0}^{T-1} \sqrt{\mathrm{GAL}_f(\psi^t)}. \tag{116}$$

The proof of Lemma 5 can be found in Section B.2.

Recalling that $f : \mathbb{R}^P \to \{\pm 1\}$, we have

$$\mathbb{P}\left[ \mathrm{sign}(\mathrm{NN}(x; \theta^T)) = f(x) \right] \leq \mathbb{P}\left[ \mathrm{sign}(\mathrm{NN}(x; \psi^T)) = f(x)) \right] + \mathrm{TV}(\theta^T; \psi^T) \tag{117}$$

$$\leq \frac{1}{2} + \mathrm{TV}(\theta^T; \psi^T), \tag{118}$$

$$\leq \frac{1}{2} + \frac{1}{2\tau} \sum_{t=0}^{T-1} \sqrt{\mathrm{GAL}_f(\psi^t)}. \tag{119}$$

In equation 118 we used the fact that the initialization is symmetric around 0. Since for the correlation loss $\Gamma_r(\theta) = 0$, the junk flow just adds independent Gaussian noise and the distribution of the output layer weights $\psi^T$ is also symmetric around 0 (and independent of other weights). Therefore, the distribution of $\mathrm{sign}(\mathrm{NN}(x; \psi^T))$ is also symmetric around 0 for every fixed $x$. Finally, in equation 119 we used Lemma 5.

We are now left with showing that the right-hand-side of equation 116 is small, i.e. that the junk-flow dynamics does not pick correlation with $f$ along its trajectory. Again, for the correlation loss, $\Gamma_r(\psi^t) = 0$ for all $t$, thus for all $t$, $\psi^t = A + H_\sigma + \sqrt{t}\gamma\tau H$, where $H \sim \mathcal{N}(0, \mathbb{I}_P)$. Thus, the result follows by the assumption in equation 5.

## B.2  PROOF OF LEMMA 5

This proof follows a similar argument that is used in (Abbe & Sandon (2020); Abbe & Boix-Adsera (2022)). In the following let us write $\theta := \theta^{T-1}$ and $\psi := \psi^{T-1}$ for readability. The total variation

distance $\mathrm{TV}(\theta^T; \psi^T)$ can be bounded in terms of $\theta$ and $\psi$ as follows:

$$\mathrm{TV}(\theta^T; \psi^T) = \mathrm{TV}\left(\theta - \gamma(\Gamma_f(\theta) + Z^t); \psi - \gamma(\Gamma_r(\psi) + \xi^t)\right) \tag{120}$$

$$\overset{a)}{\leq} \mathrm{TV}\left(\theta - \gamma(\Gamma_f(\theta) + Z^t); \psi - \gamma(\Gamma_f(\psi) + Z^t)\right) \tag{121}$$
$$+ \mathrm{TV}\left(\psi - \gamma(\Gamma_f(\psi) + Z^t); \psi - \gamma(\Gamma_r(\psi) + \xi^t)\right) \tag{122}$$

$$\overset{b)}{\leq} \mathrm{TV}\left(\theta; \psi\right) \tag{123}$$
$$+ \mathbb{E}_\psi \mathrm{TV}\left(\psi - \gamma(\Gamma_f(\psi) + Z^t); \psi - \gamma(\Gamma_r(\psi) + \xi^t) \mid \psi\right) \tag{124}$$

$$\overset{c)}{\leq} \mathrm{TV}(\theta; \psi) \tag{125}$$
$$+ \mathbb{E}_\psi \sqrt{\frac{1}{2} D_{\mathrm{KL}}\left(\psi - \gamma(\Gamma_f(\psi) + Z^t) \| \psi - \gamma(\Gamma_r(\psi) + \xi^t) \mid \psi\right)} \tag{126}$$

$$\overset{d)}{\leq} \mathrm{TV}\left(\theta^{T-1}; \psi^{T-1}\right) + \frac{1}{2\tau\gamma} \mathbb{E}_\psi \|\gamma\Gamma_f(\psi) - \gamma\Gamma_r(\psi)\|_2 \tag{127}$$

$$= \mathrm{TV}\left(\theta^{T-1}; \psi^{T-1}\right) + \frac{1}{2\tau} \mathbb{E}_\psi \|\Gamma_f(\psi) - \Gamma_r(\psi)\|_2 \tag{128}$$

where in $a)$ we used the triangle inequality, in $b)$ the data processing inequality (DPI) and triangle inequality again, in $c)$ Pinsker's inequality. Finally, $d)$ follows since, conditional on $\psi$, both distributions in the KL divergence are Gaussian, and due to the known formula $D_{\mathrm{KL}}(\mathcal{N}(\mu, \sigma\,\mathrm{Id}), \mathcal{N}(\mu', \sigma\,\mathrm{Id})) = \frac{\|\mu-\mu'\|^2}{2\sigma^2}$. Thus,

$$\mathrm{TV}(\theta^T; \psi^T) \leq \frac{1}{2\tau} \sum_{t=0}^{T-1} \mathbb{E}_{\psi^t} \|\Gamma_f(\psi^t) - \Gamma_r(\psi^t)\|_2 \tag{129}$$

$$\overset{(a)}{\leq} \frac{1}{2\tau} \sum_{t=0}^{T-1} \sqrt{\mathrm{GAL}_f(\psi^t)}, \tag{130}$$

where in $(a)$ we used Cauchy-Schwartz.

## B.3 PROOF OF COROLLARY 3

Let us state a claim about Gaussians with slightly different variances:

**Claim 7.** *Let $F : \mathbb{R}^P \to \mathbb{R}$ be a function such that $0 \leq F(x) \leq R$ for all $x \in \mathbb{R}^P$. Let $\theta \sim \mathcal{N}(\mu, D)$, for some $\mu \in \mathbb{R}^P$ and $D$ a diagonal matrix with diagonal entries $(\sigma_1^2, \ldots, \sigma_P^2)$, and let $\psi \sim \mathcal{N}(\mu, D')$ for some other diagonal $D'$ with entries $((\sigma_1')^2, \ldots, (\sigma_P')^2)$ such that $(\sigma_i')^2 \leq \sigma_i^2(1 + 1/P)$ for every $1 \leq i \leq P$.*

*If $\mathbb{E}F(\theta) \leq \epsilon$, for some $\epsilon$, then $\mathbb{E}F(\psi) \leq (4R + 1)\epsilon^{1/9}$.*

*Proof.* Let $M > 0$ and define the event $\mathcal{E}_M$ as $\sqrt{\sum_{i=1}^{P} \left(\frac{\psi_i - \mu_i}{\sigma_i}\right)^2} > M$. By Gaussian concentration (formula (3.5) in Ledoux & Talagrand (2013), see also MO2 (2020)):

$$\Pr[\mathcal{E}_M] \leq 4\exp\left(-\frac{M^2}{8\mathbb{E}\sum_{i=1}^{P} \left(\frac{\psi_i - \mu_i}{\sigma_i}\right)^2}\right) \leq 4\exp\left(-\frac{M^2}{16P}\right). \tag{131}$$

At the same time, if $\sum_{i=1}^{P} \left( \frac{x_i - \mu_i}{\sigma_i} \right)^2 \leq M^2$, then the density functions $\varphi_\theta$ and $\varphi_\psi$ satisfy

$$\varphi_\psi(x) = \prod_{i=1}^{P} \frac{1}{\sqrt{2\pi}\sigma_i'} \exp\left(-\frac{(x_i - \mu_i)^2}{2(\sigma_i')^2}\right) \tag{132}$$

$$\leq \exp\left(\sum_{i=1}^{P} \frac{(x_i - \mu_i)^2}{2\sigma_i^2} \cdot \frac{(\sigma_i')^2 - \sigma_i^2}{(\sigma_i')^2}\right) \prod_{i=1}^{P} \frac{1}{\sqrt{2\pi}\sigma_i} \exp\left(-\frac{(x_i - \mu_i)^2}{2\sigma_i^2}\right) \tag{133}$$

$$\leq \exp\left(\frac{M^2}{2P}\right) \varphi_\theta(x) . \tag{134}$$

So,

$$\mathbb{E}F(\psi) = \int_{x \in \mathcal{E}_M} F(x)\varphi_\psi(x) + \int_{x \notin \mathcal{E}_M} F(x)\varphi_\psi(x) \tag{135}$$

$$\leq \exp\left(\frac{M^2}{2P}\right) \epsilon + 4R \exp\left(-\frac{M^2}{16P}\right) . \tag{136}$$

Substituting $M := \sqrt{\frac{16P}{9} \ln 1/\epsilon}$, we get the bound. $\qquad\square$

Let $F(\theta) := \|\Gamma_f(\theta) - \Gamma_r(\theta)\|_2^2$. Conditional on the value of $A$, the distribution of $\theta^0$ is Gaussian $\theta^0 \sim \mathcal{N}(A, \sigma^2 D_A)$ where $D_A$ is diagonal with entries $(D_A)_{pp} = \text{Var} A_p$. Let $0 \leq \lambda \leq \gamma^2 \tau^2 T$. Then, the distribution of $\theta^0 + \lambda H$ for $H$ standard gaussian is $\theta^0 + \lambda H \sim \mathcal{N}(A, \sigma^2 D_A + \lambda^2 \mathbb{I}_P)$. Therefore, by assumption for every $1 \leq p \leq P$ it holds

$$\sigma^2 \text{Var} A_p + \lambda^2 \leq \sigma^2 \text{Var} A_p + \gamma^2 \tau^2 T \leq \sigma^2 \text{Var} A_p \left(1 + \frac{1}{P}\right) . \tag{137}$$

By Claim 7 (and averaging over $A$), it follows

$$\text{GAL}_f(\theta^0 + \lambda H) = \mathbb{E}F(\sigma^0 + \lambda H) \leq (4R + 1)\mathbb{E}F(\theta^0)^{1/9} = (4R + 1) \text{GAL}_f(\theta^0)^{1/9} . \tag{138}$$

Equation 6 now follows directly by applying Theorem 6.

### B.4 Proof of Corollary 4

For Corollary 4 we focus on fully-connected networks of bounded depth. For simplicity, we consider fully connected networks with one bias vector in the first layer, but we believe that, with a more involved argument, one could extend the proof and include bias vectors in all layers. In particular, we use the following notation:

$$x^{(1)}(\theta) = W^{(1)}x + b^{(1)} \tag{139}$$

$$x^{(l)}(\theta) = W^{(l)}\sigma(x^{(l-1)}(\theta)), \qquad l = 2, ..., L, \tag{140}$$

and we denote the network function as $\text{NN}(x; \theta) = x^{(L)}(\theta)$. We assume that the activation $\sigma$ satisfies the $H$-weak homogeneity assumption of Def. 5. We assume that each parameter of the network is independently initialized as $\theta_p^0 \sim \mathcal{N}(0, v_{l_p}^2)$, where $l_p$ denotes the layer of parameter $\theta_p$, for $p \in [P]$.

Corollary 4 follows from the following Proposition.

**Proposition 3.** *Let* $\text{NN}(x; \theta)$ *be a network that satisfies the assumptions of Corollary 4. Then, if* $\text{GAL}_f(\theta^0) < \epsilon$,

$$\text{GAL}_f(\theta^0 + \gamma\lambda H) \leq \prod_{l=1}^{L} \left(1 + \frac{\gamma^2 \lambda^2}{v_l^2}\right)^H \epsilon, \tag{141}$$

*where* $H \sim \mathcal{N}(0, \mathbb{I}_P)$.

B.5 PROOF OF PROPOSITION 3

Recall, that $\theta^0 \sim \mathcal{N}(0, V)$, where $V$ is a $P \times P$ diagonal matrix such that $V_{pp} = v_{l_p}^2$, where $l_p$ is the layer of parameter $p$, and $\psi_p^t \sim \mathcal{N}(0, U)$, where $U$ is a $P \times P$ diagonal matrix such that $U_{pp} = v_{l_p}^2 + t\gamma^2\tau^2$. Thus, $U = \overline{C}V\overline{C}^T$, where $\overline{C}$ is a $P \times P$ diagonal matrix such that

$$\overline{C}_{pp} = \sqrt{1 + \frac{t\gamma^2\tau^2}{v_{l_p}^2}}. \tag{142}$$

**Definition 7** ($\overline{C}$-Rescaling). *Let* $\mathrm{NN}(x; \theta)$ *be an $L$-layers network, with parameters* $\theta \in \mathbb{R}^P$. *Let* $C^{(1)}, ..., C^{(L)}$ *be $L$ positive constants, and let* $\overline{C}$ *be a $P \times P$ diagonal matrix such that* $\overline{C}_{pp} = C^{(l_p)}$ *where $l_p$ is the layer of parameter $\theta_p$. We say that the vector* $\overline{C} \cdot \theta$ *is a* $\overline{C}$-rescaling *of $\theta$.*

**Definition 8** (Weak Positive Homogeneity (SPH)). *We say that an architecture is $H$-weakly homogeneous (H-SPH) if for all $\overline{C}$-rescaling such that* $\min_{p \in [P]} \overline{C}_{pp} > 1$, *it holds:*

$$\mathrm{NN}(x; \overline{C} \cdot \theta) = \prod_{l=1}^{L}(C^{(l)})^H \cdot \mathrm{NN}(x; \theta), \tag{143}$$

$$\partial_{(\overline{C}\theta)_p} \mathrm{NN}(x; \overline{C} \cdot \theta) = D_{p,H} \cdot \partial_{\theta_p} \mathrm{NN}(x; \theta), \tag{144}$$

*where $D_{p,H}$ is such that* $D_{p,H} \leq \prod_{l=1}^{l_p} (C^{(l)})^H$.

**Lemma 6.** *Let* $\mathrm{NN}(x; \theta)$ *be a fully connected network as in equation 139-equation 140. Assume that the activation $\sigma$ is $H$-weakly homogeneous (as defined in Def. 5), with $H \geq 1$. Then, $\mathrm{NN}(x; \theta)$ is $H$-SPH.*

The proof of Lemma 6 is in Appendix B.6.

If we optimize over the Correlation Loss, i.e. $L_{\mathrm{corr}}(y, \hat{y}) := -y\hat{y}$, then the gradients of interest are given by:

$$\Gamma_f(\theta) = -\mathbb{E}_x\left[f(x) \cdot \nabla_\theta \mathrm{NN}(x; \theta)\right]; \tag{145}$$

$$\Gamma_r(\theta) = 0. \tag{146}$$

Thus,

$$\mathbb{E}_{\psi^t} \|\Gamma_f(\psi^t) - \Gamma_r(\psi^t)\|_2^2 = \sum_{p=1}^{P} \mathbb{E}_{\psi^t}\mathbb{E}_x\left[\partial_{\psi_p^t} \mathrm{NN}(x; \psi^t) \cdot f(x)\right]^2$$

Let $\overline{C}$ be a $P \times P$ matrix such that $\overline{C}_{pp} = \sqrt{1 + \frac{t\gamma^2\tau^2}{v_{l_p}^2}}$, where $l_p$ is the layer of $\theta_p^0$. One can verify that the $\overline{C}$-rescaling of $\theta^0$ has the same distribution as $\psi^t$. We can thus rewrite each term in the sum above as:

$$\mathbb{E}_{\psi^t}\mathbb{E}_x\left[\partial_{\psi_p^t} \mathrm{NN}(x; \psi^t) \cdot f(x)\right]^2 = \mathbb{E}_{\overline{C}\theta^0}\mathbb{E}_x\left[\partial_{(\overline{C}\theta^0)_p} \mathrm{NN}(x; \overline{C}\theta^0) \cdot f(x)\right]^2$$

$$\overset{(a)}{=} D_{p,H}^2 \cdot \mathbb{E}_{\theta^0}\mathbb{E}_x\left[\partial_{\theta_p^0} \mathrm{NN}(x; \theta^0) \cdot f(x)\right]^2$$

where in $(a)$ we used Lemma 6. Thus,

$$\mathbb{E}_{\psi^t} \|\Gamma_f(\psi^t) - \Gamma_r(\psi^t)\|_2^2 = \mathbb{E}_{\theta^0} \sum_{p=1}^{P} D_{p,H}^2 \mathbb{E}_x\left[\partial_{\theta_p^0} \mathrm{NN}(x; \theta^0) \cdot f(x)\right]^2$$

$$\overset{(a)}{\leq} K \cdot \mathbb{E}_{\theta^0} \|G_f(\theta^0)\|_2^2,$$

where $K = \prod_{l=1}^{L} \left(1 + \frac{t\gamma^2\tau^2}{v_l^2}\right)^H$, and where in $(a)$ we used that $|D_{p,H}| \leq C_{p,H}$.

### B.6 PROOF OF LEMMA 6

We proceed by induction on the network depth. As a base case, we consider a 2-layer network. Let us write explicitly the gradients of the network.

$$\nabla_{W_i^{(2)}} \text{NN}(x;\theta) = \sigma(x_i^{(1)}(\theta)), \tag{147}$$

$$\nabla_{W_{ij}^{(1)}} \text{NN}(x;\theta) = W_i^{(2)} \sigma'(x_i^{(1)}(\theta)) x_j, \tag{148}$$

$$\nabla_{b_i^{(1)}} \text{NN}(x;\theta) = W_i^{(2)} \sigma'(x_i^{(1)}(\theta)). \tag{149}$$

Notice that the weak homogeneity assumption on the activation $\sigma$ (Def. 5), we have for $l \in \{1,2\}$:

$$x_i^{(l)}(\overline{C} \cdot \theta) = \prod_{h=1}^{l} (C^{(h)})^H \cdot x_i^{(l)}(\theta), \tag{150}$$

thus equation 143 holds. Moreover,

$$\partial_{W_i^{(2)}} \text{NN}(x;\overline{C} \cdot \theta) = (C^{(1)})^H \partial_{W_i^{(2)}} \text{NN}(x;\theta), \tag{151}$$

$$\partial_{W_{ij}^{(1)}} \text{NN}(x;\overline{C} \cdot \theta) = (C^{(2)})^H \partial_{W_{ij}^{(1)}} \text{NN}(x;\theta), \tag{152}$$

$$\partial_{b_i^{(1)}} \text{NN}(x;\overline{C} \cdot \theta) = (C^{(2)})^H \partial_{b_i^{(1)}} \text{NN}(x;\theta). \tag{153}$$

Therefore, for any parameter $\theta_p$, $p \in [P]$,

$$\partial_{\theta_p} \text{NN}(x;\overline{C} \cdot \theta) = D_{p,H} \partial_{\theta_p} \text{NN}(x;\theta), \tag{154}$$

with $1 < D_{p,H} \leq \max\{(C^{(1)})^H, (C^{(2)})^H\} \leq \prod_{l=1}^{2} (C^{(l)})^H$.

For the induction step, assume that for a network of depth $L-1$, for all parameters $\theta_p$,

$$\partial_{\theta_p} \text{NN}(x;\overline{C}(\theta)) = D_{p,H} \cdot \partial_{\theta_p} \text{NN}(x;\theta), \tag{155}$$

with $1 < D_{p,H} \leq \prod_{l=1}^{L-1} (C^{(l)})^H$. Let us consider a neural network of depth $L$, and let us write the gradients,

$$\partial_{W_i^{(L)}} \text{NN}(x;\theta) = \sigma(x_i^{(L-1)}(\theta)), \tag{156}$$

$$\partial_{W_{ij}^{(l)}} \text{NN}(x;\theta) = \sum_{k=1}^{N_{L-1}} W_k^{(L)} \sigma'(x_k^{(L-1)}(\theta)) \cdot \partial_{W_{ij}^{(l)}} x_k^{(L-1)}(\theta), \qquad l = 1, ..., L-1, \tag{157}$$

$$\partial_{b_i^{(1)}} \text{NN}(x;\theta) = \sum_{k=1}^{N_{L-1}} W_k^{(L)} \sigma'(x_k^{(L-1)}(\theta)) \cdot \partial_{b_i^{(1)}} x_k^{(L-1)}(\theta), \tag{158}$$

where $N_{L-1}$ denotes the width of the $(L-1)$-th hidden layer. One can observe that $x_k^{(L-1)}(\theta)$ corresponds to the output of a fully connected network of depth $L-1$, and thus we can use the induction hypothesis for bounding $\partial_{\theta_p} x_k^{(L-1)}(\theta)$, for all parameters $\theta_p$ in the first $L-1$ layers. Thus,

$$\partial_{W_i^{(L)}} \text{NN}(x;\overline{C}(\theta)) = (C^{(L-1)})^H \cdot D_{W_i^{(L)},H} \cdot \partial_{W_i^{(L)}} \text{NN}(x;\theta), \tag{159}$$

$$\partial_{W_{ij}^{(l)}} \text{NN}(x;\overline{C}(\theta)) = \sum_{k=1}^{N_{L-1}} C^{(L)} W_k^{(L)} \sigma'(x_k^{(L-1)}(\theta)) \cdot D_{W_{ij}^{(l)},H} \partial_{W_{ij}^{(l)}} x_k^{(L-1)}(\theta), \tag{160}$$

$$l = 1, ..., L-1, \tag{161}$$

$$\partial_{b_i^{(1)}} \text{NN}(x;\overline{C}(\theta)) = \sum_{k=1}^{N_{L-1}} C^{(L)} W_k^{(L)} \sigma'(x_k^{(L-1)}(\theta)) \cdot D_{b_i^{(1)},H} \partial_{b_i^{(l)}} x_k^{(L-1)}(\theta). \tag{162}$$

Thus, the result follows.

## C  SMALL ALIGNMENT FOR GAUSSIAN INITIALIZATION: PROOFS OF THEOREM 7 AND PROPOSITION 1

In order to establish Proposition 1 and subsequently Theorem 7, we will need two calculations arising from the gradient formulas.

**Definition 9.** *Let $d \in \mathbb{N}$ and $\alpha \geq 0$ and $\beta$ be such that $\alpha + |\beta| \leq 1$. We say that random variables $(k, G_1, G_2)$ are $(d, \alpha, \beta)$-alternating Gaussians if:*

- *$k \sim \mathrm{Bin}(d, 1/2)$.*

- *Conditioned on $k$, the pair $(G_1, G_2)$ are joint centered unit variance Gaussians with covariance $(1 - 2k/d)\alpha + \beta$.*

**Lemma 7.** *For each $\alpha_0 > 0$ there exist $C', C > 0$ such that if $(k, G_1, G_2)$ are $(d, \alpha, \beta)$-alternating Gaussians for $\alpha \geq \alpha_0$, then*

$$\mathbb{E}\Big[(-1)^k \mathbb{1}(G_1 \geq 0)\mathbb{1}(G_2 \geq 0)\Big] \leq C' \exp(-Cd) . \tag{163}$$

**Lemma 8.** *For each $\alpha_0 > 0$ there exist $C', C > 0$ such that if $(k, G_1, G_2)$ are $(d, \alpha, \beta)$-alternating Gaussians for $\alpha \geq \alpha_0$, then*

$$\mathbb{E}\Big[(-1)^k \mathrm{ReLU}(G_1)\mathrm{ReLU}(G_2)\Big] \leq C' \exp(-Cd) . \tag{164}$$

A crucial element of both calculations is the following claim:

**Claim 8.** *Let $d \in \mathbb{N}$. For all $n < d$, for any polynomial $P$ of degree $n$,*

$$\sum_{k=0}^{d}(-1)^k \binom{d}{k} P(k) = 0. \tag{165}$$

*Proof.* We prove the statement by induction on $n$. If $n = 0$, then

$$\sum_{k=0}^{d}(-1)^k \binom{d}{k} = (1 - 1)^d = 0 , \tag{166}$$

and therefore the sum equation 165 indeed vanishes for every constant polynomial. Assume that the claim holds for some $n \geq 0$. By linearity, it is enough that we only prove

$$\sum_{k=0}^{d}(-1)^k \binom{d}{k} k^{n+1} = 0 . \tag{167}$$

To that end, calculate

$$\sum_{k=0}^{d}(-1)^k \binom{d}{k} k^{n+1} = \sum_{k=1}^{d}(-1)^k \binom{d}{k} k \cdot k^n \tag{168}$$

$$\overset{(a)}{=} d\sum_{k=1}^{d}(-1)^k \binom{d-1}{k-1} k^n \tag{169}$$

$$\overset{(b)}{=} -d\sum_{k=0}^{d-1}(-1)^k \binom{d-1}{k} (k+1)^n = 0, \tag{170}$$

where (a) applied $\binom{d}{k} \cdot k = \binom{d-1}{k-1} \cdot d$ and (b) is a change of variables and applying the induction. $\square$

### C.1  PROPOSITION 1 IMPLIES THEOREM 7

Let $0 \leq \lambda^2 \leq \gamma^2 \tau^2 T$. In order to apply Theorem 6 for $A = 0$, we need to check the gradient alignment for initializations $\theta + \lambda H$, where $H \sim \mathcal{N}(0, \mathrm{Id}_P)$. More precisely, that means we have initialization with independent coordinates where

$$w_{ij} \sim \mathcal{N}\left(0, \frac{1}{d} + \lambda^2\right), \, b_i \sim \mathcal{N}\left(0, \sigma^2 + \lambda^2\right), \, v_i \sim \mathcal{N}\left(0, \frac{1}{n} + \lambda^2\right) . \tag{171}$$

Let us normalize by dividing $w$ and $b$ by $\sqrt{1+d\lambda^2}$ and $v$ by $\sqrt{1+n\lambda^2}$. That gives new initialization $\widetilde{\theta}_\lambda = (\widetilde{w}, \widetilde{b}_\lambda, \widetilde{v})$ such that

$$\widetilde{w}_{ij} \sim \mathcal{N}\left(0, \frac{1}{d}\right), \widetilde{b}_{\lambda,i} \sim \mathcal{N}\left(0, \frac{\sigma^2 + \lambda^2}{1+d\lambda^2}\right), \widetilde{v}_i \sim \mathcal{N}\left(0, \frac{1}{n}\right) . \tag{172}$$

In particular, the variance of $\widetilde{b}_{\lambda,i}$ is $\frac{\sigma^2+\lambda^2}{1+d\lambda^2} \le \sigma^2 + \frac{\lambda^2}{1+\lambda^2} \le \sigma^2 + O(1)$. By Proposition 1, we have a uniform bound

$$\mathrm{GAL}_{f_a}(\widetilde{\theta}_\lambda) \le 2C'nd\exp(-Cd) . \tag{173}$$

By homogenity $\mathrm{ReLU}(cx) = c\,\mathrm{ReLU}(x)$ for $c \ge 0$, it is easy to check that

$$\mathrm{GAL}_{f_a}(\theta + \lambda H) \le (1+d\lambda^2)(1+n\lambda^2)\,\mathrm{GAL}_{f_a}(\widetilde{\theta}_\lambda) \le \exp(-\Omega(d)) . \tag{174}$$

The result now follows directly from Theorem 6. □

## C.2 LEMMA 7 AND LEMMA 8 IMPLY PROPOSITION 1

Recall that $\mathrm{GAL}_{f_a} = \mathbb{E}_\theta \big\| \left(\mathbb{E}_x f_a(x)\nabla_\theta \mathrm{NN}(x;\theta)\right)^2 \big\|^2$. We will estimate the expectation of each squared coordinate of this vector by $O(\exp(-Cd))$. Then, equation 9 follows by summing up. Let us first write the neural network gradients for all types of weights $\theta = (w, b, v)$:

$$\nabla_{w_{ij}} \mathrm{NN} = v_i \mathbb{1}(w_i \cdot x + b_i \ge 0)x_j , \tag{175}$$

$$\nabla_{b_i} \mathrm{NN} = v_i \mathbb{1}(w_i \cdot x + b_i \ge 0) , \tag{176}$$

$$\nabla_{v_i} \mathrm{NN} = \mathrm{ReLU}(w_i \cdot x + b_i) . \tag{177}$$

The square of the expected gradient $(\mathbb{E}_x f_a(x)\nabla_{\theta_i} NN(x;\theta)^2$ can be also written as the expectation over two independent input samples $x, x'$. In particular, in the case of $w_{ij}$ from equation 175, we have

$$\mathbb{E}_\theta \left(\mathbb{E}_x f_a(x)\nabla_{w_{ij}} \mathrm{NN}\right)^2 = \mathbb{E}_{x,x'}\left(\prod_{\ell=1}^{d-a} x_\ell x'_\ell\right)\left(\mathbb{E}_{v_i} v_i^2\right) \tag{178}$$

$$\cdot \left(\mathbb{E}_{w_i,b_i}\mathbb{1}(w_i \cdot x + b_i \ge 0)\mathbb{1}(w_i \cdot x' + b_i \ge 0)\right) x_j x'_j . \tag{179}$$

Consider the set $S := \{1, \ldots, d-a\}\triangle\{j\}$, where $\triangle$ denotes the symmetric difference. Abusing notation, let us write $x = (y, z)$ and $x' = (y', z')$ where $y, y'$ containt the coordinates in $S$ and $z, z'$ the coordinates from $[d] \setminus S$. Let $k$ be the Hamming distance $k := d_H(y, y')$. Note that the distribution of $k$ is binomial $k \sim \mathrm{Bin}(|S|, 1/2)$. Then, continuing from equation 179,

$$\mathbb{E}_\theta \left(\mathbb{E}_x f_a(x)\nabla_{w_{ij}} \mathrm{NN}\right)^2 = \frac{1}{n}\mathbb{E}_{z,z',k}\left[(-1)^k \mathbb{E}_{w_i}\left[\mathbb{1}(w_i \cdot x + b_i \ge 0)\mathbb{1}(w_i \cdot x' + b_i \ge 0)\right]\right] . \tag{180}$$

Fix some values of $z, z'$ and $k$. Let $G_1 := w_i \cdot x + b_i$ and $G_2 := w_i \cdot x' + b_i$. Notice that, conditionally on $k, z, z'$, random variables $G_1$ and $G_2$ are joint centered Gaussian with $\mathrm{Var}\,G_1 = \mathrm{Var}\,G_2 = 1+\sigma^2$ and

$$\mathrm{Cov}[G_1, G_2] = \frac{1}{d}\left(d - 2k - 2d_H(z, z')\right) + \sigma^2 . \tag{181}$$

Let $\widetilde{G}_i := G_i/\sqrt{1+\sigma^2}$ for $i = 1, 2$. Then, $\widetilde{G}_1$ and $\widetilde{G}_2$ are two joint centered unit variancce Gaussians with correlation

$$\mathrm{Cov}[\widetilde{G}_1, \widetilde{G}_2] = \frac{1}{d(1+\sigma^2)}\left(d - 2k - 2d_H(z, z')\right) + \frac{\sigma^2}{1+\sigma^2} \tag{182}$$

$$= \left(1 - \frac{2k}{|S|}\right)\frac{|S|}{d(1+\sigma^2)} + \frac{d - |S| - 2d_H(z, z') + d\sigma^2}{d(1+\sigma^2)} . \tag{183}$$

Therefore, conditioned on $z$ and $z'$, random variables $(k, G_1, G_2)$ are $(d, \alpha, \beta)$-alternating Gaussians for $\alpha = \frac{|S|}{d(1+\sigma^2)} \ge \frac{1}{3(1+\sigma_0^2)} > 0$. It is also easy to check that $\alpha + |\beta| \le \frac{|S|+d-|S|+d\sigma^2}{d(1+\sigma^2)} = 1$. By Lemma 7, for some uniform constant $C > 0$ it holds

$$\mathbb{E}_{k,G_1,G_2}\left[(-1)^k\mathbb{1}(G_1 \ge 0)\mathbb{1}(G_2 \ge 0)\right] = \mathbb{E}_{k,\widetilde{G}_1,\widetilde{G}_2}\left[(-1)^k\mathbb{1}(\widetilde{G}_1 \ge 0)\mathbb{1}(\widetilde{G}_2 \ge 0)\right] \tag{184}$$

$$\le C'\exp(-Cd) . \tag{185}$$

Plugging this into equation 179 and equation 180, we get the desired bound. The case of the hidden layer bias $b_i$ proceeds by the same argument with $S := \{1, \dots, d - a\}$.

Finally, in case of $v_i$ we set $S := \{1, \dots, d - a\}$ and proceed with a similar calculation

$$\mathbb{E}_\theta \left( \mathbb{E}_x f_a(x) \nabla_{v_i} \mathrm{NN} \right)^2 = (1 + \sigma^2) \mathbb{E}_{z,z',k} \left[ (-1)^k \mathbb{E}_{\widetilde{G}_1, \widetilde{G}_2} [\mathrm{ReLU}(\widetilde{G}_1) \, \mathrm{ReLU}(\widetilde{G}_2)] \right] \tag{186}$$

$$\leq (1 + \sigma_0^2) C' \exp(-Cd) \leq C'' \exp(-Cd) \,, \tag{187}$$

where in the last line we applied Lemma 8. $\qquad \square$

## C.3 PROOF OF LEMMA 7

It is well-known (see, e.g., Chapter 11 in O'Donnell (2014)), that for two $\rho$-correlated unit variance centered joint Gaussians it holds $\mathbb{E}[\mathbb{1}(G_1 \geq 0)\mathbb{1}(G_2 \geq 0)] = f(\rho)$ where $f(x) = \frac{1}{2} - \frac{1}{2\pi}\arccos(x)$. By definition of $(k, G_1, G_2)$, conditioned on $k$, random variables $G_1$ and $G_2$ have correlation $\rho = \rho(k) = \left(1 - \frac{2k}{d}\right)\alpha + \beta$.

Hence,

$$\left| \mathbb{E}(-1)^k \mathbb{1}(G_1 \geq 0)\mathbb{1}(G_2 \geq 0) \right| = \left| \mathbb{E}_k (-1)^k f(\rho(k)) \right| \tag{188}$$

$$\leq \mathbb{P}(|k - d/2| \geq d/4) \cdot \sup_{x \in [-1,1]} |f(x)| + \left| \frac{1}{2^d} \sum_{k=\lceil d/4 \rceil}^{\lfloor 3d/4 \rfloor} (-1)^k \binom{d}{k} f(\rho) \right| \tag{189}$$

$$\overset{(a)}{\leq} 2\exp(-d/10) + \left| \frac{1}{2^d} \sum_{k=\lceil d/4 \rceil}^{\lfloor 3d/4 \rfloor} (-1)^k \binom{d}{k} f(\rho) \right| \,, \tag{190}$$

where $(a)$ follows by Hoeffding's inequality.

It remains to bound the last term in equation 190. Consider the Taylor expansion of $f$:

$$f(x) = \frac{1}{2} - \frac{1}{2\pi} \left[ \frac{\pi}{2} - \sum_{n=0}^{\infty} \frac{(2n)!}{4^n (n!)^2 (2n + 1)} x^{2n+1} \right] \tag{191}$$

$$= \frac{1}{4} + \frac{1}{2\pi} \sum_{n=0}^{\infty} \frac{(2n)!}{4^n (n!)^2 (2n + 1)} x^{2n+1} \tag{192}$$

$$= \frac{1}{4} + \frac{1}{2\pi} \sum_{n=0}^{\infty} \frac{\binom{2n}{n}}{4^n (2n + 1)} x^{2n+1} \tag{193}$$

$$= \frac{1}{4} + \sum_{2n+1 < d} a_n x^{2n+1} + \sum_{2n+1 \geq d} a_n x^{2n+1} \,, \tag{194}$$

where $a_n := \frac{\binom{2n}{n}}{2\pi 4^n (2n+1)}$. For future reference let us note that $0 \leq a_n \leq 1$ for every $n$. So the second part of the RHS of equation 190 is upper bounded by:

$$\underbrace{\left| \frac{1}{2^d} \sum_{k=\lceil d/4 \rceil}^{\lfloor 3d/4 \rfloor} (-1)^k \binom{d}{k} \left( \frac{1}{4} + \sum_{2n+1 < d} a_n \rho^{2n+1} \right) \right|}_{:=T_1} \tag{195}$$

$$+ \underbrace{\left| \frac{1}{2^d} \sum_{k=\lceil d/4 \rceil}^{\lfloor 3d/4 \rfloor} (-1)^k \binom{d}{k} \sum_{2n+1 \geq d} a_n \rho^{2n+1} \right|}_{:=T_2} \tag{196}$$

We are going to show that $|T_1| \leq 2\exp\left(-d/10\right)$ and $|T_2| \leq \frac{2}{\alpha_0}(1 - \alpha_0/2)^d$. These two bounds together with equation 190 imply the theorem statement.

Let us start with $T_2$. In the sum in equation 196 we have $d/4 \leq k \leq 3d/4$, and we can check that

$$|\rho| = \left| \left(1 - \frac{2k}{d}\right)\alpha + \beta \right| \leq \frac{1}{2}\alpha + |\beta| \leq 1 - \frac{\alpha_0}{2}. \tag{197}$$

Therefore,

$$|T_2| = \left| \frac{1}{2^d} \sum_{k=\lceil d/4 \rceil}^{\lfloor 3d/4 \rfloor} (-1)^k \binom{d}{k} \sum_{2n+1 \geq d} a_n \rho^{2n+1} \right| \tag{198}$$

$$\leq \frac{1}{2^d} \cdot \sum_{k=\lceil d/4 \rceil}^{\lfloor 3d/4 \rfloor} \binom{d}{k} \sum_{2n+1 \geq d} a_n \left(1 - \frac{\alpha_0}{2}\right)^{2n+1} \tag{199}$$

$$\leq \sum_{2n+1 \geq d} \left(1 - \frac{\alpha_0}{2}\right)^{2n+1} \leq \frac{2}{\alpha_0} \left(1 - \frac{\alpha_0}{2}\right)^d. \tag{200}$$

For $T_1$, we follow two steps. First,

$$|T_1| \leq \sum_{2n+1 < d} \left| \frac{1}{2^d} \sum_{k=\lceil d/4 \rceil}^{\lfloor 3d/4 \rfloor} (-1)^k \binom{d}{k} \rho^{2n+1} \right|. \tag{201}$$

Applying Claim 8 (for this note that $\rho$ is a linear function of $k$, and therefore $\rho^{2n+1}$ is a polynomial in $k$ of degree $2n+1$):

$$\left| \frac{1}{2^d} \sum_{k=\lceil d/4 \rceil}^{\lfloor 3d/4 \rfloor} (-1)^k \binom{d}{k} \rho^{2n+1} \right| \tag{202}$$

$$\leq \left| \frac{1}{2^d} \sum_{k=0}^{d} (-1)^k \binom{d}{k} \rho^{2n+1} \right| + \left| \frac{1}{2^d} \sum_{k:|k-d/2| \geq d/4} (-1)^k \binom{d}{k} \rho^{2n+1} \right| \tag{203}$$

$$\leq \sum_{k:|k-d/2| \geq d/4} \binom{d}{k} 2^{-d} \tag{204}$$

$$= \mathbb{P}(|k - d/2| \geq d/4) \tag{205}$$

$$\leq 2 \exp(-d/10). \tag{206}$$

Finally, we substitute into equation 201 and conclude $|T_1| \leq 2\exp(-d/10)$. □

## C.4 PROOF OF LEMMA 8

In this proof we will use the probabilist's Hermite polynomials $H_k(x) = \frac{(-1)^k}{\varphi(x)} \frac{\mathrm{d}^k}{\mathrm{d}x^k} \varphi(x)$, where $\varphi(x) = \frac{1}{\sqrt{2\pi}} \exp(-x^2/2)$ is the standard Gaussian density, see, e.g., Lebedev (1972) for more details. One property that we will need is that for two centered $\rho$-correlated unit variance joint Gaussians $G_1, G_2$ it holds

$$\mathbb{E}H_m(G_1)H_n(G_2) = \begin{cases} m! & \text{if } m = n, \\ 0 & \text{otherwise.} \end{cases} \tag{207}$$

We will also make use of the ReLU Hermite expansion, see, e.g., Proposition 6 in Abbe et al. (2022c). That is, $\mathrm{ReLU}(x) = \frac{1}{\sqrt{2\pi}} + \frac{1}{2}x + \sum_{m=1}^{\infty} a_m H_{2m}(x)$ for $a_m := \frac{(-1)^{m+1}}{\sqrt{2\pi}2^m(2m-1)m!}$ and consequently, applying equation 207,

$$\mathbb{E}\,\mathrm{ReLU}(G_1)\,\mathrm{ReLU}(G_2) = \frac{1}{2\pi} + \frac{1}{4}\rho + \sum_{m=1}^{\infty} a_m^2 (2m)! \rho^{2m}. \tag{208}$$

Furthermore, in any case we always have

$$\mathbb{E}\,\mathrm{ReLU}(G_1)\,\mathrm{ReLU}(G_2) \leq \mathbb{E}\,\mathrm{ReLU}^2(G_1) = \frac{1}{2}. \tag{209}$$

As in Lemma 7, conditioned on $k$, random variables $G_1, G_2$ are centered unit variance Gaussians with correlation $\rho = \rho(k) = \left(1 - \frac{2k}{d}\right)\alpha + \beta$. In particular, by equation 197, as long as $d/4 \le k \le 3d/4$, then $|\rho| \le 1 - \frac{\alpha_0}{2}$. Now we estimate, for $d \ge 2$, applying Claim 8 in equation 211 and again in equation 216:

$$\left| \mathbb{E}(-1)^k \operatorname{ReLU}(G_1) \operatorname{ReLU}(G_2) \right| \tag{210}$$

$$= \left| \mathbb{E}(-1)^k \left( \operatorname{ReLU}(G_1) \operatorname{ReLU}(G_2) - \frac{1}{2\pi} - \frac{1}{4}\rho \right) \right| \tag{211}$$

$$\le \Pr[|k - d/2| > d/4] + \left| \sum_{k=\lceil d/4 \rceil}^{\lfloor 3d/4 \rfloor} (-1)^k \binom{d}{k} \sum_{m=1}^{\infty} a_m^2 (2m)! \rho^{2m} \right| \tag{212}$$

$$\le 2\exp(-d/10) + \sum_{2m<d} a_m^2 (2m!) \left| \sum_{k=\lceil d/4 \rceil}^{\lfloor 3d/4 \rfloor} (-1)^k \binom{d}{k} \rho^{2m} \right| \tag{213}$$

$$+ \sum_{2m\ge d} a_m^2 (2m!) \left| \sum_{k=\lceil d/4 \rceil}^{\lfloor 3d/4 \rfloor} (-1)^k \binom{d}{k} \rho^{2m} \right| \tag{214}$$

$$\le 2\exp(-d/10) + \sum_{2m<d} \left| \sum_{k=\lceil d/4 \rceil}^{\lfloor 3d/4 \rfloor} (-1)^k \binom{d}{k} \rho^{2m} \right| + \sum_{2m\ge d} \left(1 - \frac{\alpha_0}{2}\right)^{2m} \tag{215}$$

$$\le C'\exp(-Cd) + \sum_{2m<d} \left( \left| \sum_{k=0}^{d} (-1)^k \binom{d}{k} \rho^{2m} \right| + \Pr[|k - d/2| > d/4] \right) \tag{216}$$

$$\le C'\exp(-Cd) . \tag{217}$$

$\square$

# D SMALL ALIGNMENT FOR PERTURBED INITIALIZATION: PROOF OF THEOREM 8

## D.1 PROPOSITION 2 IMPLIES THEOREM 8

Take $\sigma_0, C$ and $C'$ from Proposition 2. Let the setting be as in Theorem 8 i.e. $\theta = (w, v)$, with i.i.d. initialization $w = \frac{1}{\sqrt{d}}(r + g)$ where $r \sim \operatorname{Rad}(1/2)$, $g \sim \mathcal{N}\left(0, \sigma^2\right)$ and $v \sim \mathcal{N}\left(0, \frac{1}{n}\operatorname{Id}_n\right)$. Let's consider any $\sigma = \sigma(d) \ge \sigma_0$.

As before, we would like to apply Theorem 6. Let $0 \le \lambda^2 \le \gamma^2\tau^2 T$. Let us check the gradient alignment for $\theta + \lambda H$, where $H \sim \mathcal{N}\left(0, \operatorname{Id}_P\right)$. So we consider the weights with independent coordinates where

$$w_{\lambda,ij} = \frac{1}{\sqrt{d}}(r_{ij} + g_{ij}) + \lambda h_{ij}, v_{\lambda,i} \sim \mathcal{N}\left(0, \frac{1}{n} + \lambda^2\right) , \tag{218}$$

where $g_{ij} \sim \mathcal{N}\left(0, \sigma^2\right), r_{ij} \sim \operatorname{Rad}(1/2)$ and $h_{ij} \sim \mathcal{N}\left(0, 1\right)$. Let us rewrite $w_{\lambda,ij}$ as

$$w_{\lambda,ij} = \frac{1}{\sqrt{d}}(r_{ij} + \tilde{g}_{\lambda,ij}) \text{ with } \tilde{g}_{\lambda,ij} \sim \mathcal{N}\left(0, \sigma^2 + \lambda^2 d\right) . \tag{219}$$

Also let's normalize by dividing $v_\lambda$ by $\sqrt{1 + n\lambda^2}$. That gives a new initialization $\widetilde{\theta}_\lambda = (w_\lambda, \widetilde{v})$ such that $\widetilde{v}_i \sim \mathcal{N}\left(0, \frac{1}{n}\right)$. Since we have $\sqrt{\sigma^2 + \lambda^2 d} \ge \sigma \ge \sigma_0$, then by Proposition 2

$$\operatorname{GAL}_f(\widetilde{\theta}_\lambda) \le PC'\exp(-Cd) . \tag{220}$$

Finally, by gradient formulas and homogenity of ReLU, we have

$$\operatorname{GAL}_f(\theta + \lambda H) \le (1 + n\lambda^2)\operatorname{GAL}_f(\widetilde{\theta}_\lambda) \le \exp(-\Omega(d)) . \tag{221}$$

Therefore the result follows by Theorem 6. □

Let $g$ and $r$ be two i.i.d. vectors with $n$ coordinates such that on each coordinate $g_i \sim \mathcal{N}\left(0, \frac{1}{d}\right)$ and $r_i \sim \mathrm{Rad}(1/2)$. Let's define two values expressing the gradient alignments for weights in the hidden and output layers, respectively. For $\mu \geq 0$ and $d \in \mathbb{N}$:

$$\mathrm{GAL}_{\mathrm{hid}}(\mu, d) := \mathbb{E}_{g,r}\left[\left(\mathbb{E}_x\left[\prod_{i=1}^{d-1} x_i \mathbb{1}[(g + \mu r) \cdot x \geq 0]\right]\right)^2\right]. \tag{222}$$

$$\mathrm{GAL}_{\mathrm{out}}(\mu, d) := \mathbb{E}_{g,r}\left[\left(\mathbb{E}_x\left[\prod_{i=1}^{d} x_i \mathrm{ReLU}((g + \mu r) \cdot x)\right]\right)^2\right]. \tag{223}$$

**Lemma 9.** *There exists some $\alpha_0, C > 0$ and $D_0$ such that, for $d \geq D_0$, and $\mu \leq \frac{\alpha_0}{\sqrt{d}}$, it holds $\mathrm{GAL}_{\mathrm{hid}}(\mu, d) \leq \exp(-Cd)$.*

**Lemma 10.** *There exists some $\alpha_0, C > 0$ and $D_0$ such that, for $d \geq D_0$, and $\mu \leq \frac{\alpha_0}{\sqrt{d}}$, it holds $\mathrm{GAL}_{\mathrm{out}}(\mu, d) \leq \exp(-Cd)$.*

### D.2 LEMMA 9 AND LEMMA 10 IMPLY PROPOSITION 2

Let the setting be as in Theorem 8 i.e. $w_i = \frac{1}{\sqrt{d}}(r_i + g_i), g_i \sim \mathcal{N}\left(0, \sigma^2\right)$ and $r_i \sim \mathrm{Rad}(1/2)$. The gradient formulas for full parity:

$$\nabla_{w_{ij}} \mathrm{NN} = v_i \mathbb{1}(w_i \cdot x \geq 0) x_j , \tag{224}$$

$$\nabla_{v_i} \mathrm{NN} = \mathrm{ReLU}(w_i \cdot x) . \tag{225}$$

As the gradient has $P = nd + n$ coordinates, it is enough to show the $C' \exp(-Cd)$ bound on every coordinate of the gradient. Let us start with hidden weight coordinates. By symmetry, we can suppose w.l.o.g. that $j = d$. The alignment of hidden layer is given by:

$$\mathbb{E}_\theta \left(\mathbb{E}_x f(x) \nabla_{w_{ij}} \mathrm{NN}\right)^2 = \mathbb{E}_\theta \left(\mathbb{E}_x f(x) v_i \mathbb{1}(w_i \cdot x \geq 0) x_j\right)^2 \tag{226}$$

$$= \mathbb{E}_{v_i}[v_i^2] \mathbb{E}_{g_i,r_i}\left[\left(\mathbb{E}_x \prod_{\ell=1}^{d-1} x_\ell \mathbb{1}\left(\frac{1}{\sqrt{d}}(r_i + g_i) \cdot x \geq 0\right)\right)^2\right] \tag{227}$$

$$= \frac{1}{n} \mathbb{E}_{g_i,r_i}\left[\left(\mathbb{E}_x \prod_{\ell=1}^{d-1} x_\ell \mathbb{1}\left(\frac{1}{\sqrt{d}}(r_i + g_i) \cdot x \geq 0\right)\right)^2\right] \tag{228}$$

$$= \frac{1}{n} \mathbb{E}_{\tilde{g}_i,r_i}\left[\left(\mathbb{E}_x \prod_{\ell=1}^{d-1} x_\ell \mathbb{1}\left((\frac{1}{\sigma\sqrt{d}} r_i + \tilde{g}_i) \cdot x \geq 0\right)\right)^2\right] , \tag{229}$$

where $\tilde{g}_i \sim \mathcal{N}\left(0, \frac{1}{d}\right)$. Therefore, by Lemma 9,

$$\mathbb{E}_\theta \left(\mathbb{E}_x f(x) \nabla_{w_{ij}} \mathrm{NN}\right)^2 = \frac{1}{n} \mathrm{GAL}_{\mathrm{hid}}\left(\frac{1}{\sigma\sqrt{d}}, d\right) \leq C' \exp(-Cd) \tag{230}$$

where the constant $C'$ compensates for the fact that Lemma 9 holds for $d$ large enough.

Similarly, for the alignments of output layer weights:

$$\mathbb{E}_\theta \left(\mathbb{E}_x f(x) \nabla_{v_i} \mathrm{NN}\right)^2 = \mathbb{E}_{g_i,r_i}\left[\left(\mathbb{E}_x \prod_{\ell=1}^{d} x_\ell \mathrm{ReLU}\left(\frac{1}{\sqrt{d}}(r_i + g_i) \cdot x\right)\right)^2\right] \tag{231}$$

$$= \sigma^2 \mathbb{E}_{\tilde{g}_i,r_i}\left[\left(\mathbb{E}_x \prod_{\ell=1}^{d} x_\ell \mathrm{ReLU}\left((\frac{1}{\sigma\sqrt{d}} r_i + \tilde{g}_i) \cdot x\right)\right)^2\right] \tag{232}$$

$$= \sigma^2 \mathrm{GAL}_{\mathrm{out}}\left(\frac{1}{\sigma\sqrt{d}}, d\right) \tag{233}$$

### D.3 CORRELATED GAUSSIAN EXPECTATIONS

We give a general formula for expectation of functions of correlated Gaussians. We will then apply this formula to the cases of step function and ReLU:

**Lemma 11.** *Let $(c_k)_k$ and $(d_k)_k$ be two sequences of power series coefficients with infinite radius of convergence. Let $f(x) := \sum_{k=0}^{\infty} c_k H_k(x)$ and $F(x) := \sum_{k=0}^{\infty} c_k x^k$. Similarly, let $g(x) := \sum_{k=0}^{\infty} d_k H_k(x)$ and $G(x) := \sum_{k=0}^{\infty} d_k x^k$. Then, for every $a, b \in \mathbb{R}$ and $\rho$-correlated joint standard Gaussians $Z, Z'$:*

$$\mathbb{E}f(a+Z)g(b+Z') = \sum_{k=0}^{\infty} \frac{F^{(k)}(a)G^{(k)}(b)}{k!}\rho^k , \qquad (234)$$

*where $F^{(k)}$ denotes the $k$-th derivative of $F$.*

*Proof.* Applying the Hermite polynomial identity $H_m(a+b) = \sum_{k=0}^{m} \binom{m}{k} a^{m-k} H_k(b)$:

$$f(a+z) = \sum_{m=0}^{\infty} c_m H_m(a+z) = \sum_{m=0}^{\infty} c_m \sum_{k=0}^{m} \binom{m}{k} a^{m-k} H_k(z) \qquad (235)$$

$$= \sum_{k=0}^{\infty} \frac{1}{k!} H_k(z) \sum_{m=k}^{\infty} c_m \left( \prod_{i=0}^{k-1} m-i \right) a^{m-k} = \sum_{k=0}^{\infty} \frac{F^{(k)}(a)}{k!} H_k(z) \qquad (236)$$

Taking expectation and using $\mathbb{E}H_k(Z)H_{k'}(Z') = \mathbb{1}_{k=k'} k! \rho^k$:

$$\mathbb{E}f(a+Z)g(b+Z') = \sum_{k=0}^{\infty} \frac{F^{(k)}(a)G^{(k)}(b)}{k!}\rho^k . \qquad \square$$

Applying Lemma 11 to the case of the step function, we get the two dimensional case of the "tetrachoric series" Harris & Soms (1980).

**Claim 9.** *Using the notation above, $F(x) = (f * \phi)(x) = \mathbb{E}[f(x+Z)]$.*

*Proof.* From the convolution property $(f * g)' = f' * g$ and identity $\phi^{(k)} = (-1)^k H_k \phi$:

$$(f * \phi)^{(k)}(0) = (f * \phi^{(k)})(0) = (-1)^k (f * (H_k \phi))(0) = \int f(x) H_k(x) \phi(x) \mathrm{d}x \qquad (237)$$

$$= c_k k! = F^{(k)}(0) . \qquad (238)$$

Since the power series coefficients are equal for every $k$, the claim follows. $\square$

**Corollary 6.** $\mathbb{E}\left[\mathbb{1}_{a+Z\geq 0}\mathbb{1}_{b+Z'\geq 0}\right] = \Phi(a)\Phi(b) + \phi(a)\phi(b) \sum_{k=0}^{\infty} H_k(a)H_k(b)\frac{1}{(k+1)!}\rho^{k+1}$.

*Proof.* Using Claim 9 for $f(x) = \mathbb{1}_{x\geq 0}$, we get that

$$F(x) = \mathbb{E}[\mathbb{1}_{x+Z\geq 0}] = \Phi(x) . \qquad (239)$$

The result follows by applying Lemma 11 and $\Phi^{(k)} = \phi^{(k-1)} = (-1)^{k-1} H_{k-1}\phi$. $\square$

**Corollary 7.** *Let $R(x) = x\Phi(x) + \phi(x)$. Then,*

$$\mathbb{E}\left[\mathrm{ReLU}(a+Z)\,\mathrm{ReLU}(b+Z')\right] = R(a)R(b) + \Phi(a)\Phi(b)\rho + \phi(a)\phi(b) \sum_{k=0}^{\infty} \frac{H_k(a)H_k(b)}{(k+2)!}\rho^{k+2} . \qquad (240)$$

*Proof.* Applying Claim 9 for $f = \mathrm{ReLU}$ we get

$$F(x) = \mathbb{E}\,\mathrm{ReLU}(x+Z) = \int (x+y)\mathbb{1}_{x+y\geq 0}\phi(y)\mathrm{d}y = \int_{-x}^{\infty} (x+y)\phi(y)\mathrm{d}y \qquad (241)$$

$$= x\Phi(x) + \int_{-x}^{\infty} -\phi'(y)\mathrm{d}y = x\Phi(x) + \phi(x) = R(x) . \qquad (242)$$

Again the result follows by Lemma 11 and observing that $R'(x) = \Phi(x) + x\phi(x) + \phi'(x) = \Phi(x)$. $\square$

### D.4 PROOF OF LEMMA 9

Let us write $\mu = \alpha/\sqrt{d}$, so that by assumption $\alpha \leq \alpha_0$. We have,

$$\mathrm{GAL}_{\mathrm{hid}} = \mathbb{E}_{g,x,x',r}\left[\prod_i x_i x_i' \mathbb{1}[g \cdot x + \mu r \cdot x, g \cdot x' + \mu r \cdot x' \geq 0]\right] \tag{243}$$

$$= \mathbb{E}_{x,x',r}\left[\prod_i x_i x_i' \Pr_g[g \cdot x + \mu r \cdot x, g \cdot x' + \mu r \cdot x' \geq 0]\right] =: \mathbb{E}_{x,x',r}F(x,x',r) \,. \tag{244}$$

As for every $x, x', r, s \in \{-1,1\}^d$ we have $F(x,x',r) = F(x \odot s, x' \odot s, r \odot s)$ (where $\odot$ denotes the Hadamard product), it follows

$$\mathbb{E}_{x,x',r}F(x,x',r) = \mathbb{E}_{x,x'}F(x,x',1^d) \,, \tag{245}$$

so we can rewrite

$$\mathrm{GAL}_{\mathrm{hid}} = \mathbb{E}_{x,x'}\left[\prod_i x_i x_i' \Pr_g[g \cdot x + \mu \cdot x, g \cdot x' + \mu \cdot x' \geq 0]\right] = \mathbb{E}_{x,x'}F(x,x',1^d) \,.$$

Fix $x$ and assume w.l.o.g. that $x = (1^{d-d'}, -1^{d'})$ for some $0 \leq d' \leq d$. Furthermore, divide $x' = (y,z)$ such that $y \in \{-1,1\}^{d-d'}$ and $z \in \{-1,1\}^{d'}$. Assume that $d' \geq d/2$ and fix $y$. (If $d' < d/2$ we exchange the roles of $y$ and $z$ and proceed with an entirely symmetric argument.) Let $G(x,y,z) = F(x,(y,z),1^d)$. We want to analyze $\mathbb{E}_z G(x,y,z)$ so that the bound on $\mathbb{E}_{x,x'}F(x,x',1^d) = \mathbb{E}_{x,y,z}G(x,y,z)$ will follow by averaging. Let $\rho = \frac{1}{d}x \cdot x'$ and $k$ be the number of $-1$ entries in $z$. Note that we have $\rho = \frac{1 \cdot y + 2k - d'}{d}$. Continuing:

$$|\mathbb{E}_z G(x,y,z)| = \left|(-1)^{d'} \prod_{i=1}^{d-d'} y_i \mathbb{E}_z\left[(-1)^k \Pr_g[g \cdot x + \mu \cdot x, g \cdot x' + \mu \cdot x' \geq 0]\right]\right| \tag{246}$$

$$= \left|\mathbb{E}_k\left[(-1)^k \Lambda_\rho(\mu(d-2d'), \mu(1 \cdot y + d' - 2k))\right]\right| \tag{247}$$

$$= \left|\mathbb{E}_k\left[(-1)^k \Lambda_\rho(\mu(d-2d'), -\mu(d\rho - 2 \cdot y))\right]\right| \,, \tag{248}$$

where $\Lambda_\rho(a,b) = \Pr_{g,g'}[g+a, g'+b \geq 0] = \Pr_{g,g'}[g \leq a, g' \leq b]$, where $g, g'$ are two standard $\rho$-correlated joint Gaussians. Note that the distribution of $k$ is binomial, that is $\Pr[k = k^*] = 2^{-d'}\binom{d'}{k^*}$ for $0 \leq k^* \leq d'$.

In particular, conditioned on $x, y$, the expectation in equation 248 can be written as $|\mathbb{E}_k G(x,y,z)| = |\sum_{k=0}^{d'}(-1)^k\binom{d'}{k}W(\rho)|$ for some function $W$ that depends only on $\rho$. Since $\rho$ is a linear function of $k$, as in the Gaussian case, we will now expand $W$ as a power series and apply Claim 8.

Let

$$A := \mu(d - 2d') \,, B := 2\mu \cdot y \,, C := -\mu d \,, \text{ and } w := B + C\rho \,. \tag{249}$$

Take some $\beta > 0$, where later on we will choose it to be a small enough universal constant (in fact $\beta = 0.005$ will be enough). Let us define two "bad" events: $\mathcal{E}_1$ is $|\rho| \geq 1/2$ and $\mathcal{E}_2$ is $|w| \geq \beta\sqrt{d}$ and let $\mathcal{F}$ be the complement of $\mathcal{E}_1 \cup \mathcal{E}_2$.

First, let us argue that $\Pr[\mathcal{E}_1 \cup \mathcal{E}_2] \le \exp(-c\beta^2 d)$ for some universal $c > 0$ and $d$ large enough:

$$\Pr\left[\mathcal{E}_1 \cup \mathcal{E}_2\right] \le \Pr\left[\mathcal{E}_1\right] + \Pr[\mathcal{E}_2] \tag{250}$$

$$= \Pr\left[|\rho| \ge 1/2\right] + \Pr\left[|w| \ge \beta\sqrt{d}\right] \tag{251}$$

$$\le \Pr\left[\left|\sum_{i=1}^d x_i x_i'\right| \ge \frac{d}{2}\right] + \Pr\left[|B| \ge \beta\frac{\sqrt{d}}{2}\right] + \Pr\left[|C\rho| \ge \beta\frac{\sqrt{d}}{2}\right] \tag{252}$$

$$\le \Pr\left[\left|\sum_{i=1}^d x_i x_i'\right| \ge \frac{d}{2}\right] + \Pr\left[\left|\sum_{i=1}^{d-d'} y_i\right| \ge \beta\frac{d}{4\alpha}\right] + \Pr\left[\left|\sum_{i=1}^d x_i x_i'\right| \ge \frac{\beta d}{2\alpha}\right] \tag{253}$$

$$\le 2\exp(-\frac{d}{8}) + 2\exp\left(-\frac{\beta^2 d^2}{32\alpha^2(d-d')}\right) + 2\exp\left(-\frac{\beta^2 d}{8\alpha^2}\right) \tag{254}$$

$$\le \exp(-c\beta^2 d) , \tag{255}$$

where equation 254 is by Hoeffding's inequality. Using equation 248, our bound becomes

$$\mathrm{GAL_{hid}} \le \mathbb{E}_{x,y}\left|\mathbb{E}_k (-1)^k \Lambda_\rho(A, w)\right| \tag{256}$$

$$\le \Pr[\mathcal{E}_1 \cup \mathcal{E}_2] + \mathbb{E}_{x,y}\left|\mathbb{E}_k(-1)^k\Lambda_\rho(A,w)\mathbb{1}_\mathcal{F}\right| \tag{257}$$

$$\le \exp(-c\beta^2 d) + \mathbb{E}_{x,y}\left|\mathbb{E}_k(-1)^k\Lambda_\rho(A,w)\mathbb{1}_\mathcal{F}\right| . \tag{258}$$

To study the expression $\Lambda_\rho(A, w)$, let us recall some facts about the Gaussians. We have the following expansions:

$$\Phi(z) = \frac{1}{2} + \frac{1}{\sqrt{2\pi}}\sum_{k=0}^\infty \frac{(-1)^k}{2^k k!(2k+1)}z^{2k+1} \tag{259}$$

$$\phi(z) = \frac{1}{\sqrt{2\pi}}\sum_{k=0}^\infty \frac{(-1)^k}{2^k k!}z^{2k} , \tag{260}$$

as well as the tetrachoric series for $\Lambda$ (convergent for every $a, b \in \mathbb{R}$ and $|\rho| < 1$) Harris & Soms (1980), Vasicek (1998):

$$\Lambda_\rho(a, b) = \Phi(a)\Phi(b) + \phi(a)\phi(b)\sum_{k=0}^\infty H_k(a)H_k(b)\frac{1}{(k+1)!}\rho^{k+1} . \tag{261}$$

Substituting into equation 258,

$$\mathrm{GAL_{hid}} \le \exp(-c\beta^2 d) \tag{262}$$

$$+ \mathbb{E}_{x,y}\left|\mathbb{E}_k(-1)^k\mathbb{1}_\mathcal{F}\left(\Phi(A)\Phi(w) + \phi(A)\phi(w)\sum_{\ell=0}^\infty H_\ell(A)H_\ell(w)\frac{\rho^{\ell+1}}{(\ell+1)!}\right)\right| \tag{263}$$

$$\le \exp(-c\beta^2 d) + \mathbb{E}_{x,y}\left|\mathbb{E}_k(-1)^k\mathbb{1}_\mathcal{F}\Phi(A)\Phi(w)\right| \tag{264}$$

$$+ \sum_{\ell=0}^\infty \mathbb{E}_{x,y}\left|\mathbb{E}_k(-1)^k\mathbb{1}_\mathcal{F}\phi(A)\phi(w)H_\ell(A)H_\ell(w)\frac{\rho^{\ell+1}}{(\ell+1)!}\right| \tag{265}$$

$$\le \exp(-c\beta^2 d) + \underbrace{\mathbb{E}_{x,y}\left|\mathbb{E}_k(-1)^k\mathbb{1}_\mathcal{F}\Phi(w)\right|}_{=:T_1} \tag{266}$$

$$+ \underbrace{\sum_{\ell=0}^\infty \mathbb{E}_{x,y}\left|\mathbb{E}_k(-1)^k\mathbb{1}_\mathcal{F}\phi(w)H_\ell(w)\frac{\rho^{\ell+1}}{\sqrt{\ell!}}\right|}_{=:T_2} , \tag{267}$$

where in the last line we used the estimate from (Harris & Soms, 1980, proof of Theorem 2),

$$|H_\ell(A)| \le 2\exp(A^2/4)\sqrt{\ell!} , \tag{268}$$

which implies

$$|\phi(A)H_\ell(A)| \leq \sqrt{\ell!} \, . \tag{269}$$

For tighter estimates on Hermite polynomials, see also Bonan & Clark (1990).

It remains to show that both $T_1$ and $T_2$ are exponentially small.

Let us start with $T_1$. Recall equation 259 and let $a_\ell = \frac{(-1)^\ell}{\sqrt{2\pi}2^\ell \ell!(2\ell+1)}$. Using equation 259 and triangle inequality,

$$T_1 \leq \mathbb{E}_{x,y} \left| \mathbb{E}_k (-1)^k \mathbb{1}_{\mathcal{F}} \left( \frac{1}{2} + \sum_{\ell < d/10} a_\ell w^{2\ell+1} \right) \right| + \sum_{\ell \geq d/10} |a_\ell|(\beta\sqrt{d})^{2\ell+1} \tag{270}$$

$$\leq \mathbb{E}_{x,y} \left| \mathbb{E}_k (-1)^k \left( \frac{1}{2} + \sum_{\ell < d/10} a_\ell w^{2\ell+1} \right) \right| + \mathbb{E}_{x,y} \left| \mathbb{E}_k (-1)^k \mathbb{1}_{\mathcal{E}_1 \cup \mathcal{E}_2} \left( \frac{1}{2} + \sum_{\ell < d/10} a_\ell w^{2\ell+1} \right) \right| \tag{271}$$

$$+ \sum_{\ell \geq d/10} |a_\ell|(\beta\sqrt{d})^{2\ell+1} \tag{272}$$

$$\leq \Pr[\mathcal{E}_1 \cup \mathcal{E}_2] \left( \frac{1}{2} + \sum_{\ell < d/10} |a_\ell|(\alpha\sqrt{d})^{2\ell+1} \right) + \sum_{\ell \geq d/10} |a_\ell|(\beta\sqrt{d})^{2\ell+1} \tag{273}$$

$$\leq \exp(-c\beta^2 d) \left( \frac{1}{2} + \sum_{\ell < d/10} |a_\ell|(\alpha\sqrt{d})^{2\ell+1} \right) + \sum_{\ell \geq d/10} |a_\ell|(\beta\sqrt{d})^{2\ell+1} \, . \tag{274}$$

In the right term in equation 270 we used that event $\mathcal{F}$ implies $|w| \leq \beta\sqrt{d}$. In equation 271, we apply Claim 8 to the first term. This is valid since $w$ is a linear function of $k$, and since $2\ell + 1 < 2d/10 + 1 \leq d/2 \leq d'$, which holds for $d \geq 4$. In bounding the second term in equation 271, we used a uniform bound $|w| = |\mu x'| \leq \alpha\sqrt{d}$.

We will now argue that both terms in equation 274 are exponentially small. Let us start with the second term:

$$\sum_{\ell \geq d/10} |a_\ell|(\beta\sqrt{d})^{2\ell+1} \leq \sum_{\ell \geq d/10} \frac{(\beta\sqrt{d})^{2\ell+1}}{\ell!} \leq \beta\sqrt{d} \sum_{\ell \geq d/10} \exp(\ell \ln d + 2\ell \ln \beta - \ell \ln \ell + \ell) \tag{275}$$

$$\leq \beta\sqrt{d} \sum_{\ell \geq d/10} \exp(2\ell \ln \beta + \ell \ln 10 + \ell) \tag{276}$$

$$= \beta\sqrt{d} \sum_{\ell \geq d/10} (10e\beta^2)^\ell \leq \beta\sqrt{d} \sum_{\ell \geq d/10} 2^{-\ell} \leq 2\beta\sqrt{d}2^{-d/10} \leq \exp(-c'd) \, , \tag{277}$$

where the first inequality in equation 277 follows if $\beta$ satisfies $10e\beta^2 \leq 1/2$.

Now let us move to the left-hand side term in equation 274. It is sufficient to prove $1/2 + \sum_{\ell < d/10} |a_\ell|(\alpha\sqrt{d})^{2\ell+1} \leq \exp(c\beta^2 d/2)$ and this is what we are going to show. Indeed,

$$\sum_{\ell < d/10} |a_\ell|(\alpha\sqrt{d})^{2\ell+1} \leq \sum_{\ell < d/10} \frac{(\alpha\sqrt{d})^{2\ell+1}}{\ell!} \leq \alpha\sqrt{d} \sum_{\ell < d/10} \frac{(e\alpha\sqrt{d})^{2\ell}}{\ell^\ell} \, . \tag{278}$$

Consider the function $f(\ell) = \frac{(e\alpha\sqrt{d})^{2\ell}}{\ell^\ell}$. We check that its derivative is $f'(\ell) = f(\ell)\big(\ln\big((e\alpha)^2 d\big) - 1 - \ln \ell\big)$. Therefore, $f$ achieves its maximum at $\ell^* = \alpha^2 ed$ and we have

$$\frac{(e\alpha\sqrt{d})^{2\ell}}{\ell^\ell} = f(\ell) \leq f(\ell^*) = \exp(e\alpha^2 d) \tag{279}$$

for every $\ell \geq 0$. For $\alpha$ small enough, for example if $\alpha^2 e \leq c\beta^2/2$, we can substitute into equation 278 to get $\sum_{\ell<d/10} |a_\ell|(\alpha\sqrt{d})^{2\ell+1} \leq \alpha d\sqrt{d}\exp(e\alpha^2 d)$ and consequently

$$1/2 + \sum_{\ell<d/10} |a_\ell|(\alpha\sqrt{d})^{2\ell+1} \leq \exp(c\beta^2 d/2) \ . \tag{280}$$

In summary, by combining equation 274, equation 277, and equation 280, the inequality $T_1 \leq \exp(-\Omega(d))$ is established for large enough $d$.

We now turn to bounding $T_2$. The idea is essentially the same with a more complicated calculation. Recall equation 260, let $b_m := \frac{1}{\sqrt{2\pi}} \frac{(-1)^m}{2^m m!}$ and note for later that $|b_m| \leq 1/m!$. We write down

$$T_2 = \sum_{\ell=0}^{\infty} \left| \mathbb{E}_k (-1)^k \mathbb{1}_{\mathcal{F}} \phi(w) H_\ell(w) \frac{\rho^{\ell+1}}{\sqrt{\ell!}} \right| \tag{281}$$

$$\leq \underbrace{\sum_{\ell<d/10} \left| \mathbb{E}_k (-1)^k \mathbb{1}_{\mathcal{F}} \left( \sum_{m<d/10} b_m w^{2m} \right) H_\ell(w) \frac{\rho^{\ell+1}}{\sqrt{\ell!}} \right|}_{=:T_3} \tag{282}$$

$$+ \underbrace{\sum_{\ell<d/10, m\geq d/10} \frac{1}{m!} \mathbb{E}_{x,y,k} \left| \mathbb{1}_{\mathcal{F}} w^{2m} \frac{H_\ell(w)}{\sqrt{\ell!}} \right|}_{=:T_4} \tag{283}$$

$$+ \underbrace{\sum_{\ell\geq d/10} \mathbb{E}_{x,y,k} \left| \mathbb{1}_{\mathcal{F}} \phi(w) H_\ell(w) \frac{\rho^{\ell+1}}{\sqrt{\ell!}} \right|}_{=:T_5} \ . \tag{284}$$

Let us argue in turns that each of $T_3, T_4, T_5$ is exponentially small proceeding in the reverse order. For $T_5$, we use equation 269 and the fact that event $\mathcal{F}$ implies $|\rho| \leq 1/2$:

$$T_5 \leq \sum_{\ell\geq d/10} 2^{-\ell+1} \leq 2^{-d/10} \ . \tag{285}$$

For $T_4$, we invoke equation 268 and event $\mathcal{F}$ implying $|w| \leq \beta\sqrt{d}$:

$$T_4 \leq 2d \exp(\beta^2 d/4) \sum_{m\geq d/10} \frac{(\beta^2 d)^m}{m!} \leq 2d \exp(\beta^2 d/4) \sum_{m\geq d/10} (10e\beta^2)^m \ . \tag{286}$$

If $\beta$ is chosen such that $(10e\beta^2)^{1/10} \leq 1/2$ and $\exp(\beta^2/4) \leq 1.01$, then we can continue and obtain the desired bound

$$T_4 \leq 2d(1.01)^d 2^{-d} \leq \exp(-c'd) \ . \tag{287}$$

Finally, we turn to $T_3$:

$$T_3 \leq \sum_{\ell<d/10} \mathbb{E}_{x,y} \left| \mathbb{E}_k (-1)^k \left( \sum_{m<d/10} b_m w^{2m} \right) H_\ell(w) \frac{\rho^{\ell+1}}{\sqrt{\ell!}} \right| \tag{288}$$

$$+ \sum_{\ell<d/10} \mathbb{E}_{x,y} \left| \mathbb{E}_k (-1)^k \mathbb{1}_{\mathcal{E}_1 \cup \mathcal{E}_2} \left( \sum_{m<d/10} b_m w^{2m} \right) H_\ell(w) \frac{\rho^{\ell+1}}{\sqrt{\ell!}} \right| \tag{289}$$

$$\leq 2d \Pr[\mathcal{E}_1 \cup \mathcal{E}_2] \exp(\alpha^2 d/4) \sum_{m<d/10} \frac{(\alpha^2 d)^m}{m!} \tag{290}$$

$$\leq 2d^2 \exp(-c\beta^2 d) \exp(\alpha^2 d/4) \exp(e\alpha^2 d) \leq \exp(-c'd) \ . \tag{291}$$

The sum in equation 288 is equal zero by Claim 8: Indeed both $w$ and $\rho$ are linear functions of $k$, so the expression inside the absolute value is a polynomial of degree at most $2m + \ell + (\ell + 1) < 4d/10 + 1 \leq d/2 \leq d'$. To bound the sum in equation 289, we applied $|b_m| \leq 1/m!$, $|w| \leq 3\alpha\sqrt{d}$, equation 268 and $|\rho| \leq 1$. Finally, to bound equation 290 we applied equation 255 and equation 279 and the final inequality follows if we choose $\alpha_0$ small enough so that, e.g., $\alpha^2/4 + e\alpha^2 \leq c\beta^2/2$ (recall that $\beta$ is already chosen to be a small enough absolute constant).

Summing up, equation 285, equation 287 and equation 291 substituted into equation 284 give $T_2 \leq \exp(-\Omega(d))$. Together with $T_1 \leq \exp(-\Omega(d))$, substituted into equation 267, we established $\mathrm{GAL}_{\mathrm{hid}}(\mu, d) \leq \exp(-\Omega(d))$, which is what we set out to prove. $\qquad\square$

### D.5 PROOF OF LEMMA 10

This proof follows a similar process as the proof of Lemma 9, so we will skip some details and refer to Appendix D.4. Let us write $\mu = \alpha/\sqrt{d}$, so that by assumption $\alpha \leq \alpha_0$. We have,

$$\mathrm{GAL}_{\mathrm{out}} = \mathbb{E}_{g,x,x',r}\left[\prod_i^d x_i x_i' \operatorname{ReLU}(g \cdot x + \mu r \cdot x) \operatorname{ReLU}(g \cdot x' + \mu r \cdot x')\right] \tag{292}$$

$$= \mathbb{E}_{x,x',r}\left[\prod_i x_i x_i' \mathbb{E}_g\left[\operatorname{ReLU}(g \cdot x + \mu r \cdot x) \operatorname{ReLU}(g \cdot x' + \mu r \cdot x')\right]\right] \tag{293}$$

$$:= \mathbb{E}_{x,x',r} F(x, x', r) . \tag{294}$$

We still have for every $x, x', r, s \in \{-1, 1\}^d$, $F(x, x', r) = F(x \odot s, x' \odot s, r \odot s)$, therefore

$$\mathrm{GAL}_{\mathrm{out}} = \mathbb{E}_{x,x'} F(x, x', 1^d)$$

$$= \mathbb{E}_{x,x'}\left[\prod_i x_i x_i' \mathbb{E}_g\left[\operatorname{ReLU}(g \cdot x + \mu \cdot x) \operatorname{ReLU}(g \cdot x' + \mu \cdot x')\right]\right] .$$

Let's recall the notations from Appendix D.4: let's fix $x$ and assume w.l.o.g. that $x = (1^{d-d'}, -1^{d'})$ for some $0 \leq d' \leq d$, $x' = (y, z)$ such that $y \in \{-1, 1\}^{d-d'}$ and $z \in \{-1, 1\}^{d'}$. Assume that $d' \geq d/2$ and fix $y$. Let $G(x, y, z) = F(x, (y, z), 1^d)$. We are going to analyze $\mathbb{E}_z G(x, y, z)$ so that the bound on $\mathbb{E}_{x,x'} F(x, x', 1^d) = \mathbb{E}_{x,y,z} G(x, y, z)$ will follow by averaging. Let $\rho = \frac{1}{d} x \cdot x' = \frac{1 \cdot y + 2k - d'}{d}$, where $k$ be the number of $-1$ entries in $z$. We have,

$$|\mathbb{E}_z G(x, y, z)| = \left|(-1)^{d'} \prod_{i=1}^{d-d'} y_i \mathbb{E}_z\left[(-1)^k \mathbb{E}_g\left[\operatorname{ReLU}(g \cdot x + \mu \cdot x) \operatorname{ReLU}(g \cdot x' + \mu \cdot x')\right]\right]\right| \tag{295}$$

$$= \left|\mathbb{E}_k\left[(-1)^k \Lambda_\rho(\mu(d - 2d'), -\mu(d\rho - 2 \cdot y))\right]\right| , \tag{296}$$

where in this case $\Lambda_\rho(a, b) = \mathbb{E}_{g,g'}\left[\operatorname{ReLU}(g + a) \operatorname{ReLU}(g' + b)\right]$, with $g, g'$ are two standard $\rho$-correlated joint Gaussians. Let

$$A := \mu(d - 2d') , B := 2\mu \cdot y , C := -\mu d , \text{ and } w := B + C\rho . \tag{297}$$

Let us define two "bad" events: $\mathcal{E}_1$ is $|\rho| \geq 1/2$ and $\mathcal{E}_2$ is $|w| \geq \beta\sqrt{d}$ (for some $\beta$ that we will set later) and let $\mathcal{F}$ be the complement of $\mathcal{E}_1 \cup \mathcal{E}_2$.

Using the same argument as in Appendix D.4 (see Equationsequation 250-equation 255), we can show that $\Pr[\mathcal{E}_1 \cup \mathcal{E}_2] \leq \exp(-c\beta^2 d)$ for some universal $c > 0$ and $d$ large enough. Continuing,

$$\mathrm{GAL}_{\mathrm{out}} \leq \mathbb{E}_{x,y}\left|\mathbb{E}_k(-1)^k \Lambda_\rho(A, w)\right| \tag{298}$$

$$\leq \exp(-c\beta^2 d) + \mathbb{E}_{x,y}\left|\mathbb{E}_k(-1)^k \Lambda_\rho(A, w)\mathbb{1}_{\mathcal{F}}\right| . \tag{299}$$

From Corollary 7, we have

$$\Lambda_\rho(A, w) = R(A)R(w) + \Phi(A)\Phi(w)\rho + \phi(A)\phi(w)\sum_{\ell=0}^{\infty} \frac{H_\ell(A)H_l(w)}{(\ell+2)!}\rho^{\ell+2} , \tag{300}$$

with $R(x) = x\Phi(x) + \phi(x)$. Substituting the above into equation 300

$$\text{GAL}_{\text{hid}} \le \exp(-c\beta^2 d) + \mathbb{E}_{x,y}\left|\mathbb{E}_k(-1)^k \mathbb{1}_{\mathcal{F}}\cdot\right. \tag{301}$$

$$\left(R(A)R(w) + \Phi(A)\Phi(w)\rho + \phi(A)\phi(w)\sum_{\ell=0}^{\infty}\frac{H_\ell(A)H_l(w)}{(\ell+2)!}\rho^{\ell+2}\right)\bigg| \tag{302}$$

$$\le \exp(-c\beta^2 d) + \mathbb{E}_{x,y}\left|\mathbb{E}_k(-1)^k \mathbb{1}_{\mathcal{F}}R(A)R(w)\right| + \mathbb{E}_{x,y}\left|\mathbb{E}_k(-1)^k \mathbb{1}_{\mathcal{F}}\Phi(A)\Phi(w)\rho\right| \tag{303}$$

$$+ \sum_{\ell=0}^{\infty}\mathbb{E}_{x,y}\left|\mathbb{E}_k(-1)^k \mathbb{1}_{\mathcal{F}}\phi(A)\phi(w)H_\ell(A)H_\ell(w)\frac{\rho^{\ell+2}}{(\ell+2)!}\right|, \tag{304}$$

$$\le \exp(-c\beta^2 d) + \mathbb{E}_{x,y}|R(A)|\left|\mathbb{E}_k(-1)^k \mathbb{1}_{\mathcal{F}}w\Phi(w)\right| + \mathbb{E}_{x,y}|R(A)|\left|\mathbb{E}_k(-1)^k \mathbb{1}_{\mathcal{F}}\phi(w)\right| \tag{305}$$

$$+ \mathbb{E}_{x,y}\left|\mathbb{E}_k(-1)^k \mathbb{1}_{\mathcal{F}}\Phi(w)\rho\right| + \sum_{\ell=0}^{\infty}\mathbb{E}_{x,y}\left|\mathbb{E}_k(-1)^k \mathbb{1}_{\mathcal{F}}\phi(w)H_\ell(w)\frac{\rho^{\ell+2}}{\sqrt{\ell!}}\right|, \tag{306}$$

$$\le \exp(-c\beta^2 d) + \mu d\underbrace{\mathbb{E}_{x,y}\left|\mathbb{E}_k(-1)^k \mathbb{1}_{\mathcal{F}}w\Phi(w)\right|}_{=:T_{11}} + \mu d\underbrace{\mathbb{E}_{x,y}\left|\mathbb{E}_k(-1)^k \mathbb{1}_{\mathcal{F}}\phi(w)\right|}_{=:T_{12}} \tag{307}$$

$$+ \underbrace{\mathbb{E}_{x,y}\left|\mathbb{E}_k(-1)^k \mathbb{1}_{\mathcal{F}}\Phi(w)\rho\right|}_{=:T_{13}} + \underbrace{\sum_{\ell=0}^{\infty}\mathbb{E}_{x,y}\left|\mathbb{E}_k(-1)^k \mathbb{1}_{\mathcal{F}}\phi(w)H_\ell(w)\frac{\rho^{\ell+2}}{\sqrt{\ell!}}\right|}_{=:T_{22}}, \tag{308}$$

The bound in equation 307 and equation 308 follow because of equation 269 and the fact that $|R(A)| \le 2|A| = 2\mu(d - d') \le \mu d$. It remains to show that $T_{11}$, $T_{12}$, $T_{13}$ and $T_{22}$ are exponentially small. The term $T_{22}$ differs from $T_2$ in equation 267 by the exponent of $\ell + 2$ in $\rho$ instead of $\ell + 1$. Thus for $d$ large enough, a similar proof as for $T_2$ will show that $T_{22}$ is exponentially small. The process to handle $T_{11}$, $T_{12}$ and $T_{13}$ is the same as in $T_1$. Indeed, for example:

$$T_{11} \le \mathbb{E}_{x,y}\left|\mathbb{E}_k(-1)^k \mathbb{1}_{\mathcal{F}}\left(\frac{1}{2}w + \sum_{\ell<d/10}a_\ell w^{2\ell+2}\right)\right| + \sum_{\ell\ge d/10}|a_\ell|(\beta\sqrt{d})^{2\ell+2} \tag{309}$$

$$\le \exp(-c\beta^2 d)\left(\frac{1}{2}\alpha\sqrt{d} + \sum_{\ell<d/10}|a_\ell|(\alpha\sqrt{d})^{2\ell+2}\right) + \sum_{\ell\ge d/10}|a_\ell|(\beta\sqrt{d})^{2\ell+2}. \tag{310}$$

Both of the above terms can be handled similarly as in Appendix D.4.

# E EXPERIMENT DETAILS AND ADDITIONAL EXPERIMENTS

## E.1 EXPERIMENT DETAILS

All experiments were performed using the PyTorch framework (Paszke et al. (2019)) and they were executed on NVIDIA Volta V100 GPUs.

**Architectures.** For the results presented in the main, we used mainly a 4-layer MLP architecture trained by SGD with the hinge loss. In this Section, we also present some experiments obtained with a 2-layer MLP trained by SGD with the squared loss.

- **4-layer MLP.** This is a fully-connected architecture of 3 hidden layers of neurons of size $512, 512, 64$, and ReLU activation.

- **2-layer MLP.** This is again a fully-connected architecture, with 1 hidden layer of $512$ neurons, and ReLU activation,

**Initializations.** We compare few initialization schemes. In the following, $\dim$ denotes the input dimension of the layer of the corresponding parameter. All layers weights and biases are independently initialized according to:

- **$\sigma$-perturbed Rademacher:** $\left(\mathrm{Rad}(1/2) + \mathcal{N}(0, \sigma^2)\right) \cdot \frac{1}{\sqrt{\dim \cdot (1+\sigma^2)}}$.

- **Gaussian:** $\mathcal{N}(0, \frac{1}{\dim})$.

- **$s$-sparsified Rademacher:** $\mathrm{Ber}(1-s) \cdot \mathrm{Rad}(1/2) \cdot \frac{1}{\sqrt{\dim \cdot (1-s)}}$.

- **Uniform $\sigma$-perturbed Rademacher:** $\left(\mathrm{Rad}(1/2) + \mathrm{Unif}[-\sqrt{3}\sigma, \sqrt{3}\sigma]\right) \cdot \frac{1}{\sqrt{\dim \cdot (1+\sigma^2)}}$.

- **Discrete perturbed Rademacher:** $\mathrm{Unif}\{-2, -1, 1, 2\} \cdot \sqrt{\frac{2}{5 \cdot \dim}}$.

**Training procedure.** We consider mainly the hinge loss: $L_{\mathrm{hinge}}(\hat{y}, y) := \max(0, 1 - \hat{y}y)$. In some experiments we consider the $\ell_2$ loss: $L_{\ell_2}(\hat{y}, y) := (\hat{y} - y)^2$. We train the architectures using SGD with batch size $64$. In the online setting, we sample fresh batches of samples at each iterations. In the offline setting, we sample batches from a fixed dataset and we stop training when the training loss is less than $0.01$.

**Hyperparameter tuning.** The primary goal of our experiments is to conduct a fair comparison of different initialization methods. Thus, we did not engage in extensive hyperparameter tuning. We tried different batch sizes and learning rates, and we did not observe significant qualitative difference. We chose to report the experiments obtained for a standard batch size of $64$ and a learning rate of $0.01$.

**Additional details for Figure 2.** In the left plot of Figure 2, we are computing the quantity $\mathbb{E}_w \left[ \mathbb{E}_{x,r} \left[ \frac{\partial L(w,x,f(x))}{\partial w_d} - \frac{\partial L(w,x,r)}{\partial w_d} \right]^2 \right]$, where $w \sim \mathcal{N}(0, \frac{1}{d} \mathrm{Id}_d)$ for one case and $w \sim \mathrm{Rad}(1/2)$ for the other case, $f$ is the full parity, $r \sim \mathrm{Rad}(1/2)$ and $L(w,x,y) := \max\left(0, 1 - y\,\mathrm{ReLU}(w.x)\right)$ is the hinge loss. For the approximated part we update the weights according to $\psi^{t+1} = \psi^t - \gamma\left(\Gamma_r(\psi^t)\right)$, with $\psi^0 \sim \mathcal{N}(0, \frac{1}{d} \mathrm{Id}_d)$ and $\gamma = 1$, and we calculate $\mathbb{E}_{\psi^t} \left[ \mathbb{E}_{x,r} \left[ \frac{\partial L(\psi^t,x,f(x))}{\partial \psi_d^t} - \frac{\partial L(\psi^t,x,r)}{\partial \psi_d^t} \right]^2 \right]$.

## E.2 ADDITIONAL EXPERIMENTS

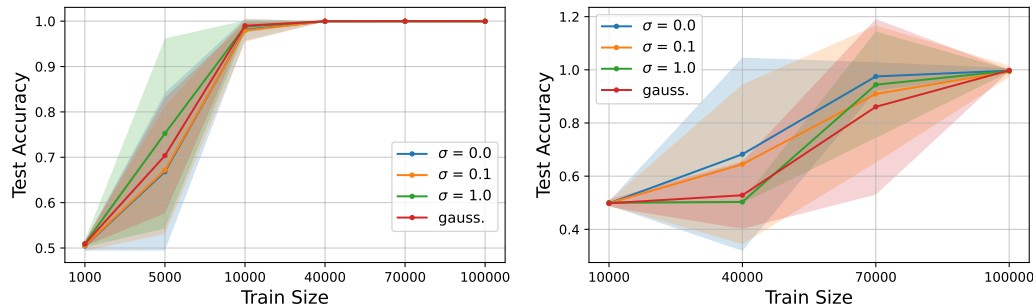

Figure 4: Learning 3-parity (left) and 5-parity (right) with Rademacher, $\sigma$-perturbed and Gaussian initializations, with SGD with the hinge loss on a 4-layer MLP, with $d = 50$. We plot the test accuracy, for several training set sizes.

**Sparse Parities.** In Figure 4 we train a 4-layer MLP with Rademacher initialization and $\sigma$-perturbation ($\sigma \in 0.1, 1$) on two sparse parities: degree 3 (left) and degree 5 (right). We observe no significant difference between these initializations, unlike the full parity case. This is because, for sparse parities, the learning bottleneck lies in recovering the support, which takes $d^{\Omega(k)}$ time for

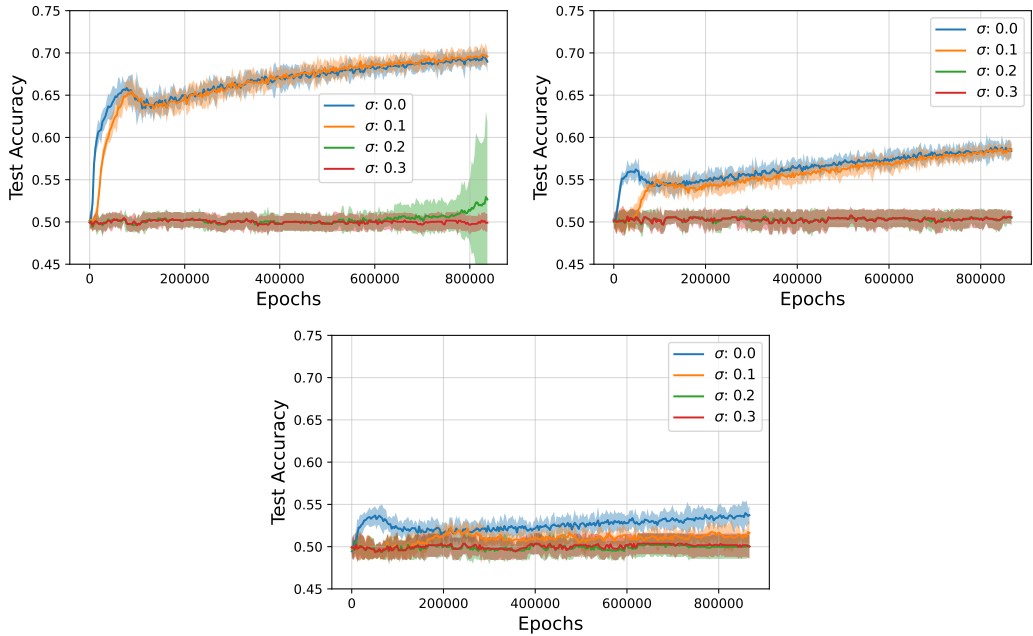

Figure 5: Learning the full parity with $\sigma$-perturbed initialization by SGD with the hinge loss on a 4-layer MLP, with input dimension $d = 100$ (top-left), $d = 150$ (top-right) and $d = 200$ (bottom), with online fresh samples.

any i.i.d. initialization. Hence, the initial embedding does not play the same role as in the full parity scenario.

**Larger input dimension.** In Figure 5, we plot the test accuracy achieved by a 4-layer MLP trained with the hinge loss on the full parity task, with different $\sigma$-perturbed initializations. We report only the curves for small $\sigma$. We observe that for fixed $\sigma$, learning becomes hard as $d$ increases.

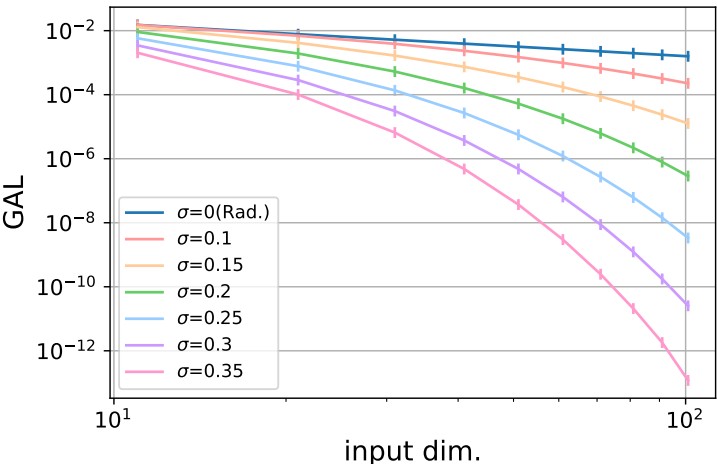

Figure 6: Computing numerically $\mathrm{GAL}_f$ for correlation loss for one-neuron with threshold activation. We report the estimated $\mathrm{GAL}_f$ for different values of the input dimension, in a log-log plot.

**Alignment for correlation loss.** Figure 6 completes Figure 2 (right) in the main. Here we plot the numerically computed $\mathrm{GAL}_f$ for larger values of $\sigma$. We observe that the $\mathrm{GAL}_f$ becomes con-

sistently smaller as $\sigma$ increases. Moreover, from the plot the decay seems super-polynomially small for all $\sigma > 0$.

**Two-layer MLP and squared loss.** In Figure 7 we train a 2-layer MLP with the squared loss and online fresh samples. In the left plot, we initialize the weights according to $\sigma$-perturbed Rademacher, for different values of $\sigma$. In the right plot, we initialize with other perturbations of the Rademacher initialization, namely a mixture of (continuous) uniform distributions of mean $+1$ and $-1$ and standard deviation $\sigma$ and $s$-sparsified Rademacher with $s = 1/3$. We observe in both plots a similar behavior as for the 4-layer MLP with the hinge loss.

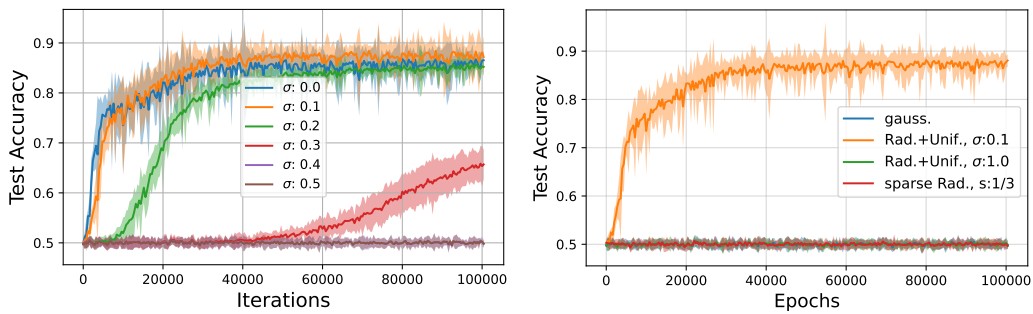

Figure 7: Learning the full parity with $\sigma$-perturbed Rademacher (left) and uniform and sparse perturbed Rademacher (right) with a 2-layer MLP, with input dimension $d = 50$, trained with the squared loss, with online fresh samples.

**Effect of the Loss.** We consider the following Boolean function:

$$f(x) := \frac{1}{8}x_1 x_2 x_3 + \frac{3}{8}x_1 x_2 x_4 + \frac{1}{4}x_1 x_3 x_4 + \frac{1}{4}x_2 x_3 x_4. \tag{311}$$

In (Joshi et al. (2024)), the authors show that this function is learned more efficiently by SGD with L1-loss than with L2-loss (see Section 7.1 therein). In Figure 8, we observe that such difference is captured by our loss-dependent notion of Initial Gradient Alignment (GAL). This motivates future work in comparing our GAL with previously defined measures (e.g. LGA (Mok et al. (2022))) in a broader setting.

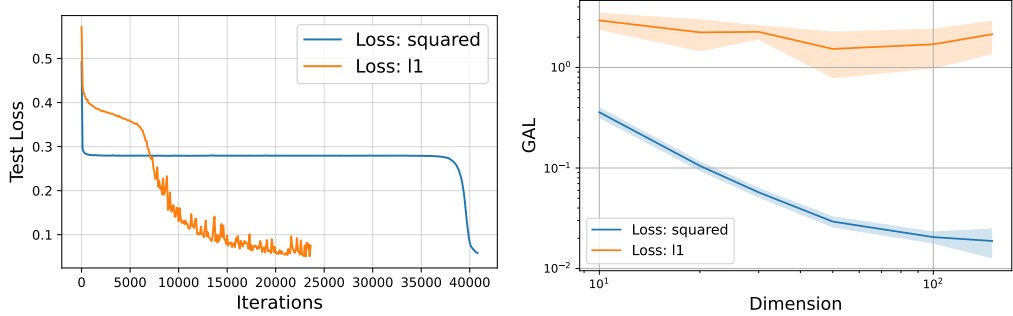

Figure 8: (left) Learning $f$ (Eq. equation 311) with SGD with the L1 and L2 (squared) loss on a 4-layer MLP, with input dimension $d = 50$. (right) Initial GAL for $f$ on the same architecture, with the two losses.

**Output Layer Training with Correlation Loss.** The purpose of Figure 9 is to empirically verify our positive theoretical result from Theorem 5. To that purpose, we train a two-layer fully connected network with Rademacher initialization with ReLU and clipped-ReLU activation $\sigma(x) = \min(1, \max(0, x))$ on the full parity task. We train only the output layer, consistently with our positive result, with SGD with large batch size (1024) with the correlation loss and online fresh

samples, until convergence of the test accuracy. We show the test accuracy achieved for different input dimensions ($d$) and different widths of the hidden layer ($w$). Consistently with our theory, with clipped-ReLU, $d^2$ hidden neurons are sufficient to achieve accuracy 1 (left). For ReLU, we observe that $w = O(d^2)$ is not enough to achieve perfect accuracy and we believe that our theoretical bound (i.e $\Omega(d^4)$) for learning with accuracy 1 may not be tight and that $d^3$ or $d^{3.5}$ may be sufficient (right).

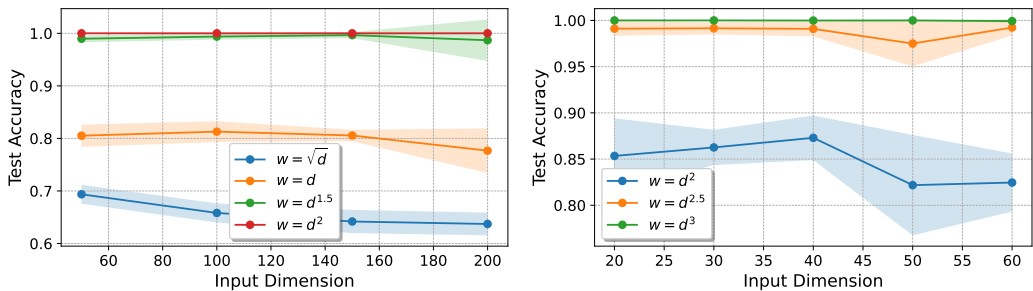

Figure 9: Learning the full parity with a 2-layer network, where only the output layer is trained by SGD with the correlation loss. We report the test accuracy achieved after training, for clipped-ReLU activation (left) and ReLU (right), for different input dimensions ($d$) and hidden layer width ($w$).

