# OpenReview forum: "Learning High-Degree Parities: The Crucial Role of the Initialization"
_ICLR.cc/2025/Conference — ICLR 2025 Poster_

### Official Review · Reviewer_mApU · 2024-10-31

**Soundness:** 3
**Presentation:** 2
**Contribution:** 3
**Rating:** 8
**Confidence:** 3

**Summary:**

The paper proves that the full parity is learnable with SGD on 2-layer rademacher-initialized neural networks, with the hinge and correlation loss. The paper presents a notion of gradient alignment, which measures the distance between the gradients on the true distribution and those of a randomly-labeled distribution. The authors then use this measure to upper bound the accuracy any NN with a linear output layer can achieve with noisy GD and the correlation loss, providing a general negative result for learning. They follow up with application of that result to show that almost-full parities are not learnable when initialized differently enough from the rademacher init., all in all providing a separation result for learning full parities, depending on the initialization.

**Strengths:**

The authors advance an active and open area of research, providing answers on the learnability of full parities and almost-full parities.
The authors introduce a new measure that might be used to derive negative learning results in future research.

**Weaknesses:**

The phrasing in the abstract and introduction can be a bit misleading, in a way that suggests that your positive results include almost-full parities.

While the term "full parity" is mentioned in the abstract, no place indicates that the positive results refer to that case, and not the more general one of almost full parities. This omission is then continued in the introduction (lines 59-78), where you mention Abbe & Boix-Adsera who considered learning almost-full parities, then stating "First, we show that SGD on a two-layer fully connected
ReLU network with Rademacher initialization can achieve perfect accuracy, thus going beyond weak learning". Which could imply again that you have showed learnability of almost-full parities, as the last case you referred to is that of almost-full parities.

Personally I was confused, and have searched around the paper for the more general case for a while. (the only place where the result is carefully stated is the conclusion). I suggest changing the phrasing to avoid this confusion. It could be as simple as changing the above sentence to "...can achieve perfect accuracy on full parities, thus going beyond weak learning". Similarly, changing the abstract to "enables efficient learning of full parities".

Theorem 4 is long. I suggest breaking it down claims / corollaries / etc, according to the items in the theorem.

I suggest using a consistent format throughout all related theorems, containing all relevant information. e.g. for the positive results the reader is referred to the text to find the definition of the target function (full parity), while the target function for the negative results is stated in the theorem.

**Questions:**

What is the obstacle in applying the technique of the positive results to almost-full parities?

---

> ### Author Response · Authors · 2024-11-19
> **Rebuttal to reviewer mApU**
>
> We thank the reviewer for constructive comments and for appreciating our work.
>
> - *Phrasing in the abstract and introduction, Positive result for almost full parities.*
>
> We thank the reviewer for pointing our some confusion in our presentation. We refer to our general response for an extension of our positive result to include almost-full parities.
>
> - *Breaking down Theorem 4, consistency of format between positive and negative sections.*
>
> We thank the reviewer for the suggestions. We will implement the changes in a revised version of the paper.

---

> > ### Author Response · Authors · 2024-11-29
> >
> > Thank you once again for taking the time and effort to review our paper. We implemented the changes suggested in the revised version, including an extension of the positive result to almost full parities (we refer to the general response for details).
> >
> > As the discussion period is approaching its conclusion, we would greatly appreciate it if you could let us know if you have any additional questions or concerns that we can address.

---

### Official Review · Reviewer_YKeF · 2024-11-03

**Soundness:** 4
**Presentation:** 3
**Contribution:** 3
**Rating:** 8
**Confidence:** 4

**Summary:**

This paper studies the problem of learning Boolean parities using gradient descent. This is a problem which in recent years has come to serve as a useful way of understanding the effectiveness and dynamics of training neural networks using gradient descent when the labels arise from a concrete, classical functional class. The set of all parities (or indeed any superpolynomial-size subset) is well-known to be SQ-hard to learn efficiently. This paper instead focuses on the the singleton function class consisting of the full parity, or the polynomial-size function class consisting of "almost-full" parities (of size $d - O(1)$). The main question tackled is the following: what is the effect of the precise neural network initialization on whether GD succeeds at learning such parities?

The main results are as follows:
- In general, for any fixed target function, whether or not GD/SGD with a particular initialization scheme succeeds is largely determined by a quantity dubbed the "gradient alignment" at initialization. Importantly, the gradient alignment is both target-specific and input distribution-specific. (Caveats: this characterization holds assuming certain types of Gaussian-smoothed, symmetric initialization schemes, neural networks with linear outer layers, and certain natural losses.)
- As an instantiation of this result, it turns out that Gaussian initializations as well as Gaussian-perturbed Rademacher initializations (with a sufficiently large perturbation) do _not_ have high gradient alignment and do not succeed at learning dense parities. (Subject to the same caveats above.)
  - In particular, this holds for poly-sized two-layer ReLU networks with pure Gaussian initializations.
- By contrast, the pure Rademacher initialization (or Gaussian-perturbed Rademacher with sufficiently small perturbation) is unusual and does succeed at efficiently learning dense parities.

**Strengths:**

The paper makes a useful contribution to our understanding of what neural networks can learn using gradient descent. In particular, the role of the initialization is clearly important in practice, and this paper considers a natural toy theoretical setting in which to study it formally.  The main results (both positive and negative) and proof techniques are interesting, and the separation between Gaussian and Rademacher is arguably a little surprising a priori. The most general negative result for arbitrary target functions (Theorem 7) seems particularly interesting and potentially useful even in practice. The technical contribution here in general is solid, although it very much builds on a large body of prior work with similar techniques. The paper is clearly written.

**Weaknesses:**

I am a little unsure about the novelty and significance of the paper. The problem of learning parities started as a natural toy theoretical setting for understanding what neural networks can learn, and there have recently been several papers that have studied this problem in increasing detail. The picture that is emerging for parities looks roughly as follows:
- In general they are trivially PAC-learnable by Gaussian elimination
- But they are SQ-hard (and also GD-hard) unless our prior information restricts the target to a small subset
- For parities of size $k$, GD and SGD on natural architectures do succeed in time $n^k$ (Barak et al 2023; Abbe et al 2023)
- For a single fixed parity, it is learnable by GD under certain initialization schemes (inner weights all 1, as in Nachum and Yehudayoff 2019, or Rademacher, as in Abbe and Boix-Adsera 2022)
- Again for a single fixed parity, it is not learnable by GD under other initialization schemes (such as the Gaussian, by the results of this paper, eg Theorem 3 --- under certain assumptions)

It would be nice if the authors could clarify how we should view this picture as a whole, especially keeping in mind the original motivation where the parities were just examples of complex functions. In general for learning a potentially complex function using deep learning, is there concrete guidance on what architectures, initializations etc do and do not work? For example, should we prefer Rademacher over Gaussian initializations in general? Should we always try to estimate the GAL before actually training? Does the GAL view subsume all prior results? My concern is that the parities are a very special case that allow various arguably superficial algebraic manipulations, and so lessons here may not be widely applicable.

The general negative result for general target functions is more interesting. I would view this as the main technical result of the paper. The techniques here borrow from prior work (esp Abbe and Boix-Adsera 2022, Abbe et al 2022), and the GAL quantity and the arguments here also seems related to quantities like the "variance" considered in Shalev-Shwartz et al 2017 and follow-up works. It seems that the main difference is that these prior works did not focus on the role of the initialization per se. Can the authors comment in more detail on the differences in proof techniques?

it would be nice to have the authors clarify their case for novelty and significance in light of these remarks and questions. On balance I am leaning positively but would like to hear their case first and hence am rating this a 6 for now.


#### References

Boaz Barak, Benjamin L. Edelman, Surbhi Goel, Sham Kakade, Eran Malach, Cyril Zhang. "Hidden Progress in Deep Learning: SGD Learns Parities Near the Computational Limits." 2022.

Ido Nachum, Amir Yehudayoff. "On Symmetry and Initialization for Neural Networks." 2019.

Emmanuel Abbe, Enric Boix-Adsera. "On the non-universality of deep learning: quantifying the cost of symmetry." 2022.

Emmanuel Abbe, Elisabetta Cornacchia, Jan Hązła, Christopher Marquis. "An initial alignment between neural network and target is needed for gradient descent to learn." 2022.

Emmanuel Abbe, Enric Boix-Adsera, Theodor Misiakiewicz. "SGD learning on neural networks: leap complexity and saddle-to-saddle dynamics." 2023.

Shai Shalev-Shwartz, Ohad Shamir, Shaked Shammah. "Failures of Gradient-Based Deep Learning". 2017.

**Questions:**

1. Repeating from above: it would be nice to get the authors' thoughts on the emerging picture around parities and complex function learning, and what the broader takeaways are.
2. Similarly regarding differences in proof techniques from prior works, esp Abbe and Boix-Adsera 2022 and Abbe et al 2022 (the INAL paper).
3. I am curious how Theorem 7 might be used in practice. It seems that the gradient alignment at initialization is something we can estimate purely using some labeled examples, without knowing the target. Does this mean that the following is a practical procedure when fitting any neural network by GD: first measure the GAL at initialization using some labeled examples, if it is small essentially abandon hope (for most reasonable architectures and initializations), otherwise proceed? Basically what power do we gain from the fact that we can estimate the GAL? Considering this is a target-specific, distribution-specific measure, this would be the dream result (namely a target-specific characterization of "practical" learnability) and seems like a very powerful operational interpretation --- if valid.
I wonder if there is an important distinction between the GAL being merely polynomially small and superpolynomially small. If it is only polynomially small, it might still be learnable in polynomial time. And to check if it is truly superpolynomially small, we would need superpolynomial sample complexity. So perhaps it is infeasible to measure it accurately enough for it to be useful?
4. Separately, a more minor technical point: it seems unfortunate that the main negative result (Thm 7) involves an additional Gaussian smoothing on top of an already smoothed initialization. I realize Corollary 1 tries to fix this but in any case it seems like an odd technical artifact. I am curious if things would be simplified if the initial Gaussian perturbation was itself defined as purely additive (i.e. $A + H_\sigma$ instead of $A + \\sqrt{Var(A)} H_\sigma$).

---

> ### Author Response · Authors · 2024-11-19
> **Rebuttal to reviewer YKeF**
>
> We thank the reviewer for the constructive comments and for a summary of the big picture for parities.
>
> - *What’s the emerging picture around parities and complex function learning, and what are the broader takeaway? What does this paper add to it?*
>
> At the theory level, the paper indicates that deep learning with regular networks is not related to Statistical-Query (SQ) learning under “stability” assumptions, where `stability' refers to small changes in the training parameters (as the initialization). Namely, several papers  (such as https://arxiv.org/pdf/2108.04190) have asked whether the paradigm of learning with regular networks trained by SGD would be PAC or SQ or something smaller. It is known that PAC cannot be reached with batches that are too large compared to the precision available. One consequence of our negative result is that SQ itself is not achievable if we do not initialize the network carefully, since the full parity is SQ learnable. From a more practical point of view, it raises the point that discrete initializations may be useful in settings where high-degree functions are expected to be learned. While expecting Rademacher initialization to be broadly useful is perhaps too optimistic, there can be other surprising cases where well chosen discrete initialization might be advantageous.
>
> The GAL concept builds on previous work including cross-predictability [Abbe-Sandon,'20] and initial alignment [Abbe et al.,'22]. Indeed, it subsumes those in the sense that we believe that their main results can be restated in terms of GAL. We further refer to the response to reviewer oGKm for comments on our notion of GAL with respect to other measures defined, e.g. the Label-Gradient Alignment (LGA).
>
> Finally, perhaps one more general lesson is a cautionary one. It seems there is a sense in which the full parity function is hard to learn for GD on a neural network, however such results can be stated only in a limited way. Even assuming the symmetry of initialization up to sign changes is not sufficient, as the Rademacher initialization has this property. Therefore, care is needed in stating and interpreting GD lower bounds.
>
> - *Difference in the proof techniques compared to [Abbe, Boix,’22, Abbe et al.,’22]*
>
> The key distinction between the negative results in this paper and [Abbe, Boix ’22; Abbe et al., ’22] lies in the nature of our proof, which does not rely on symmetries or the hardness of the orbit class of the target function under permutations. Instead, our approach involves directly bounding the gradient alignment along the junk flow—a challenge that previous works addressed through the cross-predictability of the orbit class. Furthermore, we introduce the concept of initial gradient alignment as a more appropriate complexity measure, which captures both the influence of initialization and the inherent difficulty of learning some functions that are hard despite having learnable orbit classes, such as the full parity.
>
> - *How can Theorem 7 be used in practice? Should we estimate the GAL at initialization (from labeled examples) and abandon if it is small?*
>
> Yes, in practice one can estimate the GAL and abandon learning if it is small, since a small GAL corresponds to a long time needed to achieve weak learning. Moreover, the actual value (and scaling) of the GAL seems to characterize the complexity of escaping the initial saddle, and not just whether it is poly or super-poly small, as in the case of sparse k-parities and single-index models (see response to reviewer sB62). One should notice that a large GAL does not imply a successful strong learning of the target in a short time, since the dynamic can get stuck in other saddles that are not visible at initialization.
>
> - *Define Gaussian perturbation as $A+ \sqrt{Var(A)} H_\sigma.$*
>
> Indeed, our technique requires that we establish that GAL is small not just for the perturbed initialization, but also for its further smoothing. In our proofs for parity, it is easy to conclude that further smoothing can only decrease the GAL. We considered variations of the definition of perturbed initialization, including the one suggested by the reviewer, but we do not believe it would significantly change the presentation of our results.

---

> > ### Comment · Reviewer_YKeF · 2024-11-21
> > **Intrigued by the strength of GAL**
> >
> > Thank you for the responses. The comments on the bigger picture are helpful, although ultimately I remain on the fence re the significance of the takeaways here.
> >
> > I am more intrigued by GAL subsuming prior work on cross-predictability and initial alignment (although it definitely needs a formal proof), as well as having this very strong guarantee ("if GAL is small, abandon training"). I do appreciate that it is a one-sided guarantee and that large GAL does not imply successful strong learning. Still, this appears to be an unusually strong result. For practitioners it seems to offer an invaluable way to test cheaply ahead of time if their chosen architecture and hyperparameters etc are doomed to fail without actually carrying out training at all. It is a (one-sided) way to ascertain the possibility of success without actually training. I have two questions:
> > 1. This seems almost a little too good to be true. Can the authors give an honest assessment of whether there is a catch here? What assumptions are required for this to go through, and when might they fail to hold in practice?
> > 2. And if it is indeed true (even with qualifications), this seems like a far stronger general result than the particular application to full parities, and hence worth emphasizing more. Is there a reason that GAL isn't pitched as the chief contribution of the paper?
> >
> > I would be quite curious to hear other reviewers' and the area chair's thoughts on this as well.

---

> ### Author Response · Authors · 2024-11-22
> **On the strength of GAL**
>
> Thank you for your engagement. We do like our results, though we prefer to err on the side of not overstating them.
>
> - *This seems almost a little too good to be true. Can the authors give an honest assessment of whether there is a catch here? What assumptions are required for this to go through, and when might they fail to hold in practice?*
>
> Currently Theorem 6 applies only to the correlation loss. The general upper bound in (76) applies for a general loss function, but it requires a small GAL at every time step t. In the simpler case of the correlation loss, we can show some connection between GAL at initialisation and at later times (Theorem 6, Corollaries 3 and 4). It is in fact tempting to conjecture such a connection for other losses and we have some empirical validation, see Figure 8 in the revision. However, we do not have a theoretical justification nor counterexamples (which would be interesting in their own right too).
>
> More generally, we view Theorem 6 as a guarantee for challenging cases, i.e. scenarios where the dynamics remain stuck near initialisation for a significant time (hardness of weak-learning). We expect this behaviour to happen for targets presenting symmetries and requiring some level of `logical reasoning’ (e.g. arithmetic, graphs, syllogisms), for which parities are a simple model. We do not expect such stagnation to happen e.g. for datasets like CIFAR or MNIST, but we do believe that the GAL may be useful for these other settings.
>
> - *And if it is indeed true (even with qualifications), this seems like a far stronger general result than the particular application to full parities, and hence worth emphasising more. Is there a reason that GAL isn't pitched as the chief contribution of the paper?*
>
>
> The relationship between small initial alignment and the impossibility of weak learning was explored in [ACHM22], where the authors define the initial alignment (INAL) as the average maximum correlation between neurons in the network and the target. As the authors
>  show, this notion subsumes the cross-predictability of [AS20] but their approach relies on an input extension to guarantee the hardness of the orbit class, which prevents proving the hardness of functions with a learnable orbit, such as almost-full parity
>  functions.
>
> Our work diverges from [ACHM22] in the following ways (see also the related work section):
>
> 1. Our notion of alignment (GAL) is based on the gradient norm, which is arguably
>  easier to estimate than INAL, as it does not require finding the maximum correlation between all neurons and the target.
>
> 2. Crucially, we introduce a novel proof technique that establishes hardness of learning directly in terms of the GAL, without relying on input extensions or orbit classes. This is a critical improvement both in terms of theory and potential practical relevance. However, this approach is currently limited to the correlation loss. We believe that this technique, and the bound in (76), may have broader applications and agree with the reviewer for the practical appeal.
>
> To clarify our motivations for not positioning GAL upfront as our main contribution:
>
> 1. The connection between small initial alignment and no learning with more practical implementations were already put forward in [ACHM22] and in this paper we improve and strengthen this approach. So we focused more on this improvement but we agree with the reviewer in hindsight that we should also bring back some of the more general motivations.
>
> 2. It remains open to investigate whether GAL can serve as a guarantee for strong learning.
>
> We thank the reviewer for suggesting greater emphasis on GAL. In the revised manuscript, we will highlight its relevance and provide a more detailed discussion of our contributions relative to [ACHM22]
>
>
> - *References*:
>
> [AS20]:
>  Abbe, Sandon, *On the universality of deep
>  learning*, 2020.
>
> [ACHM22]:
>  Abbe, Cornacchia, Hazla, Marquis, *An initial
>  alignment between neural network and target is needed for gradient descent to learn*,
> 2022.

---

> > ### Comment · Reviewer_YKeF · 2024-11-25
> >
> > Thank you for the further clarifications. My takeaways are as follows:
> > - The fundamental idea of GAL is similar to INAL
> > - There are some technical differences from INAL, and in particular GAL yields lower bounds for a specific target function as opposed to a function family
> > - There are still some technical limitations to GAL (only correlation loss, etc)
> > - GAL / INAL are useful in situations where you believe weak learning is the "bottleneck". I think this is an interesting / subtle point  re real-world significance, and probably worth discussing a bit somewhere in the paper or at least an appendix.
> >
> > I've updated my score to an 8.

---

> > > ### Author Response · Authors · 2024-11-29
> > >
> > > Thank you for your feedback and for increasing your score.

---

### Official Review · Reviewer_SB62 · 2024-11-04

**Soundness:** 3
**Presentation:** 3
**Contribution:** 3
**Rating:** 6
**Confidence:** 3

**Summary:**

This paper studies the role of initialization for learning dense parities with neural networks.
It is shown that the learnability by gradient descent heavily depends on the initialization through a notion of gradient alignment: if the initial gradient alignment is small, then it is hard for noisy gradient descent to learn the target.
In particular, for dense parities, initialization with Gaussian perturbations has small gradient alignment and thus prevents learning, while random Rademacher initialization is a special case that allows learning.
The results are supported by theoretical analysis and numerical experiments.

**Strengths:**

1. The paper is well written and easy to follow. The organization is very clear.
2. The paper provides an interesting perspective on the role of initialization for learning dense parities, and the observed phenomenon seems very different from the that in the case of sparse parities.
3. The negative result holds beyond the case of dense parities, and potentially leads to a better understanding of the role of initialization for broader settings.

**Weaknesses:**

1. The negative result seems to hold only for noisy (stochastic) gradient descent. How about deterministic gradient descent? There is a gap regarding this, and it would be helpful if the authors could discuss this.
2. It seems that the poly(d) width is crucial. It is mentioned in Section 4 that this ensures the linear separability of the full parity function. It would be helpful if the authors could discuss more about this design choice, and also what happens for narrower networks.
3. It is unclear how the choice of the loss function affects the results. It would be helpful if the authors could discuss this.

**Questions:**

1. Intuitively speaking, what is the role of the gradient alignment at initialization? How does it affect the learning process?
2. In other common settings, such as regression with single- or multi-index models, what is the typical value of the gradient alignment at initialization?
3. In the sparse parity case, what's the role of gradient alignment at initialization?

---

> ### Author Response · Authors · 2024-11-19
> **Rebuttal to reviewer SB62**
>
> We thank the reviewer for the constructive comments. We address the questions in the list below.
>
> - *Negative result for deterministic gradient descent (without additional noise).*
>
> Please, see the general reply.
>
> - *Positive result: width of the network. What happens for narrower networks?*
>
> The reviewer is correct that our positive proofs require a certain minimum poly$(d)$ network width. We do not know how to remove this limitation. Empirically, decreasing the network width results in worse learning time and test accuracy. However, this worsening seems to occur in a continuous manner.
>
> - *How does the choice of the loss function affect the results?*
>
> For the theoretical results, the choice of the loss function affects the distribution of the junk flow, i.e. the dynamics of a network that is trained on random labels (see Definition 6). Lemma 3 holds regardless of the loss function. However, the value of gradient alignment (see (49)) has to be bounded separately for a new loss function (this is inconvenient, but related to the fact that our techniques are sensitive enough to distinguish Rademacher and Gaussian initialization). Bounding GAL for different losses is more challenging, especially for later time steps. In the case of correlation loss it is easier to relate values of GAL at initialization and at later times. Empirically, both positive and negative results seem to hold for other losses, including cross-entropy and hinge loss. For example, Figure 2 shows the values of GAL at different times for the hinge loss seemingly decaying exponentially with input dimension $d$.
>
> - *Intuitively, what is the role of the gradient alignment at initialization? How does it affect the learning process?*
>
> The gradient alignment at initialization measures the strength of the target function's signal in the initial gradient. Specifically, it reflects how the model's gradient deviates from that of a network with random labels, which points in a random direction. This metric determines (at least in the negative sense) the time complexity of escaping initialization to achieve non-trivial learning. Our theorem shows that if the initial gradient alignment is small, it will take a long time to grow and escape the initialization.
>
> - *In other common settings (e.g. single/multi-index models) what is the typical value of the gradient alignment at initialization?*
>
> For single/multi-index models and for squared or correlation loss, the gradient alignment is related to the Hermite expansion of the target link-function, and to the information exponent (i.e. the smallest non-zero Hermite coefficient, as defined in https://arxiv.org/pdf/2003.10409, https://arxiv.org/pdf/2210.15651). For , for a single-index model, we have data $x \sim N(0,I_d) $, $y = f(w^Tx)$, where $w \in R^d$ such that $|| w ||_2 =1$ and $f:R \to R$ has information exponent $s <<d$. Assuming a two-layer neural network with Gaussian initialization and with activation $\sigma$ with non-zero s-th and (s-1)-th Hermite coefficient, one can show that the GAL at initialization is of order $\theta ( Pd^{-(s-1)/2} )$, where P is the total number of weights in the network. Thus, it correctly captures the complexity of escaping the initial saddle for target link function with high information exponent.
>
> - *In the sparse parity case, what’s the role of gradient alignment at initialization?*
>
> For sparse k-parities, the GAL is related to the size of the support, k. For a parity of degree $k<<d$ and for a two-layer ReLU network with Gaussian initialization , the GAL is of order $\theta(P d^{-k} )$, where $P$ is again the total number of weights in the network. Similarly to the single-index case, also for sparse parities, the initial GAL captures the complexity needed to identify the parity support, which corresponds to the complexity of achieving weak learning.

---

> > ### Comment · Reviewer_SB62 · 2024-11-25
> >
> > I thank the authors for the detailed response. My questions have been addressed, and I will keep my original score.

---

> > > ### Author Response · Authors · 2024-11-29
> > >
> > > Thank you for your reply and for supporting our work.

---

### Official Review · Reviewer_oGKm · 2024-11-04

**Soundness:** 3
**Presentation:** 2
**Contribution:** 2
**Rating:** 6
**Confidence:** 4

**Summary:**

This paper considers learning full parity functions over hypercube with neural networks. Their main results are twofold: : 1) Neural networks initialized near the Rademacher distribution can successfully learn full parity functions. 2) Conversely, neural networks with low gradient alignment at initialization, when trained using noisy gradient descent (GD) or stochastic gradient descent (SGD), fail to learn the task. Notably, Gaussian initialization results in poor gradient alignment, which significantly hinders network performance in this context.

**Strengths:**

The manuscript is reasonably well-written. The primary contribution—demonstrating the gap between Rademacher and Gaussian initialization—is an interesting result, particularly in the context of how initialization impacts neural network learning.

**Weaknesses:**

* I would be careful when claiming negative results as they depend on additional noise injected to SGD, i.e., Z^t terms in Eq (3), which seems unnatural compared to other theoretical results related to ReLU networks. Is there a hope to extend to the negative results to SGD without additional noise injection?

*  The positive result in the paper notably involves training only the output layer, meaning there is no feature learning. However, it is unclear whether the authors consider training input layer weights for their negative results, which introduces ambiguity in the interpretation of these results. Could the authors clarify this aspect in the rebuttal?

* Since the positive results only require training the output layer weights, it is uncertain if the sample complexity of their SGD is tight. Could the authors discuss whether their sample complexity is indeed tight, and/or whether it might improve if the initial layer were trained as well? Additionally, what aspects of their analysis limit extending training to the input layer?

* The authors introduce a new measure, termed "initial gradient alignment," for analyzing their negative results. However, in the paragraph following Definition 2, they note that this measure aligns with the previously studied "Label-Gradient Alignment." Could the authors elaborate on how their analysis diverges from or builds upon these earlier works?

**Questions:**

See the Weaknesses part.

---

> ### Author Response · Authors · 2024-11-19
> **Rebuttal to reviewer oGKm**
>
> We thank the reviewer for the constructive comments. We address the questions in the list below.
>
> - *Noisy-SGD vs. SGD without additional noise.*
>
> We refer to the general response for a discussion on the gradient noise.
>
> - *Training of the input layer in the negative result*
>
> Yes, our negative result holds for joint training, thus proving that if the initial GAL is small, feature learning does not occur.
>
> - *Sample complexity of the positive result: tightness and improvements in case of joint training.*
>
> Our positive result for hinge loss holds also for training hidden weights (this is stated in Section 4.2). However, we do not get any theoretical advantage out of training the hidden weights and the same result would hold training output weights only. We included this analysis only for hinge loss for the sake of simplicity. We do not think our sample complexity and network width are tight. Empirically, training both layers results in faster convergence and better accuracy. It would be interesting to understand it theoretically, but we do not address it in the paper.
>
> - *Comparison between GAL and previously defined measures (particularly LGA).*
>
> Our definition of GAL is loss-dependent. For square loss or correlation loss, the GAL at initialization aligns with the Label-Gradient-Alignment (LGA) proposed in prior work. Our approach improves on previous work in two ways:
> 1. Our loss-dependent GAL can capture important differences between losses. As an example, in the revision (Appendix E.2, Figure 8) we added an experiment with the following Boolean function: $f(x) =  \frac 18 x_1x_2x_3 + \frac 3 8 x_1x_2x_4 + \frac 14  x_1x_3x_4 + \frac 14 x_2x_3x_4$. In https://arxiv.org/pdf/2407.05622, the authors show that this function is learned more efficiently by SGD with L1-loss than with L2-loss. Our experiment show that such difference is captured by our GAL, while it cannot be captured by LGA since it is not loss-dependent. Although a full comparison of our GAL with prior metrics across various tasks is beyond the scope of this paper, which focuses on high-degree parities, we are hopeful that our GAL can reveal interactions among initialization, loss, and target in a broader context.
> 2. To the best of our knowledge, previous works do not provide theoretical bounds on learning complexity based on initial alignment. Our work offers a theoretical lower bound that depends directly on the GAL.

---

> > ### Comment · Reviewer_oGKm · 2024-11-23
> >
> > Thank you for the detailed responses. From what I understand, GAL extends the LGA proposed in previous work. In this context, it might have been better framed as a generalization rather than a novel metric, given that certain versions of it have appeared in the literature. Regarding Noisy-SGD vs. SGD, it seems prior work has also utilized Noisy-SGD to establish certain lower bounds. The manuscript’s conclusions about initialization are intriguing, and based on this, I am raising my score from 5 to 6.

---

> > > ### Author Response · Authors · 2024-11-29
> > >
> > > Thank you for your response and for increasing your score. We also appreciate your suggestion to frame the GAL as a generalization of the LGA, and we incorporate this perspective in the final version of the paper.

---

### Official Review · Reviewer_3tT8 · 2024-11-07

**Soundness:** 3
**Presentation:** 2
**Contribution:** 3
**Rating:** 6
**Confidence:** 3

**Summary:**

The paper provides a theoretical analysis of learning full parities (all bits parities) $k=d$ with SGD from (small perturbed) Rademacher initialization. To my knowledge, there was no analysis known. The case where it was observed that SGD succeeds with this initialization.
The authors further make some important observations on the role of initialization as to when we succeed in doing so, and it turns out that the initialization is curicial. To complement, the authors provide also hardness results for learning *almost full parities* with correlation loss and the initialization that has poor initial alignment, proposed by some metric that they define. For example, as a corollary, they can show that Gaussian initialization is poor.

1. **positive**: For the perturbed Radamacher initialization with noise level $\sigma=O(d^{-1})$, a 2-layer ReLU trained with SGD (Definition 3) with either correlation or hinge loss, succeeds in strongly learning the **full (all bits) parity** function within poly(d) samples/iterations.
2. **negative**: A general result that applies to a broader class of functions (including **almost full parities**) in terms of initial gradient alignment (GAL0 "signal at initialization"), where they show noisy SGD (Definition 3) has an edge over $1/2$ (null error) that is bounded by some function of poly(GAL0) and some other relevant hyperparameters, so if this GAL0 is exponentially decaying, we fail to weakly learn even with this initialization in poly$(d)$ samples. One example is Gaussian initialization.

**Strengths:**

See summary. The paper has a nice negative result for correlation loss SGD trained network with poor alignment initialization.

**Weaknesses:**

My main criticism of this paper is two-fold.
1. The paper, throughout the beginning (including the abstract), gives the impression that they show the success of SGD for even almost full parity $k=d-O(1)$, but the positive result is only shown for the full parity case $k=d$. For example, the abstract lines (14-17). Later also in Lines (65-77). I found this very confusing when I got to the main formal results. (See also my Question 1)
2.  To me the real contribution is the negative result. Particularly, showing that with Gaussian initialization, we fail with SGD on correlation loss. As far as I knew, this was open (see [Abbe & Boix-Adsera] discussion). However, the fact that this only holds for correlation loss works against its generality. Also, the presentation of the paper should be in a way that it highlights this as a primary contribution in a clear way.

Overall, both in terms of contributions and presentation, this paper requires more for me to recommend acceptance. Therefore, I am leaning toward a weak rejection.

**Questions:**

1. How big is this poly($d$) for positive results for sample complexity? This is my other minor quibble. The phenomenon was already known that it is possible to learn, so the value in positive results is in doing a careful analysis. Or at least empirically observe and report what is the complexity of learning full parity.
2. What are the main techniques used in upper bounding the success probability in terms of initial gradient alignment? Also, an intuitive way of thinking why initial alignment is poor for Gaussian initialization but good for Rademacher or its "mildly" perturbed variant? To me, this is the main interesting contribution, but I did not find a clear description of the key ideas in the main text.

**Suggestion / Typos**
1. Perhaps it is worth expanding on the technique you used as I suggested above. And also for future work on deeper networks and other losses, there could be a much closer discussion on what are the difficulties.
2. In Line 237, the full parity function has missing $x_i$ in the product. Also, for positive result, it would be much more helpful to just explicitly define $f$ (full parity) in the Theorem statements only. Or at least clearly refer to it in an equation as the result only applies to just that function.

---

> ### Author Response · Authors · 2024-11-19
> **Rebuttal to reviewer 3tT8**
>
> We thank the reviewer for the constructive comments. We address the questions in the list below.
>
> - *Positive result for almost full parities and high-lighting of main contributions.*
>
> Please, see the general response for an extension of our positive result to almost full parities. We thank the reviewer for the feedback on our presentation. We will include a positive result for almost full parity and correct the flow accordingly. We will also highlight the negative result as our main contributions in the revision.
>
> - *Sample complexity for positive result.*
>
> For learning the full parity with a 2-layer ReLU network using SGD with Rademacher initialization and correlation loss, we demonstrate that $\tilde O(d^7)$ samples are sufficient (Theorem 4, part 3). We can further improve such sample complexity if we instead use a clipped-ReLU activation and obtain an upper bound of $\tilde O(d^3)$ samples (Remark 1). We remark that previous results on full parity learning concerned only weak learning and noisy full-batch gradient descent ([Abbe,Boix,’22]), thus with no bound on the sample complexity.
>
> - *Negative result: main techniques and difficulties.*
>
> To upper bound the success probability in terms of gradient alignment, our approach consists of the following key steps:
>
> 1. *Defining the "junk flow":* This represents the training dynamics of a network trained on random noise.
>
> 2. *Coupling the GD dynamics with the junk flow:* We show that if the Gradient Alignment (GAL) remains small along the junk flow, then the noisy-GD dynamics stay close to the junk flow in total variation (TV) distance, meaning that the network does not learn (Lemma 3).
>
> 3. *Proving that GAL is small along the junk flow:* For correlation loss, we demonstrate that the GAL remains small along the junk flow. In particular, under certain conditions (Corollaries 2 and 3), if the GAL is initially small, it continues to stay small throughout the junk flow.
>
> 4. *Almost-full parities:* For almost-full parities, perturbed initializations, and 2-layer ReLU networks and correlation loss, we show that the GAL along the junk flow remains small.
>
> Notably, steps (1) and (2) apply to any architecture with a linear output layer, symmetric initialization, and any loss function. However, step (3) is currently limited to the correlation loss, as tracking junk flow dynamics for other losses is more complex. Empirically, we also observe that the GAL remains small for the hinge loss (Figure 2, left). Extending step (3) to other losses would require new techniques to analyse the junk flow dynamics, and it is an interesting direction for future work. Bounding the initial GAL for step (4) involves precise calculations, which we limit in this paper to 2-layer ReLU networks. Extending this to deeper networks would establish a negative result for learning almost-full parities in deeper architectures, though we do not expect such an extension to be technically easy. In the revision, we will move the junk flow definition to the main text, outline the main techniques and challenges, and, if space permits, include a proof outline for Theorem 7.
>
> - *Intuition on why initial alignment is large for Rad. init and poor for Gaussian perturbation.*
>
> 1. *GAL large for Rademacher init:* Under Rademacher initialization the gradient of each hidden neuron i is related to the correlation of the full parity with the Boolean majority function $ {\rm Maj}(x)= {\rm sgn}(\sum_{i=1}^d x_i)$  on a uniform input in $\{\pm 1\}^d$. (see e.g., [O’Donnel,'14]). This correlation (at least for odd d) is of the order $1/ \sqrt(d)$. Extension of the proof to almost-full parities is related to the fact that the majority also can have 1/poly(d) correlation with almost-full parities.
>
> 2. *GAL small for Gaussian init:* For Gaussian initialization, the continuity of the distribution allows us to use a series expansion. Using such expansion, we find that low order terms cancel out due to the properties of the parity function, and high order terms are exponentially small.
>
> 3. *Perturbed initialization:* For a perturbed initialization $r+\sigma g$, with $r$ Rademacher and $g$ Gaussian, roughly, for $\sigma$ sufficiently small, $r$ is the dominant term, while for $\sigma$ sufficiently large, the Gaussian part becomes dominant. It seems interesting to determine the behaviour also in the intermediate range of $\sigma.$
>
> - *Typos and suggestions.*
>
> We thank the reviewer for pointing out a typo in line 237 and for suggesting improvements to the presentation, which we will include in the revision.

---

> ### Comment · Reviewer_3tT8 · 2024-11-25
>
> Thank you for your detailed response!
>
> 1. Could you help me further see the argument for high GAL for Rademacher init? I am familiar with the claim from O'Donnell. Also, it would be good to add this in the main text.
> 2. What is $d$ in Figure 7? As far as I found it, only Fig 7 is the experiment with 2-layer MLP, correct? The rest (Fig 5 and the main paper experiments) are with 4-layer MLPs.
> 3. Width: What width have you used in Fig 7? Also, what width have you used for Fig 5 for different $d$ s? Is it the same as (512,512,64) for the hidden layers?
> 4. It seems that for $d=150,200$, it did not succeed in strong learning. Should I view Fig 7 as a success or failure? I am slightly confused here. Overall, it would be good to conduct experiments with just correlation loss with an appropriate width on a 2-layer MLP and demonstrate success with higher $d$ s, to at least verify the theoretical results as they are.
> 5. Also, is the sample complexity in experiments really that sensitive to the activation used (as you mentioned in theory)? Also, how real is the dependence on width and iteration complexity derived in the formal results? In experiments, do you see similar blow-ups in the complexities when (a) going from full to almost full parties and (b) changing activations?
>
> Overall, I am still confused with the experiments, and perhaps the addition of more comprehensive experiments in the final version would be nice. The main thing is, besides the initialization, how sensitive is the effect of the activation and loss on the required width and sample complexity? Also, as stated, I request the authors to at least add
> 1. One convincing experiment in which we succeed in learning with higher $d$ s, ideally using the same settings as in the theoretical results.
> 2. the intuitive explanation behind low and high GAL for Gaussiand and Rad initialization respectively, in the main text in an easily accessible subsection/paragraph.
>
> The paper has an important contribution. Hoping the authors will improve their presentation further, I raise my score to 6 and support acceptance.
>
> PS: for Fig 8, where the function output is non-binary, is the null/ junk label distribution to measure GAL still Rad(1/2), as used in the GAL definition eq 2?

---

> ### Author Response · Authors · 2024-11-29
>
> Thank you for your engagement and for increasing the score. Regarding your further questions:
>
> 1. *Further details about the argument for high GAL for Rademacher initialization.*
>
> Let us give the following simplified argument for the GAL of the first (hidden) layer.
> The gradient of the first layer’s weights can be written as:
> $$ \partial_{w_{ji}} L = - a_i E_x [ f(x) x_j 1(w_i x+b_i)], $$ where $f(x) = \prod_{i=1}^d x_i$ is the full parity function.
> For simplicity assume that $b_i=0$, and let us drop the subscript $i$ for clarity. Then  $1(wx+b) =  \frac 12 + \frac 12 {\rm Maj} (w \odot x) $, where $x \odot w$ denotes the Boolean vector with j-th entry $w_j x_j$, and ${\rm Maj}(w \odot x) : =  {\rm sgn}(\sum_{k=1}^d w_k x_k)$ is the majority function. Now, consider the Fourier-Walsh expansion of ${\rm Maj}$  as ${\rm Maj} (w \odot x) : = \sum_{S \subseteq [d]}  \hat {\rm Maj} (S) \prod_{k \in S} x_k w_k$, where $ \hat {\rm Maj} (S)$ are the Fourier-Walsh coefficients. Replacing this in the formula above, and using the orthogonality of the Fourier basis elements, we obtain:
> $$ \partial_{w_{j}} L = - a_i \prod_{k\ne j} w_k \hat {\rm Maj} ([d] \setminus  j ) ,$$
>  and accordingly $\partial_{w_{j}} L^2=a_i^2  \hat {\rm Maj} ([d] \setminus  j )^2$.
> If $d$ is even, the majority coefficient of $[d]\setminus j$ has magnitude $1/\mathrm{poly}(d)$ (using similar calculations as in Chapter 5 of [O’Donnell,'14]) and the GAL lower bound follows for this coordinate of the gradient.
> For the second layer weights, analogous calculations can be made using ReLU instead of the threshold.
>
> Please note that in the paper we do not compute the GAL for Rademacher initialization. By our negative result, sufficiently high GAL is necessary for strong learning, but in general it is not sufficient. Rather, our positive proof proceeds by direct analysis of the (S)GD process. An important element of the positive proof is the quantity (4), which shows up in the gradient computation in (20). Estimating (4) involves binomial coefficients, which can be seen in the proofs in appendix A.3, e.g., Lemma 1. We can add the GAL computation for Rademacher as outlined above in a later revision if you think it would be informative.
>
> 2. *What is d in Figure 7? As far as I found it, only Fig 7 is the experiment with 2-layer MLP, correct? The rest (Fig 5 and the main paper experiments) are with 4-layer MLPs.*
>
> In figure 7, d=50. We added it in the caption in the revision, thanks for pointing this omission out. Yes, Figure 7 (and now Figure 9) uses 2-layer MLPs, while the rest use 4-layer MLPs.
>
> 3. *What widths were used in Fig. 7 and Fig. 5 for different d’s? Is it the same as (512,512,64) for the hidden layers?*
>
> Yes, in Figure 5 (and 7) the widths of the hidden layers are kept constant, and only the input layer’s dimension is changed.
>
> 4. *It seems that for d=150, 200, it did not succeed in strong learning. Should I view Fig 7 as a success or failure?*
>
> We assume you meant Figure 5, please let us know if we are mistaken. We added Figure 5  as a negative result for weak learning with perturbed initialization. In particular, while with $d=50$ (figure 1) a perturbation of e.g. $\sigma=0.1$ allows to achieve performance close to Rademacher, for larger $d$, the performances seem to deviate, and for $d=200$, only with Rademacher initialization the network seems to pick signal in the first iterations. To achieve strong learning with Rademacher initialization for larger $d$ we have to increase the width of the hidden layers, see new plot (Figure 9).
>
> 5. *Overall, it would be good to conduct experiments with just correlation loss with an appropriate width on a 2-layer MLP and demonstrate success with higher d’s, to at least verify the theoretical results as they are.*
>
> We thank the reviewer for the suggestion. We added a preliminary plot for this, see Figure 9 in the Appendix of the revision. We show that a 2-layer network with clipped-ReLU activation and $d^2$ hidden neurons, with only the output layer trained by SGD with the correlation loss can achieve accuracy 1 for $d = 50, 100, 150, 200$. We will run more experiments and reorganize the order of the figures in the final version of the paper.

---

> ### Author Response · Authors · 2024-11-29
>
> 6. *Also, is the sample complexity in experiments really that sensitive to the activation used (as you mentioned in theory)? Also, how real is the dependence on width and iteration complexity derived in the formal results? In experiments, do you see similar blow-ups in the complexities when (a) going from full to almost full parties and (b) changing activations?*
>
> We tried training the same 2-layer network with $d^2$ hidden neurons as in the previous point, using the same setting (SGD with the correlation loss trained on the output layer only), while replacing the clipped-ReLU activation with ReLU activation. We observed that the network does not achieve accuracy 1: from our preliminary experiments it achieves accuracy around 0.55 for d=200, suggesting that the activation does play a role in the complexity of learning, in particular that in our setting ReLU requires more hidden neurons than clipped-ReLU.
>
> From preliminary experiments it is also apparent that almost-full parities require more resources than full parities. We intend to investigate further and add our findings to a later revision.
>
> These findings are subject to some caveats:
> - We do not make theoretical claims that our bounds are tight.
> - We are somewhat confident that our empirical positive findings are qualitatively robust. In particular, we managed to obtain good accuracies for full parity for various numbers of layers, with or without training hidden weights, and different losses (square, cross-entropy, hinge). The main necessary factor seems to be the Rademacher initialization (also a piecewise linear activation might play a role). However, the particulars can change when the architecture is changed. For example, the difference between clipped-ReLU and ReLU might be empirically validated in the exact setting of our theorem, but disappear (or even reverse) for other losses and architectures.
>
> 7a. *I request the authors to at least add:
> One convincing experiment in which we succeed in learning with higher d’s, ideally using the same settings as in the theoretical results.*
>
> We added a preliminary plot in Figure 9. Please, let us know if you have any further comment about it.
>
> 7b. *the intuitive explanation behind low and high GAL for Gaussiand and Rad initialization respectively, in the main text in an easily accessible subsection/paragraph.*
>
> We thank the reviewer for this suggestion, which will improve the presentation of our paper. We will add a paragraph with such an explanation in the informal contribution section.
>
> 8. *For Fig 8, where the function output is non-binary, is the null/ junk label distribution to measure GAL still Rad(1/2), as used in the GAL definition eq 2?*
>
> We thank the reviewer for the question. Yes, in the current plot, we used the same definition. We remark that with the squared loss any junk label distribution such that E[y_junk] = 0 gives the same gradients. We will run experiments replacing the junk flow Rad(1/2) with the marginal distribution of f.

---

### Official Review · Reviewer_hKEm · 2024-11-10

**Soundness:** 3
**Presentation:** 3
**Contribution:** 2
**Rating:** 5
**Confidence:** 3

**Summary:**

This paper considers the learning of full parity and nearly full parity when the input is sampled from the $d$-dimensional hypercube. It shows that when a two-layer ReLU network is initialized with the first layer weights following the Rademacher initialization (uniformly distributed $\pm 1$ values), and the noise level $\sigma$ is sufficiently small, the network can learn the full parity problem with polynomial sample complexity when trained with GD/SGD of the second layer of the network. On the other hand, if the distribution of the weights at initialization deviates significantly from the uniform distribution on the hypercube (e.g., using a Gaussian distribution), the network is unable to learn the full or nearly full parity.

**Strengths:**

**Novelty** :
While sparse parity problems ($k=O_d(1)$) are well studied in the context of representation learning with two-layer neural networks, I did not know that full parity can be learned with polynomial complexity. Thinking analogously to the sparse parity problems, we would need to train the first layer matrix but the gradient would vanish as the order of the parity gets larger. However, for the full parity, if the Rademacher initialization used, the gradient of the second layer is exactly computed and does not vanish, and furthermore, changing the weight of the second layer suffices to solve the problem. The proof largely depends on the Rademacher initialization, but even so, the analysis seems novel and interesting.

**Interesting phase transition**:
This paper shows that by adding small noise to the initialization then learning full and nearly full parities becomes significantly difficult. In addition to the Gaussian initialization, this paper extend this negative result to the midpoint between the Rademacher initialization and Gaussian initialization by introducing the quantity that characterizes the initial alignment of the gradient.

**Weaknesses:**

**Motivation**:
Although this paper trains the two-layer neural networks, the mechanism of learning is significantly different from sparse parity.
In the sparse parity problems, people mainly discusses how the first layer weights align with the meaningful subspace.
On the other hand, every direction is equivalent in this full parity setting, thus I am not sure whether this paper is motivated as a feature learning paper.
Thus the paper should explain why solving full parity how neural network is motivated, especially connections to real-world phenomena.

**Generality of the result**:
As the lower bound suggests, this result is specific to the Rademacher initialization.
The fact that the gradient does not vanish and such a transformation of equations is possible is quite interesting. However, if the results no longer hold after adding a small perturbation to the initial weights, then I wonder if those results lose their general significance.
As an analysis this is quite interesting, though.

**Questions:**

**Extension to $(d-k)$-parity**
In the lower bound, we are able to handle the $(d-k)$-parity. On the other hand, the upper bound is stated only for full parity. Given the complexity theory and feature learning literature, I would like to see the corresponding between $k$-parity and $(d-k)$-parity. Can you derive something like $d^{\Theta(k)}$ complexity? Or $k$-parity and $(d-k)$-parity are essentially different for $k\geq 0$?

**Connection to real-world problems**
I would like to know if there are any real-world domains where learning high-degree or full parities is relevant,

---

> ### Author Response · Authors · 2024-11-19
> **Rebuttal to reviewer hKEm**
>
> We thank the reviewer for the constructive comments. We address the questions in the list below.
> - *Relevance for feature learning.*
>
> Indeed, our positive result does not address feature learning, as it depends only on training the second layer (in the case of the hinge loss the positive result holds also when training the first layer, but it is not necessary). In contrast, our negative result demonstrates that, even when both layers are trained using perturbed or Gaussian initialization and with the correlation loss, feature learning does not occur. Therefore, our negative result is relevant for highlighting the limitations of feature learning under these conditions.
>
> - *Relevance for real-world problems.*
>
> High-degree parity functions are simple yet challenging targets because learning them requires capturing global associations among the inputs. These functions cannot be meaningfully approximated using low-degree representations, making them a clear and well-defined benchmark for evaluating learning algorithms. As a result, they have become a key focus in recent deep learning research, particularly in studies involving global relationships. In relation to “real-world” problems, beyond offering a simple test for “global reasoning”, they relate to more recent logic and reasoning benchmarks, such as related to arithmetic, coding, genomics, algorithms or networks applications.
>
>
> - *Extension of positive result to d-k parity.*
>
> We refer to the general response for a discussion on the extension of the positive result to d-k parities. We additionally note that indeed almost full d-k parities seem learnable with d^\Theta(k) complexity under Rademacher initialization. This is the case even though feature learning is not involved.

---

> > ### Author Response · Authors · 2024-11-29
> >
> > Thank you once again for taking the time and effort to review our paper. We would like to kindly highlight that we have extended our positive result to d-k parities, and we refer to the general response for details.
> >
> > As the discussion period is approaching its conclusion, we would greatly appreciate it if you could let us know if you have any additional questions or concerns that we can address.

---

### Author Response · Authors · 2024-11-19
**General Response to all Reviewers**

We thank all reviewers for the useful feedbacks and for appreciating our work. We address here few questions raised by multiple reviewers.

- *Revision*

We posted a revision of the paper including an additional experiment (Figure 8, Appendix E) for a specific target function, showing that our GAL can capture differences in the loss function used for training. It appears that our phrasing in the abstract and introduction was misleading with regard to the positive result. We apologize for that. We will change the phrasing in the revision, also taking into account the extension of our positive result to almost-full parities, see below.

- *Extension of positive result to (d-k)-parities (hKEm, 3tT8, mApU)*

Several reviewers (hKEm, 3tT8, mApU) inquired about an extension of our positive result to cover almost-full parities (e.g., degree d−O(1)).
We initially opted not to include a positive result for nearly full parities. However, we believe that our proof can indeed be extended to encompass these cases. In particular, our bound on the number of training steps and hidden neurons needed depends on a quantity that we defined as $\Delta$ in eq (15) (roughly this is the correlation achieved by a single ReLU neuron with the target), and this definition can be generalized so that we have a bound for $d-a$ parities in terms of some general correlation $\Delta^{(a)}$. It remains to bound $\Delta^{(a)}$ for almost-full parities. We computed this for $(d-1)$-parities and $(d-2)$-parities, from which we obtain that with a ReLU network of order $d^6$ width and $d^{11}$ samples we can also learn both $d-1$ and $d-2$ parities (as compared to $d^4$ width and $d^7$ samples for the full $d$-parity).

We will submit a revised version of our paper that includes cases $k=d-1$ and $k=d-2$. We are working on an updated version for any $k=d-O(1)$ (which has some interesting technical complications) and plan to submit it either for the camera-ready version or in a later version of this paper.

- *Noisy SGD vs. SGD without additional noise (Reviewer oGKm, SB62)*

Indeed, our separation between Rademacher and Gaussian/perturbed initialization holds for noisy-GD, where the `noisy’ refers to additional Gaussian noise added to the gradient at each step. We do need the Gaussian noise for the total variation bound in Lemma 3, specifically it is needed to obtain a linear dependence in the training length T. With no noise and for Lipschitz networks, we could use a direct bound on the norm of the gradient, which would however give an $\exp(T)$ dependence. We remark that several known lower bounds in deep learning use this definition of (S)GD with noise (e.g. https://arxiv.org/pdf/2309.03800, https://arxiv.org/pdf/2208.03113, https://arxiv.org/pdf/1812.06369). We will add a remark on this and we will clarify the need of the gradient noise in the contributions section. We further point out that our negative results for parity hold for the magnitude of gradient noise as small as $\tau=\exp(-o(d))$, that is any function larger than inverse exponential, e.g., for $\tau=\exp(-\sqrt{d})$. It can be argued that such small additional noise is not significant compared to, e.g., floating point rounding errors, though we do not have a rigorous proof.

---

> ### Author Response · Authors · 2024-11-20
> **New revision**
>
> We uploaded a new revision of the paper. In this revision:
> * We added an additional experiment (Figure 8, Appendix E) for a specific target function, showing that our GAL can capture differences in the loss function used for training.
> * We clarified the language in the abstract and the introduction regarding the positive results.
> * We added a positive result for ReLU activation and correlation loss for almost full parities $d=k-1$ and $d=k-2$, see Corollary 2 in Section 4.1.
> * We added a more precise calculation for a bounded ReLU variant, where $\Omega(d^2)$ hidden neurons are sufficient for learning of the full parity function. See Corollary 1 in Section 4.1
> * We made changes aiming to improve the presentation of positive results in Section 4.

---

> > ### Author Response · Authors · 2024-11-25
> > **Another revision**
> >
> > We uploaded another revision of the paper. We thank all reviewers for their engagement and we are happy to answer any outstanding questions in the remaining time of the rebuttal period.
> >
> > The changes in the most recent version include:
> > * Section 4: We completed the proof of the positive result for Rademacher initialization for $k=d-a$ for arbitrary constant $a$. Indeed, the complexity of our result (required width of the network and the number of steps in the SGD) for learning $(d-a)$-parities is $n^{\Theta(a)}$, complementing the analogous result for sparse parities. Some potentially interesting points:
> >     * The result still requires only one step of full GD and training only output weights.
> >     * As far as we understand, in our construction the linear separation in the output layer is not achieved by just one "good neuron". Rather, each neuron provides a weak signal to the gradient and the aggregation of many of them points the gradient in the "correct" direction.
> > * Section 5.1: We made changes in order to improve presentation, including adding the proof outline for the general lower bound in Theorem 6.
> > * Section 5.2: For the case of perturbed initialization, we initially included the GAL calculation only for hidden layer neurons. We now added the GAL bound also for output layer neurons. As a result, we have a rigorous lower bound for learning for sigma-perturbed Rademachers (for large enough constant sigma > 0), compare Theorem 8 with Theorem 7.
> >
> > For reference, the changes included in the previous revision:
> > * We added an additional experiment (Figure 8, Appendix E) for a specific target function, showing that our GAL can capture differences in the loss function used for training.
> > * We clarified the language in the abstract and the introduction regarding the positive results.
> > * We added a positive result for ReLU activation and correlation loss for almost full parities $d=k-1$ and $d=k-2$, see Corollary 2 in Section 4.1.
> > * We added a more precise calculation for a bounded ReLU variant, where $\Omega(d^2)$ hidden neurons are sufficient for learning of the full parity function. See Corollary 1 in Section 4.1
> > * We made changes aiming to improve the presentation of positive results in Section 4.

---

### Meta-Review · Area_Chair_VSEV · 2024-12-05

**Metareview:**

Separation results for the learnability of (almost-)full parity functions within the class of two-layer neural networks are presented, demonstrating how the success of (certain variants of) gradient-based methods crucially depends on the choice of the initialization distribution. Key issues, such as the positioning of some concepts introduced in the paper relative to existing work, as well as presentation-related concerns, were discussed and clarified during the review process. Reviewers found the separation results both compelling and instructive, praised the theoretical rigor and empirical work, and ultimately voted for acceptance.

**Additional Comments On Reviewer Discussion:**

See above:

Key issues, such as the positioning of some concepts introduced in the paper relative to existing work, as well as presentation-related concerns, were discussed and clarified during the review process.

---

### Decision · Program_Chairs · 2025-01-22

Accept (Poster)